



**The interpretation of temperature and salinity variables in numerical**
**ocean model output, and the calculation of heat fluxes and heat content**

4           by

Trevor J. McDougall[1], Paul M. Barker[1], Ryan M. Holmes[1,2],
Rich Pawlowicz[3], Stephen M. Griffies[4] and Paul J. Durack[5]
[1]School of Mathematics and Statistics, University of New South Wales,
Sydney, NSW 2052, Australia
[2]Australian Research Council Centre of Excellence for Climate Extremes, University
of New South Wales, Sydney, NSW 2052, Australia
[3]Dept. of Earth and Ocean Sciences, University of British Columbia,
Vancouver, B.C. V6T 1Z4, Canada
[4]NOAA/Geophysical Fluid Dynamics Laboratory, Princeton, New Jersey, USA
[5]Program for Climate Model Diagnosis and Intercomparison, Lawrence Livermore
National Laboratory, Livermore, California, USA
Corresponding author email: Trevor.McDougall@unsw.edu.au
Full address
Trevor J McDougall
School of Mathematics and Statistics
University of New South Wales, NSW 2052, Australia
tel: +61 2 9385 3498
fax: +61 2 9385 7123
mob: +61 407 518 183
keywords    ocean modelling, CMIP, ocean model intercomparison, TEOS-10,
EOS-80,





**Abstract**

The 2010 international thermodynamic equation of seawater, TEOS-10, defined the
enthalpy and entropy of seawater, thus enabling the global ocean heat content to be
calculated as the volume integral of the product of in situ density, $\rho$, and potential
enthalpy, $h^0$ (with reference sea pressure of 0 dbar). In terms of Conservative
Temperature, $\Theta$, ocean heat content is the volume integral of $\rho c_p^0 \Theta$, where $c_p^0$ is a
constant "isobaric heat capacity".
However, several ocean models in CMIP6 (as well as all of those in previous
Coupled Model Intercomparison Project phases, such as CMIP5) have not been
converted from EOS-80 (Equation of State - 1980) to TEOS-10, so the question arises of
how the salinity and temperature variables in these models should be interpreted. In
this article we address how heat content, surface heat fluxes and the meridional heat
transport are best calculated in these models, and also how these quantities should be
compared with the corresponding quantities calculated from observations. We
conclude that even though a model uses the EOS-80 equation of state which expects
potential temperature as its input temperature, the most appropriate interpretation of
the model's temperature variable is actually Conservative Temperature. This
interpretation is needed to ensure that the air-sea heat flux that leaves/arrives-in the
atmosphere is the same as that which arrives-in/leaves the ocean.
We also show that the salinity variable carried by TEOS-10 based models is
Preformed Salinity, while the prognostic salinity of EOS-80 based models is also
proportional to Preformed Salinity. These interpretations of the salinity and
temperature variables in ocean models are an update on the comprehensive Griffies et
al (2016) paper that discusses the interpretation of many aspects of coupled model
runs.



## 1. Introduction

Numerical ocean models simulate the real ocean by calculating the acceleration of fluid parcels in response to various forces, some of which are related to spatially-varying density fields, as well as solving transport equations for the two tracers on which density depends, namely heat content ("temperature") and dissolved matter ("salinity"). For computational reasons it is useful for the numerical schemes involved to be conservative, meaning that the overall amount of heat and salt in the ocean changes only due to fluxes across the ocean's boundaries. This is guaranteed even for the long runs required for climate studies to within a very small error, since these numerical models are designed on the basis of volume integrated tracer conservation. However, this apparent numerical success ignores some difficult theoretical issues with the equation set being numerically solved, related to the properties of seawater, which have only recently been widely recognized as a result of research that resulted in the Thermodynamic Equation of Seawater 2010 (TEOS-10). These issues mean that the intercomparison of different models, and comparison with ocean observations, needs to be undertaken with care.

In particular, it is now widely recognized that the traditional measure of heat content in the ocean, the so-called potential temperature, is not a conservative variable (McDougall, 2003). The potential temperature of a mixture of water masses is not the average of the initial potential temperatures, so potential temperature is "produced" or "destroyed" by mixing within the ocean's interior. This empirical fact is an inherent property of seawater, and so treating potential temperature as a conservative tracer (as well as making certain other assumptions related to the modelling of heat and salt) results in inherent contradictions, which have been built into most ocean models to varying degrees.

These contradictions have always existed, but have generally been ignored or overlooked, probably because many other oceanographic factors were thought to be of greater concern. However, as global heat budgets and their imbalances are now a critical factor in understanding changes in climate, it is important to examine the consequences of these assumptions, and perhaps correct them even at the cost of





introducing problems elsewhere. This is particularly important when heat budgets are
being compared between different models, and with similar calculations made with
observed conditions in the real ocean.

The purpose of this paper is to describe these theoretical difficulties, to estimate

the magnitude of errors that result, and to make recommendations about resolving them
both in current and future modelling efforts. For example, the insistence that a model's
temperature variable is potential temperature involves errors in the air-sea heat flux in
some areas that are as large as the mean rate of global warming. A simple re-
interpretation of the model's temperature variable overcomes this inconsistency and
allows the coupled ocean-atmosphere model to conserve heat.

The reader who wants to skip straight to the recommendations on how the

salinity and temperature outputs of CMIP models should be interpreted can go straight
to section 6.

**2. Background**
***Thermodynamic measures of heat content***

It is well-known that in-situ temperature is not an appropriate measure of the

"heat content" of a water parcel because the in situ temperature of a water parcel
changes as the ambient pressure changes (i.e. if a water parcel is transported to a
different depth in the ocean). This change is of order 0.1C as pressure changes 1000
dbar, and is large relative to the precision of 0.01C required to understand deep ocean
circulation patterns. The utility of in situ temperature lies in the fact that it is easily
measured with a thermometer, and that air-sea boundary heat fluxes are to some degree
proportional to in situ temperature differences.

Traditionally, potential temperature has been used as an improved measure of

ocean heat content. Potential temperature is defined as the temperature that a parcel
would have if moved isentropically and without exchange of mass to a fixed reference
pressure (usually taken to be surface pressure), and can be calculated from measured
ocean in situ temperatures using empirical correlation equations based on laboratory
measurements. However, the heat capacity of seawater varies nonlinearly with





temperature and salinity (Fig. 1) and this results in non-conservative behavior under
mixing (McDougall (2003), section A.17 of IOC et al. (2010)). Thus the potential
temperature field in the ocean is subject to internal sources and sinks, and is thus not
conservative.

With the development of a Gibbs function for seawater, based on empirical fits to

measurements of known thermodynamic properties (Feistel (2008), IOC et al, 2010) it
became possible to apply a more rigorous theory for quasi-equilibrium thermodynamics
to study heat content problems in the ocean. As a practical matter, calculations can now
be made that allow for an estimate of the magnitude of non-conservative terms in the
current ocean circulation. By integrating over water depth this production rate can be
expressed as an equivalent heat flux per unit area.

Non-conservation of potential temperature was thus found to be equivalent to a

root mean square surface heat flux of about $60\,\mathrm{mW\,m^{-2}}$ (Graham and McDougall, 2013),
and an average value of $16\,\mathrm{mW\,m^{-2}}$ (see below) which can be compared to a current
estimated global-warming surface heat flux imbalance of about $300\,\mathrm{mW\,m^{-2}}$ (Zanna et
al., 2019). These equivalent heat fluxes and subsequent similar values are gathered into
Table 1 for reference. In the context of a conceptual ocean model being driven by known
heat fluxes, the absence of the non-conservation of potential temperature causes SST
errors seasonally in the equatorial region of about 0.5K (i.e. 0.5°C), while the error (in all
seasons) at the outflow of the Amazon is 1.8K (see section 9 of McDougall, 2003). With
different boundary conditions these temperature errors will show up in different parts
of the ocean model.

Unfortunately, no single alternative thermodynamic variable has been found that

is both independent of pressure, and conservative under mixing. For example, specific
entropy is produced in the ocean interior when mixing occurs, with the depth-integrated
production being equivalent to an imbalance in the air-sea heat flux of a root mean
square value of about $500\,\mathrm{mW\,m^{-2}}$ (Graham and McDougall, 2013), while specific
enthalpy is conservative under mixing at constant pressure, but is intrinsically pressure-
dependent.





However, it was found that a constructed variable, potential enthalpy
(McDougall, 2003) has a mean non-conservation error in the global ocean of only about
$0.3\,\mathrm{mW\,m^{-2}}$ (this is the mean value of an equivalent surface heat flux, equal to the depth
integrated interior production of potential enthalpy (Graham and McDougall, 2013)).
The potential enthalpy is the enthalpy of a water parcel after being moved isentropically
and adiabatically to the reference pressure 0 dbar where the temperature is equal to the
potential temperature, $\theta$, of the water parcel:
$$\tilde{h}^0\left(S_\mathrm{A},\theta\right) = h\left(S_\mathrm{A},\theta,0\,\mathrm{dbar}\right),\tag{1}$$
where the function $h$ is the specific enthalpy of TEOS-10 (defined as a function of
Absolute Salinity, in situ temperature and sea pressure) whereas $\tilde{h}^0$ is the potential
enthalpy function and the over-twiddle implies that the temperature input to this
function is potential temperature, $\theta$.   By way of comparison, the area-averaged
geothermal input of heat into the ocean bottom is about $86\,\mathrm{mW\,m^{-2}}$, and the interior
heating of the ocean due to viscous dissipation, is equivalent to a mean surface heat flux
of about $3\,\mathrm{mW\,m^{-2}}$ (Graham and McDougall, 2013).   Thus potential enthalpy, although
not a theoretically ideal conservative parameter, can be treated as one for all practical
purposes in ocean circulation problems.
Since potential enthalpy is not a widely-understood property, a decision was
made in the development of TEOS-10 to define a new variable, called Conservative
Temperature, $\Theta$, which would have units of temperature but be proportional to
potential enthalpy:
$$\Theta = \tilde{\Theta}\left(S_\mathrm{A},\theta\right) = \tilde{h}^0\left(S_\mathrm{A},\theta\right)\big/ c_p^0 ,\tag{2}$$
where the proportionality constant $c_p^0 \equiv 3991.867\,957\,119\,63\ \mathrm{J\,kg^{-1}\,K^{-1}}$, now embedded in
the TEOS-10 standard, was chosen so that the average value of Conservative Temperature
at the ocean surface matched that of potential temperature.   Although in hindsight other
choices (e.g., with fewer significant digits) might have been more useful, this value of $c_p^0$
is now built into the TEOS-10 standard and cannot be changed.
Note that at specific locations in the ocean, in particular at low salinities and high
temperatures, $\Theta$ and $\theta$ can differ by more than 1℃ (Fig. 2); the difference is a strongly





nonlinear function of temperature and salinity. $\Theta$ is by definition independent of
adiabatic and isohaline changes in pressure.

***Why is potential temperature not conservative?***
This question is answered in sections A.17 and A.18 of the TEOS-10 Manual (IOC
et al., 2010) as well as McDougall (2003), Graham and McDougall (2013) and Tailleux
(2015). The answer is that potential enthalpy referenced to the sea surface pressure, $h^0$,
which is an (almost totally) conservative variable in the real ocean, is not simply a linear
function of potential temperature, $\theta$, and Absolute Salinity, $S_A$ (and note that both
enthalpy and entropy are unknown and unknowable up to separate linear functions of
Absolute Salinity). If potential enthalpy were a linear function of potential temperature
and Absolute Salinity then the "heat content" per unit mass of seawater could be
accurately taken to be proportional to potential temperature, and the isobaric specific
heat capacity at zero sea pressure would be a constant. As an example of the
nonlinearity of $\tilde{h}^0(S_A, \theta)$, the isobaric specific heat at the sea surface pressure
$c_p(S_A, \theta, 0\,\mathrm{dbar}) \equiv h^0_\theta$ varies by 6% across the full range of temperatures and salinities
found in the World Ocean (Fig. 1). By way of contrast, the potential enthalpy of an ideal
gas is proportional to its potential temperature. An early expression for the enthalpy of
seawater was published fifty years ago by Connors (1970), and since 2010
oceanographers have switched from using potential temperature to Conservative
Temperature as part of the switch from EOS-80 to TEOS-10.

***How conservative is Conservative Temperature?***
This question is addressed in McDougall (2003) as well as in section A.18 of the
TEOS-10 Manual (IOC et al., 2010), Graham and McDougall (2013) and Tailleux (2015).
Enthalpy is conserved when mixing occurs but enthalpy does not posses the "potential"
property, but rather, an adiabatic and isohaline change in pressure causes a change in
enthalpy according to $\hat{h}_P = v$, where $v$ is the specific volume. This is illustrated in Fig. 3
where it is seen that for an adiabatic and isohaline increase of pressure of $1000\,\mathrm{dbar}$, the
increase in enthalpy is the same as that caused by an increase in Conservative



Temperature of more than 2.4°C. If enthalpy variations at constant pressure were a
linear function of Absolute Salinity and Conservative Temperature, the contours in Fig.
3 would be parallel equidistant straight lines, and Conservative Temperature would be
totally conservative. Since this is not the case, this figure illustrates the (small) non-
conservation of Conservative Temperature.

*Seawater Salinity*
To a degree of approximation which is useful for many purposes, the dissolved
matter in seawater ("sea salt") can be treated as a material of uniform composition,
whose absolute salinity (i.e. the grams of solute per kilogram of seawater) changes only
due to the addition and removal of freshwater through rain, evaporation, and river
inflow. This is because the processes that govern the addition and removal of the
constituents of sea salt have extremely long time scales, relative to those that affect the
pure water component of seawater, so that we can treat the total ocean salt content as
approximately constant, while subject to spatially and temporally varying boundary
fluxes of fresh water which give rise to salinity gradients.
The utility of this definition of uniform composition of sea salt lies in its
conceptual simplicity, well suited to theoretical and numerical ocean modelling at time
scales of up to 100s of years. However, to the demanding degree required for observing
and understanding deep ocean pressure gradients, sea salt is neither uniform in
composition, nor is it a conserved variable, nor can its absolute amount be measured
precisely in any reasonable manner. The repeatable precision of various technologies
used to estimate salinity can be as small as 0.001 g/kg, but the non-ideal nature of
seawater means that these estimates can be different by as much as 0.025 g/kg relative to
the true Absolute Salinity in the open ocean, and as much 0.1 g/kg in coastal areas
(Pawlowicz, 2015).
The most important interior source and sink factors governing changes in the
composition of sea salt are biogeochemical processes that govern the biological uptake of
dissolved nutrients, calcium, and carbon in the upper ocean, and the remineralization of
these substances from sinking particles at depth. At present it is thought that changes



resulting from hydrothermal vent activity, fractionation from sea ice formation, and
through multi-component molecular diffusion processes are of local importance only,
but little work has been done to quantify this.

In order to address this problem, TEOS-10 defines a Reference Composition of

seawater, and a number of slightly different salinity variables that are necessary for
different purposes to account for the non-ideal nature of sea salt.  The TEOS-10 Absolute
Salinity, $S_A$, is the absolute salinity of Reference Composition Seawater of a measured
density (note that capitalization of variable names denotes a precise definition in TEOS-
10).  It is the only salinity variable that can be properly used in calculations of density
using the TEOS-10 Gibbs function.  Solution Absolute Salinity, $S_A^{soln}$, is the (difficult to
measure) absolute salinity of a seawater sample whose composition differs from
Reference Composition; its use would be limited to budgets of dissolved matter.
Preformed Salinity, $S_*$, is the salinity of a seawater parcel with the effects of
biogeochemical processes removed, somewhat analogous to a chlorinity-based salinity
estimate.  It is thus a conservative tracer of seawater, suitable for modelling purposes,
but neglects the spatially-variable small portion of sea salt involved in biogeochemical
processes that is required for the most accurate density estimates.

Finally, ocean observations require a completely different variable, the Practical

Salinity.  This variable, which predates TEOS-10, is essentially based on a measure of the
electrical conductance of seawater, normalized to conditions of fixed temperature and
pressure by empirical correlation equations, and scaled so that ocean salinity
measurements that have been made through a variety of technologies over the past 120
years are numerically comparable.  Practical Salinity measurement technologies involve
a certified reference material called IAPSO Standard Seawater, which for our purposes
can be considered the best available artifact representing seawater of Reference
Composition.

Practical Salinity was not designed for numerical modelling purposes and does not

accurately represent the mass fraction of dissolved matter.  We can link Practical
Salinity, $S_P$, to the Absolute Salinity of Reference Composition seawater (so-called
Reference Salinity, $S_R$) using a fixed scale factor, $u_{PS}$, so that





$$S_R = u_{PS}\, S_P \qquad \text{where} \qquad u_{PS} \equiv (35.165\,04/35)\ \text{g}\,\text{kg}^{-1}. \tag{3}$$

Conversions to and between the other "salinity" definitions, however, involve
knowledge about spatial and temporal variations in seawater composition. Fortunately,
the largest component of these changes occurs in a set of constituents involved in
biogeochemical processes, whose co-variation is known to be strongly correlated. Thus
the Absolute Salinity of real seawater can be determined globally to useful accuracy
from the Reference Salinity by the addition of a single parameter, the so-called Absolute
Salinity Anomaly, $\delta S_A$,
$$S_A = S_R + \delta S_A, \tag{4}$$

which has been tabulated in a global atlas for the current ocean (McDougall et al., 2012),
and is estimated in coastal areas by considering the effects of river salts (Pawlowicz,
2015). It can also be determined from measurements of either density or of carbon and
nutrients (IOC et al., 2010). For purposes of numerical ocean modelling, it could in
theory be obtained by separately tracking the carbon cycle and nutrients, and applying
known correction factors, but we are not aware of any attempts to do so.
Finally, chemical modelling (Pawlowicz (2010), Wright et al. (2011), Pawlowicz et
al. (2012)) suggests the approximate relation
$$S_A - S_* \approx 1.35\,\delta S_A \equiv 1.35\left(S_A - S_R\right), \tag{5}$$

and these relationships are schematically illustrated in Fig. 4. The magnitude of the
Absolute Salinity Anomaly is around -.005 to +0.025 g/kg in the open ocean, relative to a
mean Absolute Salinity of about 35 g/kg. The correction it implies may be important
when initializing models, or comparing them with observations, but its major effect is
likely in producing biases in calculated isobaric density gradients.

***Seawater density***
The density of seawater is the most important thermodynamic property affecting
oceanic motions, since its spatial changes (along with changes to the sea-surface height)
give rise to pressure gradients which are the primary driving force for currents within
the ocean interior through the hydrostatic relation. The "traditional" equation of state is
known as EOS-80 (UNESCO, 1981), and is standardized as a function of Practical

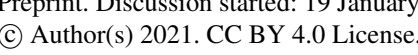
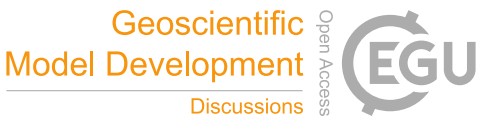

Salinity and in-situ temperature, $\rho = \rho(S_P, t, p)$, which has 41 numerical terms. An
additional equation (the adiabatic lapse rate) is required for conversion of temperature
to potential temperature. However, for ocean models, the equation of state is usually
taken to be the 41-term expression written in terms of potential temperature,
$\rho = \tilde{\rho}(S_P, \theta, p)$, of Jackett and McDougall (1995), where the over-twiddle indicates that
this rational function fit was made with Practical Salinity $S_P$ and potential temperature
$\theta$ as the input salinity and temperature variables.
The current standard for describing the thermodynamic properties of seawater,
known as TEOS-10, provides an equation of state, $v = 1/\rho = v(S_A, t, p)$, in the form of a
function which involves 75 coefficients (IOC et al., 2010) and is an analytical pressure
derivative of the TEOS-10 Gibbs function. However, for ocean models using TEOS-10
the equation of state used is one of those in Roquet et al. (2015); the 55-term equation of
state, $\rho = \hat{\rho}(S_A, \Theta, z)$, used by Boussinesq models and the 75-term form in terms of
specific volume, $v = \hat{v}(S_A, \Theta, p)$, used by non-Boussinesq ocean models.
In this paper we will not concentrate on the distinction between Boussinesq and
non-Boussinesq ocean models, and henceforth we will take the third input to the
equation of state to be pressure, even though for a Boussinesq model it is in fact a scaled
version of depth. By the same token, we will cast the discussion in terms of the in situ
density, even though the non-Boussinesq models have as their equation of state a
polynomial for the specific volume, $v = 1/\rho$.
For seawater of Reference Composition, both the TEOS-10 and EOS-80 fits
$\rho = \hat{\rho}(S_A, \Theta, p)$ and $\rho = \tilde{\rho}(S_P, \theta, p)$ are almost equally accurate (see section A.5 of IOC et
al. (2010), and in particular, note the comparison between Figures A.5.1 and A.5.2
therein). That is, if we set $\delta S_A = 0$ and use Eqn. (3) to relate Practical and Reference
Salinities (which in this case are the same as Preformed Salinities), the numerical density
values of in situ density calculated using EOS-80 are not significantly different to those
using TEOS-10.
This being the case, we can see from section A.5 and A.20 of the TEOS-10 Manual
(IOC et al. (2010)) that 58% of the data deeper than 1000 dbar in the World Ocean would
have the thermal wind misestimated by ~2.7% due to ignoring the difference between





Absolute and Reference Salinities. No ocean model has addressed this deficiency to
date, but McCarthy et al. (2015) studied the influence of using Absolute Salinity versus
Reference Salinity in calculating the overturning circulation in the North Atlantic. They
found that the overturning streamfunction changed by 0.7Sv at a depth of 2700m,
relative to a mean value at this depth of about 7 Sv, i.e. a 10% effect. Because we argue
that the salinity variable in ocean models is best interpreted as being Preformed Salinity,
$S_*$, the neglect of the distinction between Preformed and Absolute Salinities in ocean
models means that they mis-estimate the overturning streamfunction by 1.35 (see Figure
4) times 0.7Sv, namely ~1Sv, i.e. a 13.5% effect.

***Air-sea heat fluxes***

Sensible, latent and long-wave radiative fluxes are affected by near-surface

turbulence and are usually calculated using bulk formulae involving air and sea
surface water temperatures (the air and sea in situ temperatures), as well as other
parameters (e.g., the latent heat involves the isobaric evaporation enthalpy, commonly
called the latent heat of evaporation, which is actually a weak function of temperature
and salinity; see Eqn. 6.28 of Feistel et al. (2010) and Eqn. (3.39.7) of IOC et al. (2010)).
The total air-sea heat flux, $Q$, is then translated into a water temperature change by
dividing by a heat capacity $c_p^0$, which has always been taken to be constant in
numerical models (Griffies et al., 2016). Although this is appropriate for CT
(assuming that the TEOS-10 value is used for $c_p^0$), it is not appropriate when potential
temperature is being considered. The flux of potential temperature into the surface of
the ocean should be $Q$ divided by $c_p(S_*, \theta, 0)$. The use of a constant specific heat
capacity, in association with the interpretation of the ocean's temperature variable as
being potential temperature, means that the ocean has received a different amount of
heat than the atmosphere actually delivers to the ocean, and this issue will be
explored in section 3.

When precipitation ($P$) occurs at the sea surface, this addition of freshwater

brings with it the associated potential enthalpy $h(S_A = 0, t, 0\,\text{dbar})$ per unit mass of
freshwater, where $t$ is the in situ temperature of the rain drops as they arrive at the sea



surface. The temperature at which rain enters the ocean is not yet treated consistently in
coupled models, and section K1.6 of Griffies et al. (2016) suggests that this effect could
be equivalent to an area-averaged extra air-sea heat flux of between $-150\,\mathrm{mW\,m^{-2}}$
and $-300\,\mathrm{mW\,m^{-2}}$, representing a heat loss for the ocean.

***Numerical ocean models***
In deciding how to numerically model the ocean, an explicit choice must be made
about the equation of state, and one would think that this choice would have
implications about the precise meaning of the temperature and salinity variables in the
model, which we will call $T_{\mathrm{model}}$ and $S_{\mathrm{model}}$ respectively. We can divide ocean models
into two general classes, EOS-80 models and TEOS-10 models:

EOS-80 models
One class of CMIP models is based around EOS-80, and these models have the
following characteristics:
1.  The model's equation of state, $\rho = \tilde{\rho}\left(S_{\mathrm{P}}, \theta, p\right)$, expects to have Practical Salinity

and potential temperature as the salinity and temperature input parameters.

2.  $T_{\mathrm{model}}$ is advected and diffused in the ocean interior in a conservative manner.
3.  $S_{\mathrm{model}}$ is advected and diffused in the ocean interior in a conservative manner.
4.  The air-sea heat flux is delivered to/from the ocean using a constant isobaric

specific heat, $c_p^0$, to convert the air-sea heat flux into a surface flux of $T_{\mathrm{model}}$. [A

EOS-80 based model's value of $c_p^0$ is generally only slightly different than the

TEOS-10 value.]

5.  $T_{\mathrm{model}}$ is initialized from an atlas of values of potential temperature, and $S_{\mathrm{model}}$ is

initialized with values of Practical Salinity.

Naively, it then seems reasonable to assume that $T_{\mathrm{model}}$ is potential temperature, and
$S_{\mathrm{model}}$ Practical Salinity; but this assumption implies that theoretical errors arising from
items 2 and 3 and 4 are ignored (since neither potential temperature nor Practical
Salinity are conservative variables). In this paper we show that these interpretations of





the model's temperature and salinity variables are not defensible, and we propose
alternative interpretations.

TEOS-10 models
Other models have begun to implement TEOS-10 features. These models generally have
the following characteristics.

1. The model's equation of state, $\rho = \hat{\rho}\left(S_{\mathrm{A}}, \Theta, p\right)$, expects to have Absolute Salinity

and Conservative Temperature as its salinity and temperature input parameters.

2. $T_{\mathrm{model}}$ is advected and diffused in the ocean interior in a conservative manner.

3. $S_{\mathrm{model}}$ is advected and diffused in the ocean interior in a conservative manner.

4. At each time step of the model, the value of potential temperature at the sea

surface (i.e. SST) is calculated from the $T_{\mathrm{model}}$ (which is assumed to be

Conservative Temperature) and this value of SST is used to interact with the

atmosphere via bulk flux formulae.

5. The air-sea heat flux is delivered to/from the ocean using the TEOS-10 constant

isobaric specific heat, $c_p^0$, to convert the air-sea heat flux into a surface flux of

$T_{\mathrm{model}}$.

6. $T_{\mathrm{model}}$ is initialized from an atlas of values of Conservative Temperature, and

$S_{\mathrm{model}}$ is initialized with values of one of Absolute Salinity, Reference Salinity or

Preformed Salinity.

Implicitly, it has then been assumed that $T_{\mathrm{model}}$ is a Conservative Temperature, and $S_{\mathrm{model}}$
is Absolute Salinity.

There is one CMIP6 ocean model that we are aware of, ACCESS-CM2 (Bi et al.

2013), whose equation of state is written in terms of Conservative Temperature, but the
salinity argument in the equation of state is Practical Salinity. The salinity in this model
is initialized with atlas values of Practical Salinity.

From the above, it is clear that there are small but significant theoretical

incompatibilities between different models, and between models and the actual ocean.
These issues become apparent when dealing with the technicalities of intercomparisons,





and various choices must be made. We now consider the implications of these different
choices.

**3. The Interpretation of salinity in ocean models**
Note that in both the EOS-80 and TEOS-10 cases, we still have the question of
how the salinity variable that is carried in the ocean models of CMIP6 should be
interpreted. Consider first the case of the TEOS-10 based models. During the long pre-
industrial spin-up period of order a thousand years, the salinity variable in the model is
treated as being totally conservative, as is Preformed Salinity, $S_*$. Whether the model
was initialized with values of Absolute Salinity, Reference Salinity or Preformed
Salinity, these initial salinity values are nearly identical in the upper ocean, and any
differences between the three initial conditions in the deeper ocean would be diffused
away within the spin-up period. In the absence of the non-conservative biogeochemical
source terms that are needed to force Absolute Salinity away from being conservative
and towards the observed values of Absolute Salinity (or the smaller source terms that
are needed to maintain Reference Salinity) the model's salinity variable will drift away
from the initial condition and will behave as a conservative salinity variable. Hence we
conclude that the salinity variable that a TEOS-10 based model contains after the long
spin-up phase is Preformed Salinity $S_*$, irrespective of whether the model was
initialized with values of Absolute Salinity, Reference Salinity or Preformed Salinity.
Likewise, the prognostic salinity variable after a long spin-up period of an EOS-
80 based model is most accurately interpreted as being Preformed Salinity divided by
$u_{\mathrm{PS}} \equiv (35.165\,04/35)\,\mathrm{g\,kg^{-1}}$, $S_*/u_{\mathrm{PS}}$.
We clearly need more estimates of the magnitude of the dynamic effects of the
variable seawater composition, but for now we might take a change in 1 Sv in the
meridional transport of deep water masses in each ocean basin (based on the work of
McCarthy et al., 2015) as an indication of the magnitude of the effect of neglecting the
effects of biogeochemistry on salinity. At this stage of model development, since all
models are equally deficient in their thermophysical treatment of salinity, at least this
aspect does not present a problem as far as making comparisons between CMIP models.






### 4. Model Heat Flux Calculations


It is clear from the details described above that both types of numerical ocean models
suffer from some internal contradictions with thermodynamical best practice. For
example, for the EOS-80 based models, if $T_{\text{model}}$ is assumed to be potential temperature,
the use of EOS-80 is correct for density calculations but the use of conservative equations
for $T_{\text{model}}$ ignores the non-conservative production of potential temperature. The use of a
constant heat capacity is also in error if $T_{\text{model}}$ is interpreted as potential temperature.
Conservative equations are, however, appropriate for Conservative Temperature. In
addition, if $S_{\text{model}}$ is assumed to be either Practical Salinity or Absolute Salinity, then the
use of conservative equations ignores the changes in salinity that arise from
biogeochemical processes.
One use for these models is to calculate heat budgets and heat fluxes – both at the
surface and between latitudinal bands, and inherent to CMIP is the idea that these
different models should be intercompared. The question of how this intercomparison
should be done, however, was not clearly addressed in Griffies et al. (2016). Here we
begin the discussion by considering two different options for interpreting $T_{\text{model}}$ in EOS-
80 ocean models.

### 4.1 Option 1: interpreting the EOS-80's model's temperature as being potential


*temperature*
Under this option the model's temperature variable $T_{\text{model}}$ is treated as being potential
temperature $\theta$; this is the prevailing interpretation to date. With this interpretation of
$T_{\text{model}}$ one wonders whether Conservative Temperature $\Theta$ should be calculated from the
model's (assumed) potential temperature before calculating (i) the global Ocean Heat
Content as the volume integral of $\rho c_p^0 \Theta$, and (ii) the advective meridional heat transport
as the area integral of $\rho c_p^0 \Theta v$ at constant latitude, where $v$ is the northward velocity.
This question was not clearly addressed in Griffies et al. (2016), and here we emphasize
one of the main conclusions of the present paper, namely that ocean heat content and
meridional heat transports should be calculated using the model's prognostic





temperature variable. Any subsequent conversion from one temperature variable to
another (such as potential to Conservative) in order to calculate heat content and heat
transport is incorrect and confusing, and should not be attempted.

*4.1.1 Issues with the potential temperature interpretation*
There are several thermodynamic inconsistencies that arise from option 1. First,
the ocean model has assumed in its spin-up phase (for perhaps a millennium) that
potential temperature is conservative, so during the whole spin-up phase and beyond,
the contribution of the known non-conservative interior source terms of potential
temperature have been absent, and hence the model's temperature variable has not
responded to these absent source terms and so this temperature field cannot be potential
temperature. Also, since the temperature field of the model is not potential temperature
(because of these absent source terms) the velocity field of the model will also not be
forced correctly due to errors in the density field which in turn affect the pressure force.
The second inconsistent aspect of option 1 is that the air-sea flux of heat is
ingested into the ocean model, both during the spin-up stage and during the transient
response phase, as though the model's temperature variable is proportional to potential
enthalpy. For example, consider some time during the year at a particular location
where the sea surface is fresh (a river outflow, or melted ice). During this time, any heat
that the atmosphere loses or gains should have affected the potential temperature of the
upper layers of the ocean using a specific heat that is 6% larger than $c_p^0$ (see Figure 1).
So if the ocean model's temperature variable is interpreted as being potential
temperature, a 6% error is made in the heat flux that is exchanged with the atmosphere
during these periods/locations. That is, the changes in the ocean model's (assumed)
potential temperature caused by the air-sea heat flux will be exaggerated where and
when the sea surface salinity is relatively fresh. This 6% flux error is not corrected by
subsequently calculating Conservative Temperature from potential temperature; for
example, these temperatures are the same at low temperature and salinity (see Figure 2),
and yet at low values of salinity, the specific heat is 6% larger than $c_p^0$.





This second inconsistent aspect of option 1 can be restated as follows. The
adoption of potential temperature as the model's temperature variable means that there
is a discontinuity in the heat flux of the coupled air-sea system right at the sea surface;
for every Joule of heat (i.e. potential enthalpy) that the atmosphere gives to the ocean,
under this Option 1 interpretation, up to 6% too much heat arrives in the ocean over
relatively fresh waters. In this way, the adoption of potential temperature as the model
temperature variable ensures that the coupled ocean atmosphere system will not
conserve heat. Rather, there appear to be non-conservative sources and sinks of heat
right at the sea surface where heat is unphysically manufactured or destroyed.
The third inconsistent aspect is a direct consequence of the second; namely that if
one is tempted to post-calculate Conservative Temperature $\Theta$ from the model's
(assumed) values of potential temperature, the calculated ocean heat content as the
volume integral of $\rho c_p^0 \Theta$ would no longer be accurately related to the heat that the
atmosphere exchanged with the ocean. Neither would the area integral between latitude
bands of the air-sea heat flux be exactly equal to the difference between the calculated
meridional heat transports that cross those latitudes. Rather, during the running of the
model the heat that was lost from the atmosphere actually shows up in the ocean as the
volume integral of the model's prognostic temperature variable. We agree with
Appendix D3.3 of Griffies et al. (2016) and strongly recommend that Conservative
Temperature is not calculated a posteriori in order to evaluate heat content and heat
fluxes in these EOS-80 based models.

*4.1.2 Quantifying the air-sea flux imbalance*
Here we quantify the air-sea flux errors involved with assuming that $T_{\text{model}}$ of
EOS-80 models is potential temperature. These EOS-80 based models calculate the air-
sea flux of their model's temperature as the air-sea heat flux, $Q$, divided by $c_p^0$.
However, since the isobaric specific heat capacity of real seawater is $c_p(S_*,\theta,0)$, the flux
of potential temperature into the surface of the ocean should be $Q$ divided by $c_p(S_*,\theta,0)$.
So, if the model's temperature variable is interpreted as being potential temperature, the
EOS-80 model has a flux of potential temperature entering the ocean that is too large by



the difference between these transports, namely by $Q/c_p^0$ minus $Q/c_p(S_*,\theta,0)$. This
means that the ocean has received a different amount of heat than the atmosphere
actually delivers to the ocean, with the difference, $\Delta Q$, being $c_p(S_*,\theta,0)$ times the
difference in the surface fluxes of potential temperature, namely (for the last part of this
equation, see Eqn. (A.12.3a) of IOC et al., 2010)
$$\Delta Q \;=\; Q\left(\frac{c_p(S_*,\theta,0)}{c_p^0}-1\right) \;=\; Q(\tilde{\Theta}_\theta - 1). \tag{6}$$

We plot this quantity from the pre-industrial control run of ACCESS-CM2 in
Figure 5c and show it as a cell area-weighted histogram in Figure 5e (note that while
these plots apply to EOS-80 based ocean models, to generate these plots we have
actually used data from ACCESS-CM2 which is a mostly TEOS-10 compliant model).
The calculation takes into account the penetration of shortwave radiation into the ocean
but is performed using monthly-averages of the thermodynamics quantities. The
temperatures and salinities at which the radiative flux divergences occur are taken into
account in this calculation, but the result is little changed if the sea surface temperatures
and salinities are used with the radiative flux divergence assumed to take place at the
sea surface. Results from similar calculations performed using monthly and daily-
averaged quantities in ACCESS-OM2 (Kiss et al. 2020) ocean-only model simulations
were similar, suggesting that correlations between sub-monthly variations are not
significant (at least in these relatively coarse-resolution, diffuse models).
$\Delta Q$ has an area-weighted mean value of $16\,\mathrm{mW\,m^{-2}}$ and we know that this
represents the net surface flux of potential temperature required to balance the volume
integrated non-conservation of potential temperature in the ocean's interior (Tailleux
(2015)). To put this value in context, $16\,\mathrm{mW\,m^{-2}}$ corresponds to 5% of the observed trend
of $300\,\mathrm{mW\,m^{-2}}$ in the global ocean heat content from 1955-2017 (Zanna et al. 2019). In
addition to this mean value of $\Delta Q$, we see from Figure 5c that there are small regions
such as the equatorial Pacific and the western north Pacific where $\Delta Q$ is as large as the
area-averaged heat flux, $300\,\mathrm{mW\,m^{-2}}$, that the ocean has received since 1955. These local
anomalies of air-sea flux, if they actually existed, would drive local variations in
temperature. However, these $\Delta Q$ values do not represent real heat fluxes. Rather they





represent the error in the air-sea heat flux that we make if we insist that the temperature
variable in an EOS-80 based ocean model is potential temperature, with the ocean
receiving a surface heat flux that is larger by $\Delta Q$ than the atmosphere delivers to the
ocean. Figure 6 shows the zonal integration of $\Delta Q$, in units of W per degree of latitude.

Figure 5e shows that, with $T_{\mathrm{model}}$ being interpreted as potential temperature, 5%

of the surface area of the ocean needs a surface heat flux that is more than $135\,\mathrm{mW\,m^{-2}}$
different to what the atmosphere gives to/from the ocean. This regional variation of $\Delta Q$
of approximately $\pm 100\,\mathrm{mW\,m^{-2}}$ is consistent with the regional variations in air-sea flux of
potential temperature found by Graham and McDougall (2013) that is needed to balance
the depth-integrated non-conservation of potential temperature as a function of latitude
and longitude. Figures 5d,f show that much of this spread is due to the variation of the
isobaric specific heat capacity on salinity, with the remainder due to the variation of this
heat capacity with temperature. We note that if this analysis were performed with a
model that resolved individual rain showers, then these episodes of very fresh water at
the sea surface would be expected to increase the calculated values of $\Delta Q$. Interestingly,
by way of contrast, it is the variation of the isobaric heat capacity with temperature that
dominates (by a factor of four) the contribution of this heat capacity variation to the area
mean of $\Delta Q$ (with the contribution of salinity, $\Delta Q_S$, in Figure. 5d, leading to an area
mean of $4\,\mathrm{mW\,m^{-2}}$), as originally found by Tailleux (2015).

While a heat flux error of $\pm 100\,\mathrm{mW\,m^{-2}}$ is not large, it also not trivially small, and

it seems advisable to respect these fundamental thermodynamic aspects of the coupled
earth system. We will see that this $\pm 100\,\mathrm{mW\,m^{-2}}$ issue is simply avoided by realizing
that the temperature variable in these EOS-80 models is not potential temperature.

In Appendix A we enquire whether the way that EOS-80 models treat their fluid

might be made to be thermodynamically correct for a fluid other than seawater. We find
that it is possible to construct such a thermodynamic definition of a fluid with the aim
that its treatment in EOS-80 models is consistent with the laws of thermodynamics. This
fluid has the same specific volume as seawater for given values of salinity, potential
temperature and pressure, but it has different expressions for both enthalpy and
entropy. This fluid also has a different adiabatic lapse rate and therefore a different




relationship between in situ and potential temperatures. However this exercise in
thermodynamic abstraction does not alter the fact that, as a model of the real ocean, and
with the temperature variable being interpreted as being potential temperature, the
EOS-80 models have $\Delta Q$ more heat arriving in the ocean than leaves the atmosphere.
Since CMIP6 is centrally concerned with how the planet warms, it seems
advisable to adopt a framework where heat fluxes and their consequences are respected.
That is, we regard it as imperative to avoid non-conservative sources of heat at the sea
surface. It is the insistence that the temperature variable in EOS-80 based models is
potential temperature that implies that the ocean receives a heat flux from the
atmosphere that is larger by $\Delta Q$ than what the atmosphere actually exchanges with the
ocean. Since there are some areas of the ocean surface where $\Delta Q$ is as large as the mean
rate of global warming, Option 1 is not supportable. This is what motivates Option 2
where we change the interpretation of the model's temperature variable from being
potential temperature to Conservative Temperature.

**4.2 Option 2: interpreting the EOS-80's model's temperature as being Conservative**
**Temperature**
Under this option the ocean model's temperature variable is taken to be Conservative
Temperature $\Theta$. The air-sea flux of potential enthalpy is then correctly ingested into the
ocean model using the fixed specific heat $c_p^0$, and the mixing processes in the model
correctly conserve Conservative Temperature. Hence the second, fourth and fifth items
listed in section 2 are handled correctly, except for the following caveat. In the coupled
model, the bulk formulae that set the air-sea heat flux at each time step use the
uppermost model temperature as the sea surface temperature as input. So with the
Option 2 interpretation of the model's temperature variable as being Conservative
Temperature, these bulk formulae are not being fed the SST (which at the sea surface is
equal to the potential temperature $\theta$). The difference between these temperatures is
$\Theta - \theta$, which is the negative of what we plot in Figure 2. This is a caveat with this
Option 2 interpretation, namely that the bulk formula that the model uses to determine
the air-sea flux at each time step is actually a little bit different to what was intended



when the parameters of the bulk formulae were chosen. This is a caveat regarding what
was intended by the coupled modeler, rather than what the coupled model actually
experienced. That is, with this Option 2 interpretation, the air-sea heat flux, while being
a little bit different than what might have been intended, does arrive in the ocean
properly; there is no non-conservative production or destruction of heat at the air-sea
boundary as there is in Option 1.

Regarding the remaining two items involving temperature listed in section 2, we

can dismiss the fifth item, since any small difference in the initial values, set at the
beginning of the lengthy spin-up period, between potential temperature and
Conservative Temperature will be irrelevant after the long spin-up integration.

This leaves the first point, namely that the model used the equation of state that

expects potential temperature as its temperature input, $\tilde{\rho}\left(S_*/u_{\mathrm{PS}}, \theta, p\right)$, but under this
Option 2 we are interpreting the model's temperature variable as being Conservative
Temperature. In the remainder of this section we address the magnitude of this effect,
namely, the use of $\tilde{\rho}\left(S_*/u_{\mathrm{PS}}, \Theta, p\right)$ versus the correct density $\tilde{\rho}\left(S_*/u_{\mathrm{PS}}, \theta, p\right)$ which is
almost the same as $\hat{\rho}\left(S_*, \Theta, p\right)$. Note, as discussed in section 3 above, the salinity
argument of the TEOS-10 equation of state is taken to be $S_*$ while that of the EOS-80
equation of state is taken to be $S_*/u_{\mathrm{PS}}$. These salinity variables are simply proportional
to each other, and they have the same influence in both equations of state.

Under this Option 2 we are interpreting the model's temperature variable as

being Conservative Temperature, and so the density value that the model calculates
from its equation of state is deemed to be $\tilde{\rho}\left(S_*/u_{\mathrm{PS}}, \Theta, p\right)$ whereas the density should be
evaluated as $\hat{\rho}\left(S_*, \Theta, p\right)$ where we remind ourselves that the hat over the in situ density
function indicates that this is the TEOS-10 equation of state, written with Conservative
Temperature as its temperature input. To be clear, under EOS-80 and under TEOS-10
the in situ density of seawater of Reference Composition has been expressed by two
different expressions,
$$\rho = \tilde{\rho}\left(S_*/u_{\mathrm{PS}}, \theta, p\right) = \hat{\rho}\left(S_*, \Theta, p\right), \tag{7}$$
both of which are very good fits to the in situ density (hence the equals signs); the
increased accuracy of the TEOS-10 equation for density was mostly due to the





refinement of the salinity variable, and the increase in the accuracy of TEOS-10 versus
EOS-80 for Standard Seawater (Millero et al., 2008) was minor by comparison.

So we need to ask what error will arise from calculating in situ density in the

model as $\tilde{\rho}\left(S_*/u_{\mathrm{PS}}, \Theta, p\right)$ instead of as the correct TEOS-10 version of in situ density,
$\hat{\rho}\left(S_*, \Theta, p\right)$? The effect of this small difference on calculations of the buoyancy frequency
and even the neutral tangent plane is likely small, so we concentrate on the effect of this
difference on the isobaric gradient of in-situ density (the thermal wind).

Given that under this Option 2 the model's temperature variable is being

interpreted as Conservative Temperature, $\Theta$, the model-calculated isobaric gradient of
in-situ density is
$$\tilde{\rho}_{S_*}\nabla_P S_* + \tilde{\rho}_\theta \nabla_P \Theta \,, \tag{8}$$
whereas the correct isobaric gradient of in-situ density is actually
$$\hat{\rho}_{S_*}\nabla_P S_* + \hat{\rho}_\Theta \nabla_P \Theta \,. \tag{9}$$
Notice that here and henceforth we drop the scaling factor $u_{\mathrm{PS}}$ from the gradient
expressions such as Eqn. (5). In any case, this scaling factor cancels from the expression,
but we simply drop it for ease of looking at the equations; we can imagine that the EOS-
80 equation of state is written in terms of $S_*$ (which would simply require that a first
line is added to the code which divides the salinity variable by $u_{\mathrm{PS}}$).

The model's error in evaluating the isobaric gradient of in situ density is then the

difference between the two equations above, namely
$$\text{error in } \nabla_P \rho = \left(\tilde{\rho}_{S_*} - \hat{\rho}_{S_*}\right)\nabla_P S_* + \left(\tilde{\rho}_\theta - \hat{\rho}_\Theta\right)\nabla_P \Theta \,. \tag{10}$$
The relative error here in the temperature derivative of the equations of state can be
written approximately as
$$\left(\tilde{\rho}_\theta - \hat{\rho}_\Theta\right)/\hat{\rho}_\Theta = \tilde{\alpha}^\theta/\hat{\alpha}^\Theta - 1 \,, \tag{11}$$
which is the difference from unity of the ratio of the thermal expansion coefficient with
respect to potential temperature to that with respect to Conservative Temperature. This
ratio, $\tilde{\alpha}^\theta/\hat{\alpha}^\Theta$, can be shown to be equal to $c_p\left(S_*, \theta, 0\right)/c_p^0$ and we know (from Figure 1)
that this varies by 6% in the ocean. This ratio is plotted in Figure 7(a). In regions of the
ocean that are very fresh, a relative error in the contribution of the isobaric temperature





gradient to the thermal wind will be as large as 6% while in most of the ocean this
relative error will be less than 0.5%.

Now we turn our attention to the relative error in the salinity derivative of the

equation of state, which, from Eqn. (10) can be written approximately as
$$\left( \tilde{\rho}_{S_*} - \hat{\rho}_{S_*} \right)\big/ \hat{\rho}_{S_*} = \tilde{\beta}^{\theta}\big/ \hat{\beta}^{\Theta} - 1, \tag{12}$$
and the ratio, $\tilde{\beta}^{\theta}\big/\hat{\beta}^{\Theta}$, has been plotted (at $p = 0\,\mathrm{dbar}$) in Figure 7(b). This figure shows
that the relative error in the salinity derivative, $\left( \tilde{\rho}_{S_*} - \hat{\rho}_{S_*} \right)\big/\hat{\rho}_{S_*}$, is an increasing
(approximately quadratic) function of temperature, being approximately zero at $0°\mathrm{C}$, 1%
error at $20°\mathrm{C}$ and 2% error at $30°\mathrm{C}$. An alternative derivation of these implications of
Eqn. (10) is given in Appendix B.

We conclude that under Option 2, where the temperature variable of an EOS-80

based model (whose polynomial equation of state expects to have potential temperature
as its input temperature) is interpreted as being Conservative Temperature, there are
persistent errors in the contribution of the isobaric salinity gradient to the isobaric
density gradient that are approximately proportional to temperature squared, with the
error being approximately 1% at a temperature of $20°\mathrm{C}$ (mostly due to the salinity
derivative of in situ density at constant potential temperature being 1% different to the
corresponding salinity derivative at constant Conservative Temperature). Larger
fractional errors in the contribution of the isobaric temperature gradient to the thermal
wind equation do occur (of up to 6%) but these are restricted to the rather small volume
of the ocean that is quite fresh.

In Figure 8 we have evaluated how much the meridional isobaric density

gradient changes in the upper 1000 dbar of the world ocean when the temperature
argument in the expression for density is switched from $\theta$ to $\Theta$. As explained above,
this is almost equivalent to the density difference between calling the EOS-80 and the
TEOS-10 equations of state, using the same numeric inputs for each. We find that 19% of
this data has the isobaric density gradient changed by more than 1% when switching
from $\theta$ to $\Theta$. The median value of the percentage error is 0.22%; that is, 50% of the data
shallower than 1000 dbar has the isobaric density gradient changed by more than 0.22%





when switching from using EOS-80 to TEOS-10, with the same numerical temperature
input, which we are interpreting as being $\Theta$.

### 4.3  Evaluating the options for EOS-80 models

Under option 1 where $T_{\text{model}}$ is interpreted as potential temperature, there is a
non-conservation of heat at the sea surface, with the ocean seeing one heat flux, and the
atmosphere immediately above it seeing another, with the differences being typically
$\pm 100\,\text{mW}\,\text{m}^{-2}$, with a net imbalance of $16\,\text{mW}\,\text{m}^{-2}$.
Under option 2 where $T_{\text{model}}$ is interpreted as Conservative Temperature, this flux
imbalance does not arise, but two other inaccuracies arise.  First, under option 2 the bulk
formulae that determine part of the air-sea flux is based on the surface values of $\Theta$
rather than of $\theta$ (for which the bulk formulae are designed).  Second, the isobaric
density gradient in the upper ocean is typically different by ~1% to the isobaric density
gradient that would be found if the TEOS-10 equation of state had been adopted in these
models.  These two aspects of option 2 are considered less serious than not conserving
heat at the sea surface by $\pm 100\,\text{mW}\,\text{m}^{-2}$.  Neither of the two inaccuracies that arise under
option 2 are fundamental thermodynamic errors.  Rather they are equivalent to the
ocean modeler choosing (i) a slightly different bulk formulae, and (ii) a slightly different
equation of state.  The constants in the bulk formulae are very poorly known so that the
switching from $\theta$ to $\Theta$ in their use will be well within their uncertainty (Cronin et al.,
2019) while the ~1% change to the isobaric density gradient due to using the different
equations of state is at the level of our knowledge of the equation of state of sea water.
We conclude that option 2 where the $T_{\text{model}}$ in EOS-80 models is interpreted as
Conservative Temperature is much preferred as it treats the air-sea heat flux in a manner
consistent with the First Law of Thermodynamics, and the treatment of $T_{\text{model}}$ as being a
conservative variable is consistent with it being Conservative Temperature.






### 5. Comparison with ocean observations

Now that we have argued that $T_\text{model}$ should be interpreted as being Conservative Temperature, how then should the model-based estimates of ocean heat content and ocean heat flux be compared with ocean observations and ocean atlas data? The answer is by evaluating the ocean heat content correctly in the observed data sets using TEOS-10, whereby the observed data is used to calculate Conservative Temperature, and this is used together with $c_p^0$ to evaluate ocean heat content and meridional heat fluxes.

We have made the case that the salinity variable in CMIP ocean models that have been spun up for several centuries is Preformed Salinity $S_*$ for the TEOS-10 compliant models, and is $S_*/u_\text{PS}$ for the EOS-80 compliant models. It is the value of either $S_*$ or $S_*/u_\text{PS}$ calculated from ocean observations to which the model salinities should be compared. Preformed Salinity $S_*$ is different to Reference Salinity $S_\text{R}$ by only the ratio $0.26 = 0.35/1.35$ compared with the difference between Absolute Salinity and Preformed Salinity (see Figure 4), and these differences are generally only significantly different to zero at depths exceeding 500 m. Note that Preformed Salinity can be evaluated from observations of Practical Salinity using the Gibbs SeaWater (GSW) software gsw_Sstar_from_SP.

### 6. Discussion and Recommendations

We have made the case that it is advisable to avoid non-conservative sources of heat at the sea surface. It is the prior interpretation of the temperature variable in EOS-80 based models as being potential temperature that implies that the ocean receives a heat flux that is larger by $\Delta Q$ than the heat that is lost from the atmosphere. Since there are some areas of the ocean surface where $\Delta Q$ is as large as the mean rate of global warming, the issue is not unimportant. This realization has motivated the new interpretation of the prognostic temperature of EOS-80 ocean models as being Conservative Temperature (our option 2).

A consequence of this new interpretation of the prognostic temperature variable of all CMIP ocean models as being Conservative Temperature means that the EOS-80 based models suffer a relative error of ~1% in their isobaric gradient of in situ density in





the warm upper ocean. How worried should we be about this? One perspective on this
question is to simply note (from above) that there are larger relative errors (~2.7%) in the
thermal wind equation in the deep ocean due to the neglect of variations in the relative
composition of seasalt. Another perspective is to ask how well does science even know
the thermal expansion coefficient, for example. From appendices K and O of IOC et al.
(2010) (and section 7 of McDougall and Barker (2011)) we see that the rms value of the
differences between the individual laboratory-based data points of the thermal
expansion coefficient and the thermal expansion coefficient obtained from the fitted
TEOS-10 Gibbs function is $0.73x10^{-6}$ K$^{-1}$ which is approximately 0.5% of a typical value
of the thermal expansion coefficient in the ocean. Without a proper estimation of the
number of degrees of freedom represented by the fitted data points, we might estimate
the relative error of the thermal expansion coefficient obtained from the fitted TEOS-10
Gibbs function as being half of this, namely 0.25%. So a typical relative error in the
isobaric density gradient of ~1% in the upper ocean due to using $\Theta$ rather than $\theta$ as the
temperature input seems undesirable but not serious.

We must also acknowledge that all models have ignored the difference between

Preformed Salinity, Reference Salinity and Absolute Salinity (which is the salinity
variable from which density is accurately calculated). As discussed in IOC et al. (2010),
Wright et al. (2011) and McDougall and Barker (2011), glossing over these issues of the
spatially variable composition of sea salt, which is the same as glossing over the effects
of biogeochemistry on salinity and density, means that all of our ocean and climate
models have errors in their thermal wind (vertical shear of horizontal velocity) that
globally exceed 2.7% for half the ocean volume deeper than 1000 m. In the deep North
Pacific ocean, the misestimation of thermal wind is many times this 2.7% figure. The
recommended way of incorporating the spatially varying composition of seawater into
ocean models appears as section A.20 in the TEOS-10 Manual (IOC et al. (2010), and as
section 9 in the McDougall and Barker (2012), with ocean models needing to carry a
second salinity type variable. While it is true that this procedure has the effect of
relaxing the model towards the non-standard seawater composition of today's ocean, it
is clearly advantageous to make a start with this issue by incorporating the non-





conservative source terms that apply to the present ocean rather than to continue to
ignore the issue altogether. As explained in these references, once the modelling of
ocean biogeochemistry matures, the difference between the various types of salinity can
be calculated in real time in an ocean model without the need of referring to historical
data.

Nevertheless, we acknowledge that no ocean model to date has attempted to

include the influence of biogeochemistry on salinity and density, and so this is why we
recommend that the salinity from both observations and model output be treated as
Preformed Salinity $S_*$.

### 6.1 Contrasts to the recommendations of Griffies et al. (2016)

How does this paper differ from the recommendations in Griffies et al. (2016)?

That paper recommended that the ocean heat content and meridional transport of heat
should be calculated using the model's temperature variable and the model's value of
$c_p^0$, and we strenuously agree. However, in the present paper we argue that the
temperature variable carried by an EOS-80 based ocean model should be interpreted as
being Conservative Temperature, and not be interpreted as being potential temperature.
This idea was raised as a possibility in Griffies et al. (2016), but the issue was left unclear
in that paper. For example, section D2 of Griffies et al. (2016) recommends that TEOS-10
based models archive potential temperature (as well as their model variable,
Conservative Temperature) "in order to allow meaningful comparisons" with the output
of the EOS-80 based models. We now disagree with this suggestion. The thesis of the
present paper is that the temperature variables of both EOS-80 and TEOS-10 based
models are already directly comparable, and they should both be interpreted as being
Conservative Temperature, and they should both be compared with Conservative
Temperature from observations. The fact that the model's temperature variable is
labeled "theta0" in EOS-80 models and "THETA" in TEOS-10 based models, we now see
as very likely to cause confusion, since we are recommending that the temperature
outputs of both types of ocean models should be interpreted as Conservative
Temperature.





The present paper also diverges from Griffies et al. (2016) in the way that the

salinity variables in CMIP ocean models should be interpreted and thus compared to

observations. Griffies et al. (2016) interpret the salinity variable in TEOS-10 based ocean

models as being Reference Salinity $S_{\mathrm{R}}$ whereas we show that these models actually

carry Preformed Salinity $S_*$ but have errors in their calculation of densities. Similarly,

Griffies et al. (2016) interpret the salinity variable in EOS-80 based ocean models as being

Practical Salinity $S_{\mathrm{p}}$ whereas we show that these models actually carry $S_*/u_{\mathrm{PS}}$, that is,

Preformed Salinity divided by the constant, $u_{\mathrm{PS}}$. This distinction between the present

paper and Griffies et al. (2016) is negligible in the upper ocean where Preformed Salinity

is almost identical to Reference Salinity (because the composition of seawater in the

upper ocean is close to Reference Composition), but in the deeper parts of the ocean, the

distinction is not negligible; for example, based on the work of McCarthy et al. (2015) we

have shown that the use of Absolute Salinity versus Preformed Salinity leads to ~1 Sv

difference in the meridional overturning streamfunction in the North Atlantic at a depth

of 2700 m. However, in this deeper part of the ocean, even though the difference

between Absolute Salinity and Preformed Salinity is not negligible, the difference

between Preformed Salinity and Reference Salinity (which the TEOS-10 based ocean

models have to date assumed their salinity variable to be) is smaller in the ratio 0.35/1.35

= 0.26 (see Figure 4). That is, if the salinity output of a TEOS-10 based ocean model was

taken to be Reference Salinity, the error would be only a quarter of the difference

between Absolute Salinity and Preformed Salinity, a difference which limits the

accuracy of the isobaric density gradients in the deeper parts of ocean models (see

Figure 4). A similar remark applies to EOS-80 based ocean models if their salinity

output is regarded as being Practical Salinity instead of being (as we decree) $S_*/u_{\mathrm{PS}}$.


### 6.2 Summary table of ocean heat content imbalances

In Table 1 we summarize the effects of uncertainties in physical or numerical processes
in estimating ocean heat content or its changes. The first two rows are the rate of
warming (expressed in $\mathrm{mWm^{-2}}$ averaged over the sea surface) due to anthropogenic
global warming, and due to geothermal heating. The third row is an estimate of the





surface heat flux equivalent of the depth-integrated rate of dissipation of turbulent
kinetic energy, and the fourth is an estimate of the neglected net flux of potential
enthalpy at the sea surface due to the evaporation and precipitation of water occurring
at different temperatures.

The next (fifth) row is the consequence of considering the scenario where all of

radiant heat is absorbed into the ocean at a pressure of 25 dbar rather than at the sea
surface.  The derivative of specific enthalpy with respect to Conservative Temperature at
25 dbar, $\hat{h}_\Theta$, is $c_p^0$ times the ratio of the absolute in situ temperature at 25 dbar, $(T_0 + t)$,
to the absolute potential temperature, $(T_0 + \theta)$ at this pressure (see Eqn. (A.11.15) of IOC
et al. (2010)).  The ratio of $\hat{h}_\Theta$ to $c_p^0$ at 25 dbar is typically different to unity by $6x10^{-6}$,
and taking a typical rate of radiative heating of $100\ \mathrm{Wm}^{-2}$ over the ocean's surface leads
to $0.6\ \mathrm{mWm}^{-2}$ as the area-averaged rate of mis-estimation of the surface flux of
Conservative Temperature for this assumed pressure of penetrative radiation.  Since this
is so small, the use of $c_p^0$ (rather than $\hat{h}_\Theta$) to convert the divergence of the radiative heat
flux into a flux of Conservative Temperature is well supported, providing the correct
diagnostics are used for the calculation (such diagnostic issues may be responsible for
the heat budget closure issues identified by Irving et al. 2020).

The next six rows of Table 1 list the mean and twice the standard deviation of the

volume integrated non-conservation production of Conservative Temperature, potential
temperature, and specific entropy, all expressed in $\mathrm{mWm}^{-2}$ at the sea surface.  The
following two rows are the results we have found in this paper for the air-sea heat flux
error that is made if the EOS-80's temperature is taken to be potential temperature.

The final three rows show that ocean models, being cast in divergence form with

heat fluxes being passed between one grid box and the next, do not have appreciable
numerical errors in deducing air-sea fluxes from changes in the volume integrated heat
content.

The estimate from Graham and McDougall (2013) of $-10\ \mathrm{mWm}^{-2}$ is for the net

interior production of $\theta$, so this is a net destruction.  A steady state requires this amount
of extra flux of $\theta$ at the sea surface (so it can be consumed in the interior).  Our estimate





of this extra flux of $\theta$ at the sea surface is $16\,\mathrm{mWm^{-2}}$, which is only a little larger than the
estimate of Graham and McDougall (2013).

*6.3 Summary of recommendations*
In summary, this paper has argued for the following simple guidelines for
analyzing the CMIP model runs. We should
1. interpret the prognostic temperature variable of all CMIP models (whether they

are based on the EOS-80 or the TEOS-10 equation of state) as being Conservative

Temperature,

2. compare the model's prognostic temperature with the Conservative

Temperature, $\Theta$, of observational data,

3. calculate the ocean heat content as the volume integral of the product of

(i) in situ density (ii) the model's prognostic temperature, $\Theta$, and (iii) the model's

value of $c_p^0$,

4. interpret the salinity variable of the model output as being Preformed Salinity $S_*$

for TEOS-10 based ocean models, and $S_*/u_{\mathrm{PS}}$ for EOS-80 based ocean models, (so

it is advisable to post-multiply the salinity output of EOS-80 models by $u_{\mathrm{PS}}$ in

order to have the salinity outputs of all types of CMIP models as Preformed

Salinity $S_*$ ) and,

5. compare the model's salinity variable with Preformed Salinity, $S_*$, calculated

from ocean observations.

6. Sea surface temperature should be taken as the model's prognostic temperature

in the case of EOS-80 models (since this is the temperature that was used in the

bulk formulae), and as the calculated and stored values of potential temperature

in the case of TEOS-10 models.

Note that this sixth recommendation for EOS-80 based models exposes an unavoidable
inconsistency in that the surface values of the model's prognostic temperature is best
regarded internally in the ocean model as being Conservative Temperature, but we
cannot avoid the fact that this same temperature was used as the sea surface (in situ)
temperature in the bulk formulae during the running of such ocean models. Issues such



as these would not arise if all ocean models had been converted to the TEOS-10 equation
of state.
How then should the model's salinity and temperature outputs, $S_*$ and $\Theta$, be used to
evaluate dynamical concepts such as streamfunctions etc? The obvious answer that is
most consistent with the running of a model is to use the equation of state that the model
used, together with the model's temperature and salinity outputs on the native grid of
the model. But since we now have the output salinity and temperature of both EOS-80
and TEOS-10 models being the same (namely $S_*$ and $\Theta$), there is an efficiency and
simplicity argument to analyze the output of all these models in the same manner, using
algorithms from the GSW Oceanographic Toolbox (McDougall and Barker, 2011). Doing
these model inter-comparisons often involves interpolating the model outputs to
different depths (or pressures) than those used in the original ocean model, so incurring
some interpolation errors, and while the use of the GSW software means that the in situ
density will be calculated slightly differently than in some of the forward models, these
differences are small, as can be seen by comparing Figures A.5.1 and A.5.2 of the TEOS-
10 Manual, IOC et al. (2010). Hence it seems viable to evaluate density and dynamic
height using the GSW Oceanographic Toolbox, with the input salinity to this GSW code
being the model's Preformed Salinity, and the temperature input being the Conservative
Temperature, which as we have argued, are the model's prognostic variables.

**Author Contribution**
T J McD. devised this new way of interpreting CMIP ocean model variables, P. M. B and
R. M. H. provided figures for the paper, and all authors contributed to the concepts and
the writing of the manuscript.

**Acknowledgements.** We have benefitted from helpful comments from Drs. Sjoerd
Groeskamp, Fabien Roquet, Geoff Stanley, Casimir de Lavergne, John Krasting and Jan-Erik
Tesdal. This paper contributes to the tasks of the Joint SCOR/IAPSO/IAPWS Committee on
the Thermophysical Properties of Seawater. T. J. McD, P. M. B and R. M. H gratefully
acknowledge Australian Research Council support through grant FL150100090.





**Appendix A: A non-seawater thermodynamic interpretation of Option 1**

Ocean models have always assumed a constant isobaric heat capacity, and have traditionally assumed that the model's temperature variable is whatever temperature the equation of state was designed to accept. Here we enquire whether there is a way of justifying Option 1 thermodynamically in the sense that Option 1 would be totally consistent with thermodynamic principles for a fluid that is different to real seawater.

That is, we pursue the idea that these EOS-80 based ocean models are not actually models of seawater, but are models of a slightly different fluid. We require a fluid that is identical to seawater in some respects, such as that it has the same dissolved material (Millero et al., 2008) and the same issues around Absolute Salinity, Preformed Salinity and Practical Salinity, and the same in situ density as real seawater (at given values of Absolute Salinity, potential temperature and pressure). But we require that the expression for the enthalpy of this new fluid is different to that of real seawater.

The difference that we envisage between real seawater and this new fluid is that, at zero pressure, the enthalpy of the new fluid is given exactly by the constant value $c_p^0$ times potential temperature $\theta$. That is, for the new fluid, potential enthalpy $h^0$ is simply $c_p^0\theta$ (as it would be for an ideal gas), and the air-sea interaction for this new fluid would be exactly as it occurs in the EOS-80 based models. Moreover, conservation of potential temperature is justified for this new fluid, and the density and thermal wind would also be correctly evaluated in these EOS-80 based models.

The enthalpy of this new fluid is then given by (since $h_P = v$)

$$\breve{h}\left(S_A,\theta,p\right) = c_p^0\,\theta \; + \; \int_{P_0}^{P}\tilde{v}\left(S_A,\theta,p'\right)dP',$$ (A1)

while the entropy of this new fluid needs to obey the consistency relationship,

$\breve{\eta}_\theta = \breve{h}_\theta\left(p=0\right)\big/\left(T_0+\theta\right)$, which reduces to

$$\breve{\eta}_\theta = \frac{c_p^0}{\left(T_0+\theta\right)},$$ (A2)

where $T_0 = 273.15\,\mathrm{K}$ is the Celsius zero point. This consistency relationship is derived directly from the Fundamental Thermodynamic Relationship (see Table P.1 of IOC et al.,





2010). Integrating Eqn. (A2) with respect to potential temperature at constant salinity
leads to the following expression for entropy that our new fluid must obey,

$$\bar{\eta}\left(S_{\mathrm{A}},\theta\right) \;=\; c_p^0 \ln\!\left(1+\frac{\theta}{T_0}\right) \;+\; a\!\left(\frac{S_{\mathrm{A}}}{S_{\mathrm{SO}}}\right)\ln\!\left(\frac{S_{\mathrm{A}}}{S_{\mathrm{SO}}}\right). \tag{A3}$$

The variation here with salinity is taken from the TEOS-10 Gibbs-function-derived
expression for specific entropy which contains the last term in Eqn. (A3) with the
coefficient $a$ being $a = -9.310292413479596\,\mathrm{J\,kg^{-1}\,K^{-1}}$ (this is the value of the coefficient
derived from the $g_{110}$ coefficient of the Gibbs function (appendix H of IOC $et\ al.$ (2010)),
allowing for our version of the normalization of salinity, $(S_{\mathrm{A}}/S_{\mathrm{SO}})$). This term was
derived by Feistel (2008) to be theoretically correct at very small Absolute Salinities.
With these definitions, Eqns. (A1) and (A3), of enthalpy and entropy of our new
fluid, we have completely defined all the thermophysical properties of the fluid (see
Appendix P of IOC et al., 2010 for a discussion). Many aspects of the fluid are different
to seawater, including the adiabatic lapse rate (and hence the relationship between in
situ and potential temperatures), since the adiabatic lapse rate is given by $\Gamma = \bar{h}_{\theta P}/\bar{\eta}_\theta$
and while the numerator is the same as for seawater (since $\bar{h}_{\theta P} = \tilde{h}_{\theta P} = \tilde{v}_\theta$), the
denominator, $\bar{\eta}_\theta$, which is now given by Eqn. (A2), can be up to 6% different to the
corresponding function, $\tilde{\eta}_\theta$, appropriate to real seawater.
We conclude that this is indeed a conceptual way of forcing the EOS-80 based
models to be consistent with thermodynamic principles. That is, we have shown that
these EOS-80 models are not models of seawater, but they do accurately model a
different fluid whose thermodynamic definition we have given in Eqns. (A1) and (A3).
This new fluid interacts with the atmosphere in the way that EOS-80 models have
assumed to date, the potential temperature of this new fluid is correctly mixed in the
ocean in a conservative fashion, and the equation of state is written in terms of the
model's temperature variable, namely potential temperature.
Hence we have constructed a fluid which is different thermodynamically to
seawater, but it does behave exactly as these EOS-80 models treat their model seawater.
That is, we have constructed a new fluid which, if seawater had these thermodynamic
characteristics, then the EOS-80 ocean models would also have correct thermodynamics,





while being able to interpret the model's temperature variable as being potential temperature.

But this does not change the fact that in order to make these EOS-80 models thermodynamically consistent in this way we have ignored the real variation at the sea surface of the isobaric specific heat capacity; a variation that we know can be as large as 6%.

Hence we do not propose this non-seawater explanation as a useful rationalization of the behaviour of EOS-80 based ocean models. Rather, it seems less dramatic and more climatically relevant to adopt the simpler interpretation of Option 2. Under this option we accept that the model is modelling actual seawater, that the model's temperature variable is in fact Conservative Temperature, and that there are some errors in the equation of state of these EOS-80 models that amount to errors of the order of 1% in the thermal wind relation throughout much of the upper (warm) ocean. That is, so long as we interpret the temperature variable of these EOS-80 based models as Conservative Temperature, they are fine except that they have used an incorrect equation of state; they have used $\tilde{\rho}$ rather than $\hat{\rho}$. Apart from this "error" in the ocean code, Option 2 is a consistent interpretation of the ocean model thermodynamics and dynamics. In ocean models there are always questions of how to parameterize ocean mixing. To this uncertain aspect of ocean physics, under Option 2 we add the less than desirable expression that is used to evaluate density in EOS-80 based ocean models in CMIP





**Appendix B: An alternative derivation of Eqn. (10)**


Eqn. (10) is an expression for the error in the isobaric density gradient when
Conservative Temperature is used as the input temperature variable to the EOS-80
equation of state (which expects its input temperature to be potential temperature). An
alternative accurate expression to Eqn. (9) for the isobaric density gradient is
$$\tilde{\rho}_{S_*} \nabla_P S_* + \tilde{\rho}_\theta \nabla_P \theta \,, \tag{B1}$$

and subtracting this from the incorrect expression, Eqn. (8), gives the following
expression for the model's error in evaluating the isobaric gradient of in situ density,
$$\text{error in } \nabla_P \rho \;=\; \tilde{\rho}_\theta \nabla_P (\Theta - \theta)\,. \tag{B2}$$

An approximate fit to the temperature difference, $\Theta - \theta$, as displayed in Figure 2 is
$$(\Theta - \theta) \approx 0.05\,\Theta\left(1 - \frac{S_A}{S_{SO}}\right) - 1.75x10^{-3}\,\Theta\left(1 - \frac{\Theta}{25°C}\right), \tag{B3}$$

and using this approximate expression in the right-hand side of Eqn. (B2) gives
$$\frac{\text{error in } \nabla_P \rho}{\tilde{\rho}_\theta} \approx \left[0.05\left(1 - \frac{S_*}{S_{SO}}\right) - 1.75x10^{-3}\left(1 - \frac{\Theta}{12.5°C}\right)\right]\nabla_P\Theta \;-\; \frac{0.05}{S_{SO}}\,\Theta\,\nabla_P S_*\,. \tag{B4}$$

The first part of this expression that multiplies $\nabla_P\Theta$ corresponds to the proportional
error in the thermal expansion coefficient displayed in Figure 7(a). The second part of
Eqn. (B4) amounts to an error in the saline derivative of the equation of state, with the
proportional error (corresponding to Eqn. (12)), being $-\,0.05\tilde{\rho}_\theta\Theta\big/\!\left(\hat{\rho}_{S_A}\,S_{SO}\right)$, and this is
close to the error that can be seen in Figure 7(b). This error is approximately a quadratic
function of temperature since the thermal expansion coefficient $\tilde{\rho}_\theta$ is approximately a
linear function of temperature.


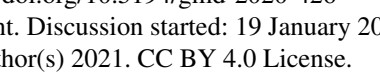




| | Heat flux contributions of different processes | $mWm^{-2}$ |
|---|---|---|
| Physical processes | Global warming imbalance (Zanna et al., 2019), mean | **+300** |
| | Geothermal heating (Emile-Geay and Madec, 2009), mean | **+86** |
| | Viscous dissipation (Graham and McDougall, 2013), mean | **+3** |
| | Atmospheric water fluxes of enthalpy (Griffies et al. 2016), mean | **- (150-300)** |
| Non-conservation errors | Extra flux of $\Theta$ if the air-sea radiative heat flux is taken to occur at a pressure of 25dbar | **-0.6** |
| | non-conservation of $\Theta$ (Graham & McDougall 2013), mean | **+0.3** |
| | non-conservation of $\Theta$ (Graham & McDougall 2013), 2*rms | **+1** |
| | non-conservation of $\theta$ (Graham & McDougall 2013), mean | **-10** |
| | non-conservation of $\theta$ (Graham & McDougall 2013), 2*rms | **$\pm$ 120** |
| | non-conservation of $\eta$ (Graham & McDougall 2013), mean | **+380** |
| | non-conservation of $\eta$ (Graham & McDougall 2013), 2*rms | **+1200** |
| | Interpreting EOS-80 T as $\theta$ (ACCESS-CM2 estimate), mean | **+16** |
| | Interpreting EOS-80 T as $\theta$ (ACCESS-CM2 estimate), 2*rms | **$\pm$ 135** |
| Numerical errors | ACCESS-OM2 single time-step | **$\pm$ 10^(-7)** |
| | ACCESS-OM2 diagnosed from OHC snapshots | **$\pm$ 0.001** |
| | ACCESS-CM2 diagnosed from OHC monthly-averages | **$\pm$ 0.03** |


**Table 1:** Summary of the impact of various processes and modelling errors on the global
ocean heat budget and its imbalance. All numbers are in units of $mWm^{-2}$. Numerical errors
are diagnosed from either ACCESS-OM2 (machine precision errors) or ACCESS-CM2
(associated with not having access to OHC snapshots). Numbers from interior processes are
converted to equivalent surface fluxes by depth integration. The sign convention here is that a
positive heat flux is heat entering the ocean, or warming the ocean by internal dissipation.




**Code Availability**

This paper has not run any ocean or climate models, and so has not produced any such computer code. Processed data and code to produce the ACCESS-CM2 figures 5 & 6 is located at the github repository https://github.com/rmholmes/ACCESS_CM2_SpecificHeat.

**Data Availability**

This paper has not produced any model data. . Processed data and code to produce the ACCESS-CM2 figures 5 & 6 is located at the github repository https://github.com/rmholmes/ACCESS_CM2_SpecificHeat.



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





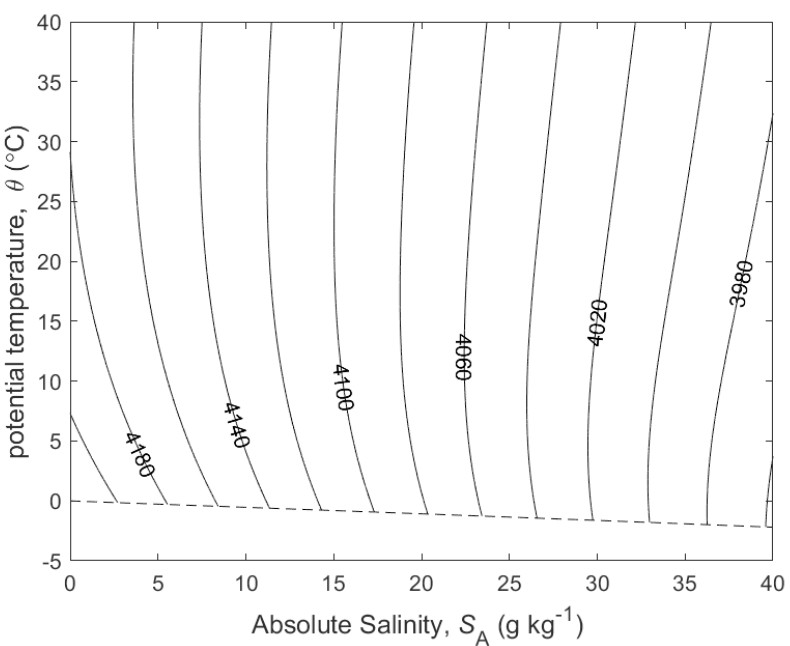


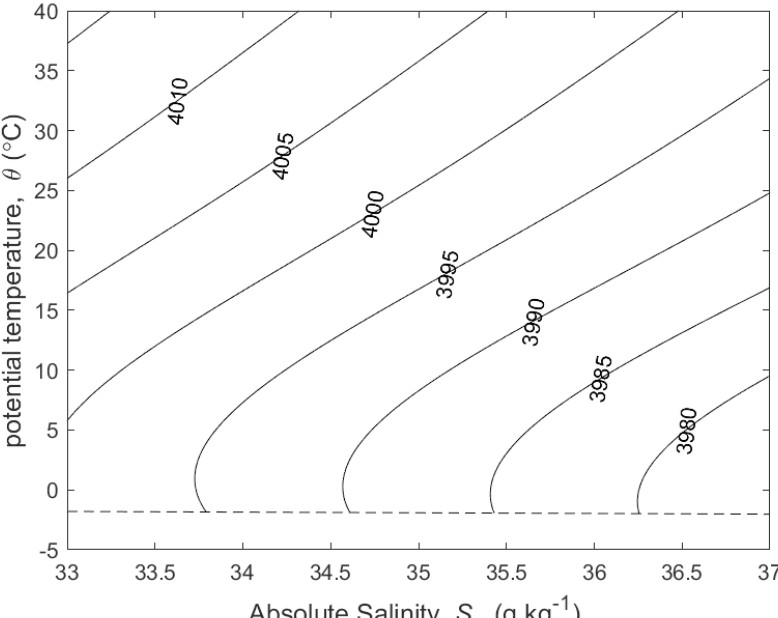

**Figure 1.** (a) Contours of isobaric specific heat capacity $c_p$ of seawater
(in J kg$^{-1}$ K$^{-1}$), at $p = 0$ dbar. (b) a zoomed-in version for a smaller range
of Absolute Salinity. The dashed line is the freezing line at $p = 0$ dbar.



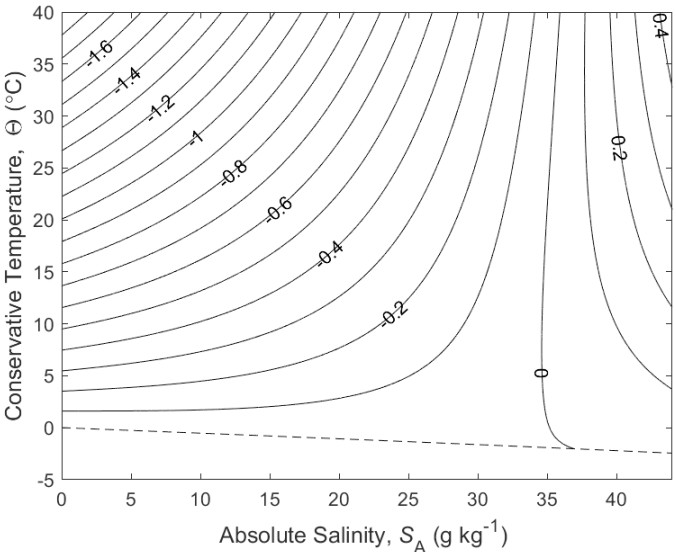

**Figure 2.** Contours (in °C) of the difference between potential temperature and
Conservative Temperature, $\theta - \Theta$.

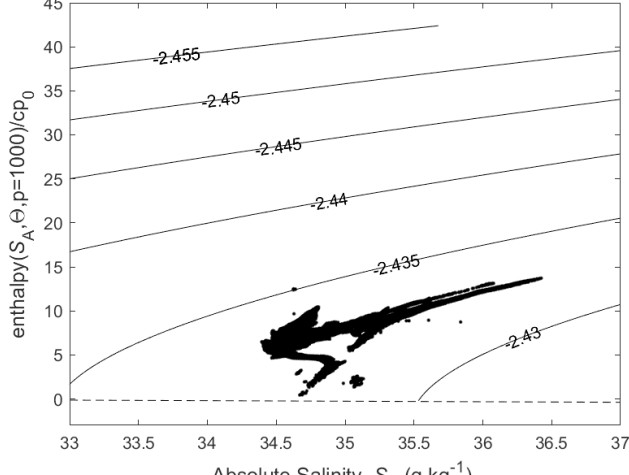


**Figure 3.** Contours of $\Theta - \hat{h}(S_A, \Theta, 1000\,\text{dbar})/c_p^0$ on the Absolute Salinity –
$\hat{h}(S_A, \Theta, 1000\,\text{dbar})/c_p^0$ diagram. Enthalpy, $\hat{h}(S_A, \Theta, 1000\,\text{dbar})$, is a conservative
quantity for turbulent mixing processes that occur at a pressure of $1000\,\text{dbar}$. The
mean value of the contoured quantity is approximately $-2.44°\text{C}$ illustrating that
enthalpy does not posses the "potential" property; that is, enthalpy increases
during adiabatic and isohaline increases in pressure. The fact that the contoured
quantity on this figure is not a linear function of $S_A$ and $\hat{h}(S_A, \Theta, 1000\,\text{dbar})$
illustrates the (small) non-conservative nature of Conservative Temperature. The
dots are data from the word ocean at $1000\,\text{dbar}$.






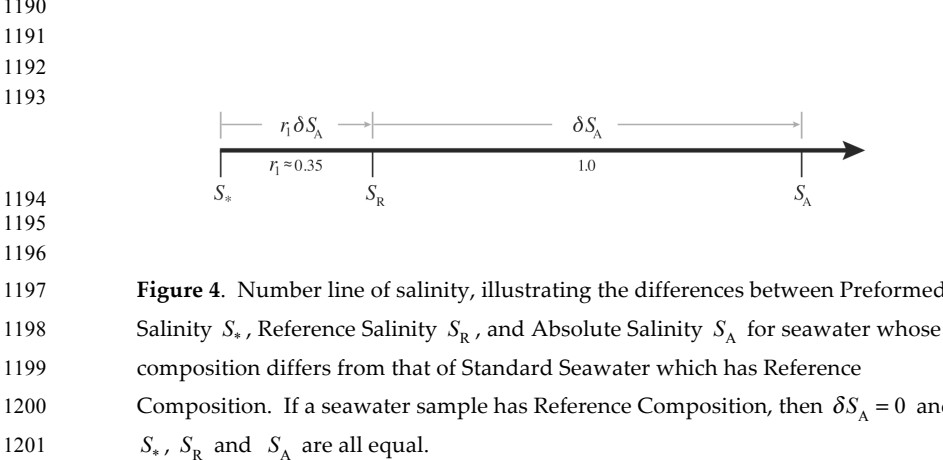


**Figure 4**. Number line of salinity, illustrating the differences between Preformed
Salinity $S_*$, Reference Salinity $S_R$, and Absolute Salinity $S_A$ for seawater whose
composition differs from that of Standard Seawater which has Reference
Composition. If a seawater sample has Reference Composition, then $\delta S_A = 0$ and
$S_*$, $S_R$ and $S_A$ are all equal.






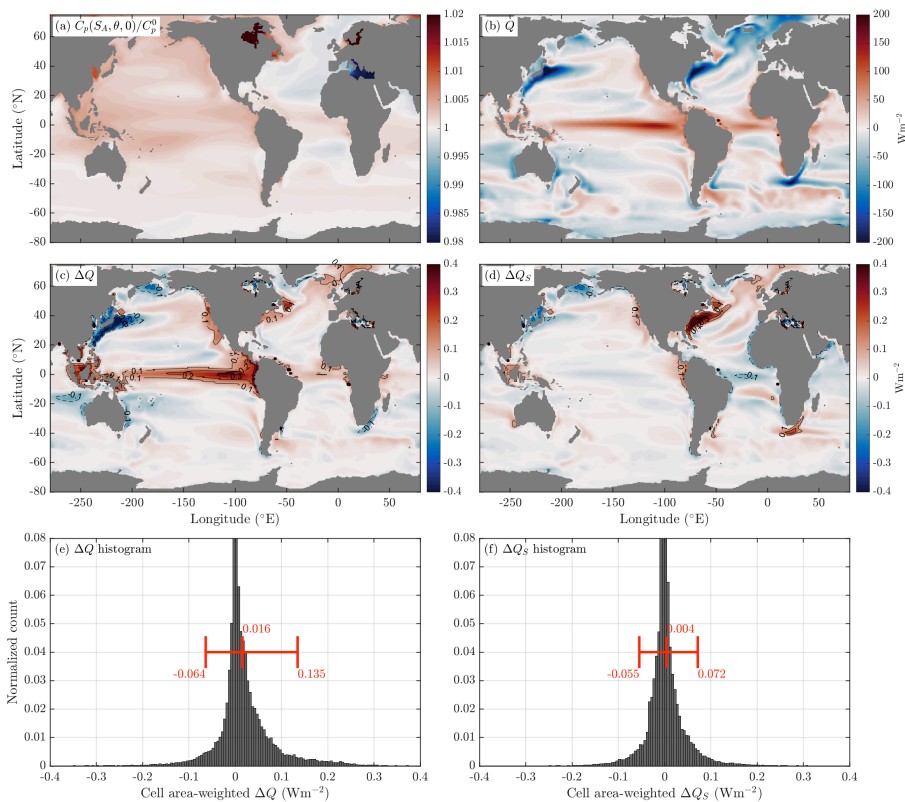


**Figure 5.** (a) The average value of the ratio of the isobaric specific heat of
seawater and $c_p^0$ for data from the ACCESS-CM2 model's pre-industrial
control simulation (600 years long). (b) The average surface heat flux $Q$ (W m⁻
²) in this same ocean model. (c) The additional heat that the ocean
receives/loses compared to the heat that the atmosphere loses/receives
(associated with assuming that an EOS-80 model's temperature variable is
potential temperature), $\Delta Q$ ( $\mathrm{W\,m^{-2}}$ , Eqn. 6). (e) a histogram of $\Delta Q$ weighted
by the area of each grid cell. (d) The contribution of salinity variations to the
air-sea heat flux discrepancy, given by $\Delta Q_S = Q(S-\overline{S})(1/c_p^0)\partial c_p/\partial S$ , where
$\overline{S}$ is the surface mean salinity and $\partial c_p/\partial S$ is the variation in the specific
heat with salinity at the surface mean salinity and potential temperature. (f)
a histogram of $\Delta Q_S$ weighted by the area of each grid cell. Shown in red in
panels e and f are the mean, 5th and 95th percentiles of the histogram ( $\mathrm{W\,m^{-2}}$ ).
Note that these calculations neglect correlations between surface properties
and the surface heat flux at sub-monthly time scales.



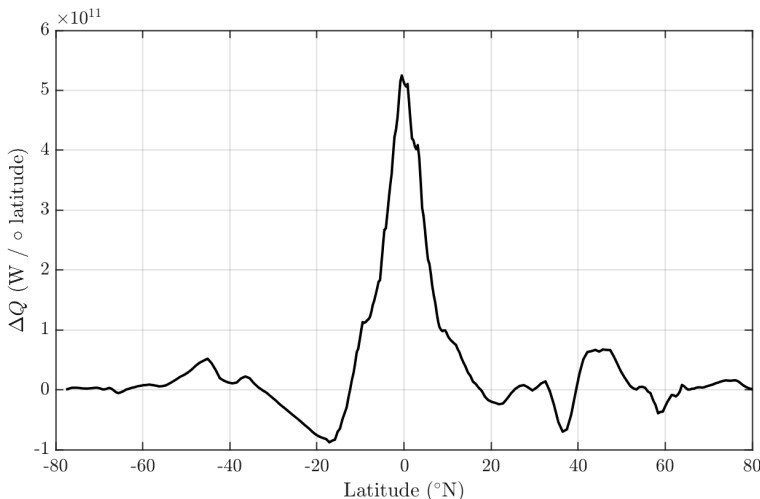

**Figure 6**. The zonally integrated $\Delta Q$ From Fig.5c, showing the imbalance
in the air-sea heat flux in Watts per degree of latitude.



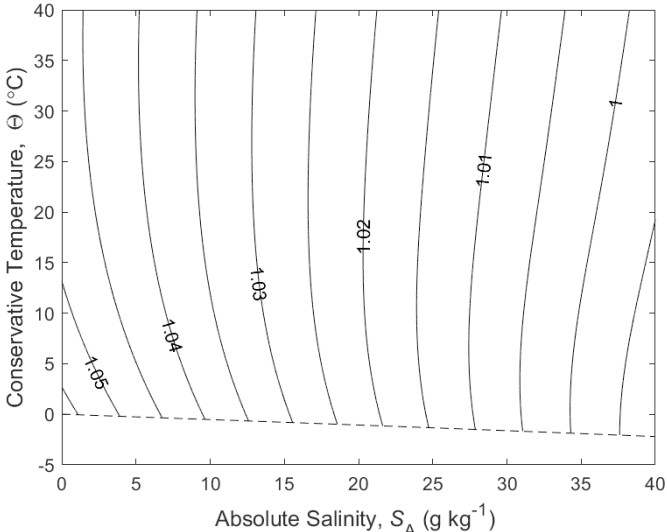


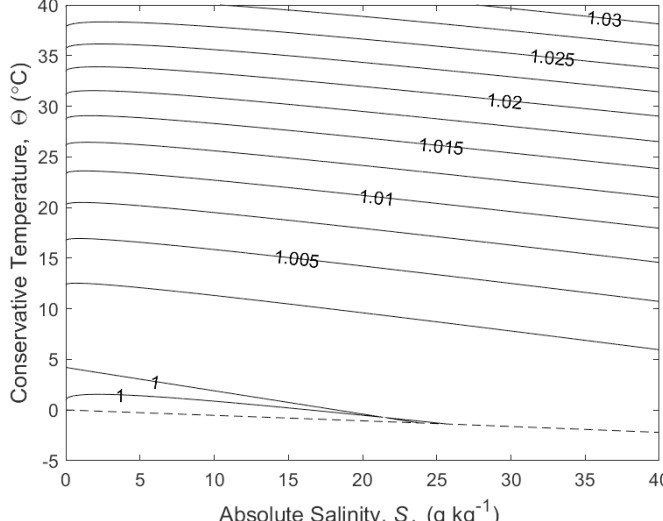


**Figure 7.** (a) The ratio of the thermal expansion coefficients with respect to Conservative
Temperature and potential temperature, $\tilde{\alpha}^{\theta}/\hat{\alpha}^{\Theta} = \tilde{\Theta}_{\theta}$. (b) The ratio of the saline
contraction coefficients at constant potential temperature to that at constant Conservative
Temperature, $\tilde{\beta}^{\theta}/\hat{\beta}^{\Theta} = 1 + \left(\hat{\alpha}^{\Theta}/\hat{\beta}^{\Theta}\right)\hat{\theta}_{S_{A}}/\hat{\theta}_{\Theta}$ at $p = 0$ dbar.






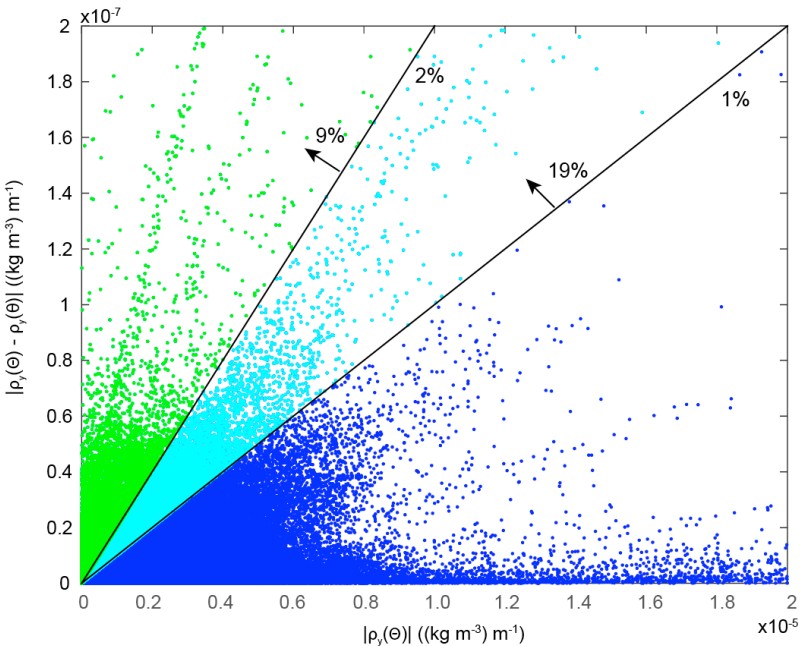


**Figure 8.** The northward density gradient at constant pressure (the horizontal axis) for
data in the global ocean atlas of Gouretski and Koltermann (2004) for $p < 1000$ dbar. The
vertical axis is the magnitude of the difference between evaluating the density gradient
using $\Theta$ versus $\theta$ as the temperature argument in the expression for density. This is
virtually equivalent to the density difference between calling the EOS-80 and the TEOS-10
equations of state, using the same numeric inputs for each. The 1% and 2% lines indicate
where the isobaric density gradient is in error by 1% and 2%. 19% of the data shallower
than 1000 dbar has the isobaric density gradient changed by more than 1% when
switching between the equations of state. The median value of the percentage error in the
isobaric density gradient is 0.22%.

1250
1251