# Peer review of "The interpretation of temperature and salinity variables in numerical ocean model output, and the calculation of heat fluxes and heat content"

_Geoscientific Model Development, 2020_

## Referee Comment (RC1)

Referee report on: The interpretation of temperature and salinity variables in numerical
ocean model output, and the calculation of heat fluxes and heat content
by McDougall, Barker, Holmes, Pawlowicz, Griffies and Durack

**General comments**   The main aim of this paper is to argue that the potential tempera-
ture $\theta$ calculated by EOS-80 based numerical ocean models should in fact be interpreted as
Conservative Temperature, thus allowing it to be directly compared to the CT calculated
by TEOS-10 based models. In this respect, the authors strongly dissent from Griffies et
al. (2016), who recommended that TEOS-10 based models diagnose $\theta$ from CT as the
temperature variable to be compared with that of EOS-80 based models. The authors also
recommend that the salinity variable of models should be interpreted as preformed salinity,
in contrast to Griffies et al. (2016), who recommended to retain the same interpretation
of salinity as before TEOS-10.

As a practicing theoretician with over 30 years experience in analysing and constructing
theoretical models of the ocean and atmosphere, there is absolutely no doubt in my mind
that the new recommendations proposed are incorrect and based on flawed logic, and that
ocean modellers should stick with Griffies et al. (2016)'s recommendations for the time
being. From a theoretical viewpoint, the identity and nature of the temperature and salinity
variables used by a numerical model must be consistent with the choice of equation of state.
As regards to salinity, the fact that all the thermodynamic properties that can be derived
from the TEOS-10 Gibbs function are only well defined for Reference Composition Salinity
$S_R$ means that there is no choice but to interpret salinity in TEOS-10 models as $S_R$. It may
be true that such an interpretation results in errors in the estimation of density gradients,
but the fact is that accounting for variations in composition in the binary fluid idealisation
underlying current theories of the ocean circulation and ocean models is fundamentally a
mathematically ill-posed problem, which implies that more sophisticated interpretations
of salinity are irrelevant for current modelling practices, as correctly recognised by Griffies
et al. (2016). In other words, any ocean model that would attempt to use Absolute
Salinity instead of $S_R$ should be regarded as fundamentally physically suspicious. Although
the authors are correct to point out that using density salinity or absolute salinity may
be useful for improving the accuracy of the thermal wind and density gradients, studies
such as Feistel et al. (2010) suggest that the approach may however introduce errors in
other key quantities such as the speed of sound, which in turn would introduce errors
in the estimation of potential densities, buoyancy frequency, enthalpy and total energy
for instance. Encouraging ocean modellers to test the impact of using Absolute Salinity
without telling them about its potential detrimental effects seems a bit misleading and likely
to generate a lot of pseudo-scientific papers of dubious physical soundness. Is this what
we really want? As regards to EOS-80, the fact that is has been calibrated using practical
salinity $S_p$ similarly implies that salinity in EOS-80 based models must be interpreted as
$S_p$. As to the evolution equations used to forward the prognostic variables forward in time,

they are not supposed to have any bearing on how such variables should be interpreted, because it is always possible to interpret such equations as approximations to the 'true' equations. As a result, the 'normal' and expected behaviour of anybody who discovers that the some of the equations used may turn out to be less accurate than previously thought is to propose a way to correct the inaccurate equations, and to run comparison tests demonstrating the superiority of the new approach over the old one. For over 17 years now, Prof. McDougall have tried to construct a 'perfect' and 'irrefutable' argument based on logic and reasoning, of which the present paper is the latest incarnation, that would convince the ocean modelling community that treating potential temperature as conservative is very bad and harmful. His proposed recommendation, namely that models should switch to CT is interesting and physically sound as far as I can judge; however, I am not at all convinced that it would be as beneficial as Prof. McDougall thinks it is. The problem is that switching to using CT does nothing in itself to confirm or refute that treating potential temperature as conservative in EOS-80 based models is as inaccurate and defective as he claims. Yet, the issue could have long been settled once and for all if Prof. McDougall had put money where his mouth is. Indeed, one important outcome of McDougall (2003), Graham and McDougall (2013), IOC et al. (2010) and my own work is that we now understand how to correct the equation for potential temperature in EOS-80 based models to conform to Prof. McDougall's requirements. All is needed is to replace the boundary condition $Q/c_p$ by replacing the currently constant heat capacity by $c_p(\theta, S, p_a)$, and by adding the missing non-conservation terms by diagnosing these in the manner proposed by Graham and McDougall (2013) or Tailleux (2015).

As far as I can judge, running an EOS-80 based model with the current and corrected equation is the only way to ascertain that the current way of treating potential temperature as conservative is as based as Prof. McDougall claims. It is therefore imperative if we are to settle this issue once and for all that somebody, preferably Prof. McDougall and his team, perform such an experiment, which is the only way I can think of to establish a sound and rigorous physical basis for switching to CT. Failing this, whether or not treating $\theta$ as conservative is as bad as Prof. McDougall claims will remain speculative and purely based on indirect evidence. In any case, I don't think that the authors' ideas and recommendations should be published until their scientific merits has been established by running an EOS-80 based model with the 'correct' potential temperature equation.

In the following, I list the major points of contention and disagreement I have with the authors, as well as specific comments on the text, which in my opinion refute the validity of their recommendations. I hope that my comments will convince the authors that they have overlooked and misunderstood too many key issues, and hence that their best course of action is to withdraw their manuscript and retract their recommendations.

**Major points of contention and disagreement**

1. **'Interpreting $\theta$ as CT' is equivalent to 'interpreting an orange as an apple'** The authors' recommendation that $\theta$ in EOS-80 models should be interpreted as CT presupposes that the two quantities are of the same nature, but we all know that this is not the case. Indeed, while $\theta$ is truly a temperature that can be experimentally measured, CT is truly a non-measurable re-scaled energy quantity disguising as temperature. Moreover, since enthalpy and potential enthalpy are defined up to a linear function of salinity, it follows that the construction of CT involves the specification of three arbitrary parameters, two associated with the said linear function of salinity, one associated with the least-square determination of $c_p^0$. Now, while Prof. McDougall knows about the values of the three arbitrary parameters determining his construction of CT, the potential temperature of an EOS-80 model obviously does not. Can the authors explain how it is somehow possible for $\theta$ to morph into CT without $\theta$ having any knowledge of the particular determination of CT it is supposed to morph into? How is it possible for $\theta$ to somehow morph into CT if it does not know which determination of CT it is supposed to morph into?

2. **$\theta - \Theta$ is a physically-meaningless object that is completely devoid of physical meaning** Because standard physics teaches us that two quantities can only be compared if they are of the same nature, it follows that one should regard the quantity $\theta - \Theta$ as completely devoid of physical meaning, since it is the difference between a temperature and an energy, whose value depends on the specification of three arbitrary parameters. Yet, the authors seem to suggest that the values of $\theta - \Theta$ — a physically meaningless object — should be regarded as somehow representative of the errors arising from treating $\theta$ as conservative in EOS-80 models. How is that possible? Have the authors somehow being taught differently in their physics classes?

3. **One of the premises of the syllogism used by the authors to prove their argument is flawed** As far as I can judge, the authors arrive at their conclusions that $\theta$ should be interpreted as CT in EOS-80 based models by using the following syllogism:

   - *Numerical models assume potential temperature to be conservative.*

   - *We know that potential temperature is not conservative.*

   - *Therefore, potential temperature in EOS80-based ocean models cannot truly be potential temperature and hence should be interpreted as Conservative Temperature.*

   Although I agree that the use of syllogisms represents a valid tool in logic to derive a conclusion deductively, it is also well understood that the validity of doing so crucially depends on the validity of the premises. While the validity of 'We know that potential temperature is not conservative' is indubitable thanks to McDougall

(2003), this is not so of 'Numerical models assume potential temperature to be conservative', which is arguably quite a misleading way to characterise pre-TEOS10 ocean modelling practice. Indeed, a much fairer characterisation closer to the truth is 'EOS80 based models assume that the errors made in using a constant heat capacity to compute surface fluxes of potential temperature and in neglecting interior non-conservation terms are sufficiently small that they are irrelevant in practice'. If this characterisation is used instead of the authors' premise, their syllogism no longer makes sense. As argued above, the authors would have a much stronger case if they could demonstrate the impact of correcting the potential temperature equation in an EOS-80 based model, which would be much more easily understandable by numerical ocean modellers.

4. **CT is only 2 or 3 times more conservative than $\theta$. The conservativeness of CT has been greatly exaggerated so far** I admit to being quite confused about the authors' explanation of the origin for the non-conservative production/destruction of $\theta$ and CT, which conflicts with the exact results of Tailleux (2010). Indeed, according to the latter, the non-conservative terms $\dot{\theta}_{irr} = \dot{\theta}_{irr}^{diff} + \dot{\theta}_{irr}^{visc}$ and $\dot{\Theta}_{irr} = \dot{\Theta}_{irr}^{diff} + \dot{\Theta}_{irr}^{visc}$ are the sum of a diffusive and viscous contribution, with $\dot{\theta}_{irr}^{visc} \approx \dot{\Theta}_{irr}^{visc} \approx \varepsilon_K$, where $\varepsilon_K$ is the local viscous dissipation rate. Yet, the authors's comparison of the relative non-conservativeness of $\theta$ and CT appears to be based solely on the diffusive part. Assuming that viscous dissipation balances about $3\,\mathrm{TW}$ of mechanical energy input by the wind, tides and surface buoyancy fluxes (which is most likely a significant underestimate) yields an equivalent surface flux of $10\,\mathrm{mK.m^{-2}}$. Based on Graham and McDougall (2013)'s estimates, a summary of the diffusive and viscous contribution to the non-conservation of each variable is therefore:

| Non-conservation | diffusive | viscous | sum |
|:---:|:---:|:---:|:---:|
| $\dot{\theta}_{irr}$ | $16 mK.m^{-2}$ | $10 mK.m^{-2}$ | $26 mK.m^{-2}$ |
| $\dot{\Theta}_{irr}$ | $0.3 mK.m^{-2}$ | $10 mK.m^{-2}$ | $10.3 mK.m^{-2}$ |

While these results show that $\Theta$ is about 50 times more conservative under the action of diffusive mixing — admittedly an important result in itself and worth mentioning — its total degree of non-conservativeness accounting for viscous dissipation is only a factor 2-3 better than that of $\theta$ overall. A priori, it is the total degree of non-conservativeness that should be compared, not just that due to diffusion effects, in order to establish whether it is justified to treat CT as exactly conservative in a numerical ocean model. Moreover, as shown by Tailleux (2015), the non-conservation arising from using a constant $c_p$ in the estimation of the surface fluxes for $\theta$ tends to be balanced by the non-conservation due to diffusive effects, at least globally, which means that one should expect to compensate, at least to some extent. As a result, it

is by no means obvious that Prof. McDougall is correct. Again, this could easily be tested by running an EOS-80 based model with a corrected evolution equation for $\theta$.

**Specific comments**

1. Line 63: *namely heat content ("temperature")* Given that standard thermodynamics teaches us that the concepts of 'heat' and 'temperature' should not be confused, starting the paper by confusing the two concepts does not bode well for the following, especially coming from the previous chair of WG127 and current chair of JSC who are supposed to teach us the right way of doing thermodynamics.

2. Lines 64-65. *For computational reasons, it is useful for numerical schemes involved to be conservative [...]* The authors seem to make this a central tenet of their argumentation, even though adding non-conservative terms in a conservative equation does not pose any particular challenge from a numerical viewpoint. Why do the authors consider it would problematic to add the missing nonconservative terms in the potential temperature equation and use the correct boundary condition for the surface fluxes of $\theta$?

3. Lines 70-72. The property of any 'heat' variable to be non-conservative is a generic property of any fluid, which is not limited to seawater, and which can be defined independently of the development of any thermodynamic standard for seawater.

4. Lines 75-77. I agree that it is now widely recognised that potential temperature is not truly conservative. However, there is nothing in thermodynamics that says that the appropriate measure of heat should be conservative or approximately conservative, which it seems important to point out. The idea that 'heat' should be a conservative is idiosyncratic to Prof. McDougall and has absolutely no root in classical thermodynamics or anywhere else in the development of the subject. The only two conservative quantities for seawater idealised as a binary fluid is salinity and total energy. As shown by Tailleux (2010), it is not possible for total energy and any 'heat' variable to be simultaneously conservative. Assuming 'heat' to be conservative is strictly equivalent to assuming that total energy is not conservative, which the authors appear to have overlooked. Given that recent developments seem to focus on the construction of energetically consistent models, e.g., Eden et al. (2014), one should anticipate that ocean modellers will seek to understand how to add the missing non-conservation of 'heat' in their models in order to achieve total energy conservation.

5. Lines 70-80. *This empirical fact is an inherent property of seawater.* I disagree. Nearly all fluids a priori suffer from the same problem, as can easily be demonstrated.

6. Lines 82-83. *[...] results in inherent contradictions.* 'Contradiction' is a loaded word here, because all what the authors have established so far is that treating potential temperature and Conservative Temperature as conservative is only approximate, and that the approximation is a better one for the latter than for the former. Using the term 'contradiction' frames the problem as one that should be solved by logic alone, whereby an illogical approach can only be corrected by a logical one. In contrast, 'an approximation' can be improved by using a more accurate formulation, such as would be achieved by adding the missing non-conservation terms in the potential temperature equation and using the correct flux of potential temperature. In using the term 'contradiction', the authors signal their intention of rooting their arguments in a 'right' versus 'wrong', rather than by a direct demonstration based on comparing two EOS-80 based model using the incorrect and correct evolution equation for $\theta$.

7. Lines 88-89. *even at the cost of introducing problems elsewhere* Why would we want to that when solutions exist to solve problems without adding new ones elsewhere?

8. LInes 94-96. *For example, the insistence that a model's temperature variable is potential temperature involves errors in the air-sea heat flux in some areas that are as large as the mean rate of global warming* This is quite a misleading statement, given that these errors are at least partly compensated by the error arising from neglecting the non-conservation of potential temperature, as suggested by the results of Tailleux (2015).

9. Lines 96-99. Heat is not a conservative property but total energy is. Why do the authors insist on conserving heat but not total energy? Why do they consider it is more logical or rational to conserve heat but not total energy?

10. Lines 105-108. *It is well known that in-situ temperature is not an appropriate measure of the "heat content" [...]* I find it very strange that the authors should discuss what is or what is not an appropriate measure of heat content in the absence of consensus on what should be the 'true' definition of heat content.

11. Lines 119 Section A.17 of IOC et al. (2010). It is interesting to see that this Appendix only discusses the diffusive part of the non-conservation of $\theta$ and $\Theta$, completely overlooking the role of viscous dissipation, and that only the non-rigorous derivations of Graham and McDougall (2013) are cited when Tailleux (2010) gives the exact and explicit forms of non-conservation for the Navier-Stokes equations written in terms of both $\theta$ and $\Theta$.

12. Line 141. Why is it unfortunate?

13. Line 149-150 [...] *has a mean non-conservation error in the global ocean of only about* $0.3\,\mathrm{mW.m^{-2}}$. As shown by Tailleux (2010) (his Equation (25)) and Tailleux

(2015) (his equation (26)), the exact expression for the non-conservative production/destruction of $\Theta$ is

$$\dot{\Theta}_{irr} = \frac{1}{c_p^0}\left[\varepsilon_k - \mathbf{F}_S \cdot \nabla\left(\mu - \frac{T\mu_r}{\theta}\right) - \mathbf{F}_\Theta \cdot \nabla\left(\frac{Tc_p^0}{\theta}\right)\right] \tag{1}$$

and is seen to include the viscous dissipation term $\varepsilon_k$, which the authors subsequently say is of the order of $3mW.m^{-2}$. Presumably, the $0.3mW/m^2$ only refer to the 'diffusive' nonconservation of $\Theta$. The true nonconservation of $\Theta$ is therefore at least $3.3mW.m^{-2}$, which is only 5 times less than the $16mW.m^{-2}$ mean nonconservation of $\theta$. This number increases to $10.3mW.m^{-2}$ if a more realistic estimate of $10mW.m^{-2}$ is used for viscous dissipation, as mentioned in my major comments section.

14. Re-reading Graham and McDougall (2013), I realise that the authors estimated the rate of viscous dissipation $\varepsilon_k$ from the formula $\varepsilon_k = DN^2/\Gamma$. Their estimate corresponds to a surface integrated value of $3.10^{-3} \times 3.10^{14} = 0.9\,\mathrm{TW}$, which is way too small. Clearly, there is at least 3 TW of mechanical energy input due to the wind and surface buoyancy fluxes, not more. A more reasonable estimate of the total viscous dissipation is therefore closer to $10\,\mathrm{mW.m^{-2}}$, which is only a factor of 2 smaller than the nonconservation of $\theta$.

15. Lines 163-164. Not really, given the above arguments. The authors are clearly applying double standards here.

16. Lines 165-167. This is a revisionist view of history, given that Conservative Temperature was introduced by McDougall (2003), long before the IAPWS group was formed, and is not actually part of the UNESCO endorsed part of TEOS-10 (as far as I am aware). This decision was Prof. McDougall's alone, and was not part of the TEOS-10 work. The SCOR/IAPSO WG127 was approved in 2005, and its first meeting took place in Warnmünde in May 2006.

17. Lines 180-183. The question of why potential temperature is non-conservative was actually answered earlier by Tailleux (2010), who showed that the non-conservation of any heat variable is dictated by the first law of thermodynamics (the law of total energy conservation). This is again a revisionist view of history where Graham and McDougall (2013) attempts to get credit for something that needs to be attributed to Tailleux (2010).

18. Lines 199 and below. How conservative is conservative temperature? As said above, this section only describes the nonconservation of Conservative Temperature arising from diffusive mixing and completely omits viscous dissipation. Given that the latter dominates, this section is at best misleading.

19. Line 382. What the authors call naive is what I call common sense. The authors should expect that many oceanographers will feel insulted here.

20. Line 448. I dispute that this is thermodynamic best practice if the authors fail to understand the results of Tailleux (2010) .

21. Lines 479. The authors confuse the terms 'contradiction' and 'approximation' - This is an idiosyncratic interpretation because this is not how idealised modelling should be viewed. Numerical modellers and oceanographers understand that potential temperature is non-conservative; it is therefore unfair to accuse them of assuming potential temperature to be conservative. Rather, they treat it as conservative because they assume that the small nonconservative terms and heat flux errors do not matter on the time scales generally considered. 'Contradiction' and 'approximation' are two completely different concepts, which it is crucial to distinguish in the present discussion, since the authors use the first interpretation in oder to be able to accuse EOS-80 based modelling practice as being illogical. The interpretation 'Models assume the potential temperature to be conservative' has been disproven and there is no reason to accuse ocean modellers of ignoring this result. On the other hand, the interpretation that 'models assume that the nonconservation of potential temperature is sufficiently small that it can be neglected in practice' has not been disproven yet. Indeed, disproving such an approach can only be achieved by running an EOS-80 based model with the incorrect and corrected equation for potential temperature. Such experiments are urgently needed so that we can stop with all the speculation.

22. Lines 735. It is not true that Conservative Temperature is consistent with the first law of thermodynamics, because it assumes that all the heat goes into heat and none into work. Indeed, it is well known from Lorenz's theory of available potential energy that there is about $0.5\,\mathrm{TW}$ of the surface buoyancy fluxes going into the production of available potential energy. This suggests that potential enthalpy includes APE — a work-like quantity — as well as heat, not just heat.

**Literature**

- Eden, C., et al 2014. Toward energetically consistent ocean models. JPO, 3160-3184.

- Feistel, R. et al., 2010. Thermophysical properties anomalies of Baltic seawater. Ocean Sciences, 6, 949–981.

- Tailleux, R., 2010. Identifying and quantifying nonconservative energy production/destruction terms in hydrostatic Boussinesq primitive equation models. Ocean Modelling, 34, 125-136. doi 10.1016/j.ocemod.2010.05.003

- Tailleux, R., 2015. Observational and energetics constraints on the non-conservation of potential/Conservative Temperature and implications for ocean modelling.

---

## Author Comment (AC1)

**Response to Reviewer Remi Tailleux (RT)**

**24th April 2021**

Reviewer RT has provided us with a lengthy and strongly worded disagreement with our manuscript. There are many aspects to this disagreement (he provides some general comments, followed by 4 main points of contention, and 22 specific comments), which we will address below. RT's comments are inserted in black text, and our responses are in blue text. A number of the specific comments, ("specific comments" numbered 4, 9, 10, 11, 17, 20 and 22) encourage us to re-work our analysis by taking Total Energy,  $\mathcal{T} = u + \frac{1}{2}\mathbf{u}\cdot\mathbf{u} + \Phi$  to be a conservative variable. These comments of the present review by RT, as well as the Tailleux (2010) and Tailleux (2015) papers are based on the assumption that Total Energy is a conservative variable, but this assumption is incorrect. Because this issue underlies so many of the reviewer's comments, and is also the basis of the Tailleux (2010) and Tailleux (2015) papers, we address it thoroughly in the Appendix of this Response where we prove that Total Energy is not a conservative variable.

We initiated the present research by asking whether it was possible to ensure that coupled climate models do not lose heat at the air-sea interface, since the usual assumption that an EOS-80 based model carries potential temperature as its temperature variable means that not all of the heat that leaves the atmosphere arrives in the ocean. Fortunately, as discussed at length in our manuscript, this issue is solved by simply interpreting the prognostic temperature variable in EOS-80 based ocean models as being Conservative Temperature. We show that doing so means that the equation of state in the EOS-80 based ocean model is not as accurate as it could be, but there are also many other aspects of ocean models that we know are not perfect; just think about the selection of diffusion coefficients, and the temporal drift of deep ocean temperatures. These are all aspects of our science that we oceanographers continue to work on, but at least let us not continue to lose heat at the air-sea interface, especially when in our paper we describe a very easy fix.

The "general comments" appear to consist mostly of opinions, and so we do not reproduce them here in full. However, four statements by RT in particular are key, since we have different views on each which we feel it is important to highlight:

(1) From a theoretical viewpoint, the identity and nature of the temperature and salinity variables used by a numerical model must be consistent with the choice of equation of state.

Although this statement is reasonable on the face of it, the fact is that our existing equations of state are not exact models of actual seawater, as RT does recognize. Thus, it is not a question of "consistent" versus "inconsistent" – instead all choices have some degree of inconsistency. Given that fact, this paper is then motivated by our insight that the tradeoff incurred by adding a little more inconsistency in the equation of state may be worthwhile, if some other advantage can be gained for a particular purpose – and, in particular, for numerical modelling and CMIP analysis, if we can take advantage of some of the strengths and features of current GFD numerical computational schemes without modifying them (incidentally, we feel that this approach is in fact directly in line with RT's later comments about the superiority of considering "approximations" rather than the true/false dichotomy he apparently attributes to us. We are looking for "better", not necessarily "ideal"). So while

our thesis will involve a little extra error in the equation of state of EOS-80 based ocean models as we have discussed in the article, at least we have provided an easy fix to one of the embarrassing aspects of these EOS-80 based climate models, namely that some of the airsea heat flux disappears.

An ocean model contains many moving parts, including the surface and sea-floor boundary conditions, the equation of state, and the ways in which interior mixing processes (of which there are many) are parameterized. So, it is clear that the equation of state is but one aspect of an ocean model. It so happens that the air-sea flux condition employed in EOS-80 based ocean models is thermodynamically inconsistent with treating the model's temperature variable as being potential temperature, but is thermodynamically consistent with interpreting the model's temperature variable as being Conservative Temperature. Hence our paper has explored the consequences of interpreting an EOS-80 based ocean model's temperature variable as being Conservative Temperature. We find that this is a viable choice, and very importantly, it means that the heat fluxes that are exchanged between the atmosphere and ocean occur without the loss of heat. We are sure that most scientists would agree that it is not advisable to lose heat at the air-sea interface when modeling climate. The point is that the equation of state is but one aspect of an ocean model, and of a coupled climate model, in which the model's temperature variable appears. The combination of all the moving part of the coupled model work together to determine what the ocean model's temperature actually is; not just the equation of state.

**In addition, the general comments also contain the inaccurate contention that:**

(2) "Encouraging ocean modellers to test the impact of using Absolute Salinity without telling them about its potential detrimental effects seems a bit misleading".

A key point about TEOS-10 is that its makers realized that no single "salinity" variable can meet all possible needs, and so rather than attempting to define a "jack of all trades, master of none" type of salinity, several different ones were devised – Preformed, Reference, Absolute, Solution, etc., to provide maximum effectiveness for different purposes. The TEOS-10 Absolute Salinity is in fact designed to provide estimates of specific volume at highest precision, after taking into account spatial changes in the relative composition of sea salt, and is therefore primarily aimed at observationalists, rather than at numerical modellers. Reference Salinity is designed to match best with conductance-based measurements, and hence is also not aimed at numerical modellers. Both of these two forms of salinity are subject to internal sources and sinks from ocean biogeochemical processes.

It was the viewpoint of the SCOR/IAPSO WG-127 that oceanographers were (as always) free to use lower precision salinity variables if that suited their needs, but the consequences of these choices could now be better understood by comparison with the "best" possible variables for a particular purpose (these points and their rationale are explained at length in Wright et al. (2011), and, incidentally, many of the shortcomings of TEOS-10 have been considered, described, and enumerated in the numerous publications that underly this standard). Conservative Temperature is also not touted as an "ideal" parameter, merely a "much better" parameter than potential temperature for certain purposes, e.g., maintaining a conservative-under-mixing behavior in the current ocean.

Note also that the samples whose measured specific volumes were incorporated into both the EOS-80 and TEOS-10 equations of state were of Standard Seawater whose composition is close to Reference Composition. Hence, the EOS-80 and TEOS-10 equations of state were actually constructed with Preformed Salinity as their salinity arguments, not Reference Salinity. That is, for a seawater sample that is not of reference composition, calling the TEOS-10 expression for specific volume with Reference Salinity as the salinity argument will not give an accurate expression for specific volume; the salinity argument should be Absolute Salinity. And for an ocean model that has no non-conservative interior source terms in its salinity evolution equation, and is initialized at the sea surface with Preformed Salinity, then the only interpretation for the model's salinity variable is Preformed Salinity, and the use of the TEOS-10 equation of state will then yield the correct specific volume.

All ocean models treat their salinity as being a conservative variable, and they also all initialize this salinity at the sea surface as Reference Salinity (or, equivalently, as Practical Salinity). At the sea surface the concentration of nutrients is small and so Reference Salinity at the sea surface is virtually the same as Preformed Salinity (and to Absolute Salinity). Since these models are initialized (and restored) to surface values of Preformed Salinity, and since both the models and Preformed Salinity are conservative, then the output salinity of these models has only one interpretation, named Preformed Salinity. The manuscript makes a clear case for this, and this review by RT has not mounted a case against this interpretation. Yes, this is different to what Griffies et al. (2016) recommended. We are not able to change what is published in that paper, but we can push forward with the science and accept the compelling arguments that arise.

**The reviewer also raises an issue concerning the general utility of Conservative Temperature vs. potential temperature, wishing that a different paper, a modelling paper that**

(3) replace[d] the boundary condition  $Q/c_p$  by replacing the currently constant heat capacity by  $c_p(\theta,S,p_a)$ , and by adding the missing non-conservation terms by diagnosing these in the manner proposed by Graham and McDougall (2013) or Tailleux (2015).

had been written by us instead of the present one under review. Now, while the utility of this task that the reviewer describes can be debated, it is our view that undertaking this task is not as straightforward or as interesting as the reviewer suggests, for reasons that we explain in the following paragraphs.

The recommendation of IOC, SCOR and IAPSO that ocean models switch from using potential temperature to using TEOS-10's Conservative Temperature was made after careful consideration of many factors. However, the reviewer is pushing the thesis of Tailleux (2015) (see especially section 5 of that paper) that ocean models can be formulated just as well in terms of potential temperature as they can be in terms of Conservative Temperature. We disagree, and here we summarize why the path suggested by Tailleux (2015) and by the present review of our manuscript, is impractical and unworkable.

The adoption of  $\Theta$  overcomes the following four rather serious disadvantages of adopting the approach advocated by Tailleux (2015) and by the present review, namely

- 1. It is not possible to accurately choose the value of the isobaric heat capacity at the sea surface that is needed when  $\theta$  is the model's temperature variable. The problem arises because of unresolved spatial and temporal variations in the sea surface salinity (SSS) and SST [for example, unresolved rain events that temporarily lower the SSS but are not represented in the time-averaged data]. These unresolved variations in SSS and SST act in conjunction with the nonlinear dependence of the isobaric specific heat on salinity and temperature to mean that it is not possible to obtain the appropriately averaged value of the isobaric specific heat.
- 2. It is not possible to accurately estimate the non-conservative source terms for  $\theta$ . These terms are the product of a turbulent flux and a mean gradient, and in an eddy-resolved ocean model, how would one go about finding the eddy flux of  $\theta$ , which depends on how the averaging is done in space and time. [How to calculate the appropriate mean gradients, over what space and time scales, and how to treat non-divergent eddy fluxes?]
- 3. Calculating the meridional heat flux through an ocean section cannot be done accurately if θ is the model's temperature variable. Because of the interior source terms, in order to calculate the heat flux through an ocean section, one would presumably need to abandon θ and do a conversion to Θ, and then evaluate its transport across the section. In this way, one would have gone full circle, back to treating Θ as the "heat-like" variable whose transport can be compared to the air-sea heat fluxes. So why not adopt Θ as the model's variable to start with? But there is a more basic point here as well. Normally the meridional or zonal heat flux across sections is done with the monthly or annual mean properties, and the conversion from one temperature variable to another cannot be done accurately when the salinity and temperature vary in space and time due to the nonlinear dependence of Θ on salinity and θ (we discuss this effect in Figure 9 below). These issues are avoided when Θ is the model variable.
- 4. Ocean modellers often use the conservation of salinity and the model's temperature variable to check the model's numerics. If  $\theta$  is adopted as the model variable, this is no longer possible because  $\theta$  is not a conservative variable.

In summary, the reviewer discusses, as does Tailleux (2015), that ocean models could well retain potential temperature  $\theta$  as the model's temperature variable, rather than adopt the TEOS-10 recommendation of using Conservative Temperature  $\Theta$ . The above 4 points show that doing so means that (1) the air-sea heat flux cannot be accurately incorporated into the ocean model, (2), the non-conservative source terms that appear in the  $\theta$  evolution equation cannot be estimated accurately, (3) neither can the ocean section-integrated heat fluxes be accurately calculated, and it seems even the inaccurate method that would be employed to do this involves adopting the TEOS-10 approach of calculating the section-wide flux of  $\Theta$ , and (4), an important and convenient conservation check that is routinely employed by ocean modelers would not be available.

Therefore, we see no advantage to adopting the approach suggested by Tailleux (2015) (as repeated in this review); rather there are the above four disadvantages. Hence, as SCOR/IAPSO Working Group 127, IAPSO and SCOR recommended, and as adopted by IOC IAPWS and IUGG, by far the cleanest way to do ocean modelling is to adopt TEOS-10's Conservative Temperature  $\Theta$  as the ocean's temperature variable. Nothing in Tailleux (2015), nor in the arguments of the present review, gives pause to this recommendation of TEOS-10.

**Finally, RT questions how we know that Conservative Temperature (CT) is much more conservative than potential temperature,**

(4) As far as I can judge, running an EOS-80 based model with the current and corrected equation is the only way to ascertain that the current way of treating potential temperature as conservative is as based as Prof. McDougall claims. It is therefore imperative if we are to settle this issue once and for all that somebody, preferably Prof. McDougall and his team, perform such an experiment, which is the only way I can think of to establish a sound and rigorous physical basis for switching to CT. Failing this, whether or not treating  $\theta$  as conservative is as bad as Prof. McDougall claims will remain speculative and purely based on indirect evidence. In any case, I don't think that the authors' ideas and recommendations should be published until their scientific merits has been established by running an EOS-80 based model with the "correct" potential temperature equation.

The use of  $\Theta$  as the model's temperature variable is expected to reduce the diffusive effects of the non-conservative nature of the model's temperature variable by two orders of magnitude, compared with using potential temperature. Of this we can be sure, since there are now at least four studies [McDougall (2003), Graham and McDougall (2013) and Tailleux (2010, 2015)] that show that the non-conservative diffusive source terms for  $\Theta$  and  $\theta$  are in the ratio of 1:100 or so. Also, the influence of the dissipation of kinetic energy  $\varepsilon$  can be added as a source term to the model's  $\Theta$  equation if and when the model's knowledge of  $\varepsilon$  at run time is considered reliable, while this addition of  $\varepsilon$  cannot sensibly be done with the model's temperature variable is  $\theta$ .

But how large are the errors caused by using  $\theta$  as the model variable? In the literature we can read at least two ways of answering this question. First, as discussed in the TEOS-10 Manual, IOC et al. (2010), if an ocean model is forced by the air-sea heat flux boundary condition (as opposed to a restoring condition on temperature), the differences between an ocean model run with  $\Theta$  and  $\theta$  is simply the differences between these variables, that is,  $\theta$  – gsw\_CT\_from\_pt( $S_A, \theta, p$ ), as illustrated in A.13.1, where we see a range of temperature differences exceeding 0.2°C.

The second way of estimating these differences is as done by Graham and McDougall (2013) where they formed vertical integrals of the non-conservative diffusive source terms, finding that those of Conservative Temperature were a factor of 120 less than those of potential temperature. Expressed in terms of an equivalent error in the air-sea heat flux, Graham and McDougall (2013) found that twice the r.m.s. value of the air-sea flux error when ignoring the non-conservative terms of potential temperature was  $\pm 120$  mWm-2. This is not a small error, and is best avoided. The area-mean value of this air-sea flux error is smaller, at around -10 mWm-2, but we oceanographers and climate scientists are concerned not only with the volume integrated heat content, but we also care about the accuracy of regional climate and regional climate projections. Hence  $\pm 120$  mWm-2 is the relevant error measure, not -10 mWm-2.

So we do already know the magnitude of the damage done to ocean models by using potential temperature as the model's prognostic variable. It is a factor of 120 larger than the corresponding non-conservative diffusive error that remains when adopting Conservative Temperature as the ocean model's prognostic variable (see Table 1). Further studies as suggested by the reviewer here could indeed be attempted, but as discussed on pages 3-4 above, there are four reasons why the approach advocated by the reviewer is impractical and unworkable, particularly in an eddy-rich ocean simulation.

We now move on to address RT's four "major points of contention and disagreement" and the 22 "specific comments". In each case we fully include the reviewer's comment in black text and we reply to each comment in blue text.

**Major Points of Contention and Disagreement**

1. "Interpreting  $\theta$  as CT" is equivalent to "interpreting an orange as an apple" The authors' recommendation that  $\theta$  in EOS-80 models should be interpreted as CT presupposes that the two quantities are of the same nature, but we all know that this is not the case. Indeed, while  $\theta$  is truly a temperature that can be experimentally measured, CT is truly a non-measurable re-scaled energy quantity disguising as temperature. Moreover, since enthalpy and potential enthalpy are defined up to a linear function of salinity, it follows that the construction of CT involves the specification of three arbitrary parameters, two associated with the said linear function of salinity, one associated with the least-square determination of  $c_p^0$ . Now, while Prof. McDougall knows

about the values of the three arbitrary parameters determining his construction of CT, the potential temperature of an EOS-80 model obviously does not. Can the authors explain how it is somehow possible for  $\theta$  to morph into CT without  $\theta$  having any knowledge of the particular determination of CT it is supposed to morph into? How is it possible for  $\theta$  to somehow morph into CT if it does not know which determination of CT it is supposed to morph into?

Both the enthalpy and the entropy of a binary fluid (such as we usually suppose seawater to be) are indeed unknown and unknowable up to linear functions of Absolute Salinity. This means that there are four unknown and unknowable constants in the Gibbs function of seawater. While having two unknown constants in the definition of enthalpy sounds as though it might have some undesirable consequences, the fact that these two constants are not only *unknown* but are also *unknowable* means that their values have no consequences. Why? Because, if the values of these two constants had any real-world consequences, then we could measure those consequences and hence determine the values of the constants. But since the constants are unknowable, there can never be any observable consequences. The unimportance of these four arbitrary constants in the seawater Gibbs function is discussed towards the end of Appendix B of the TEOS-10 Manual (IOC et al., 2010).

That is, it is well known from advanced thermodynamic texts that there are not (and cannot be) any consequences of any particular choices that are made for these four constants. TEOS-10 made specific choices for the four constants in the seawater Gibbs function, and the choices were made consistent for the Gibbs functions of ice, freshwater, seawater and humid air.

To repeat, if we were to take the TEOS-10 definition of enthalpy and add to it, for example,  $(1000 + 10S_A/gkg^{-1})Jkg^{-1}$ , we would have a new definition of enthalpy, and no measurement that has been made, or could ever be made in the history of the universe, could ever prefer one definition over the other; they both are correct and indistinguishable from each other in terms of any observable consequence. The same can be said of the two unknown and unknowable constants in the definition of the entropy of seawater; these also have no consequences and cannot ever have any consequences.

Having discussed the four unknown constants in the seawater Gibbs function, we now discuss the fifth arbitrary constant of TEOS-10, namely the value that was chosen for  $c_n^0$ . Again, this value is completely arbitrary. SCOR/IAPSO WG127 could have chosen it to

be, for example,  $1 J kg^{-1} K^{-1}$  in which case Conservative Temperature would have the same numerical value as potential enthalpy. The equation of state could still be defined in terms of this new version of Conservative Temperature, giving exactly the same values of specific volume. At the sea surface, there is still the need to convert from the model's Conservative Temperature to the SST for the calculation of air-sea fluxes from bulk formula, and this conversion could still be done. Hence, the choice of each of the five constants of TEOS-10 have no impact whatsoever on the forward ocean modelling practices of TEOS-10 as recommended by IOC, SCOR and IAPSO in the TEOS-10 Manual, IOC et al. (2010).

The above comments about the un-importance of the five arbitrary TEOS-10 constants apply to when TEOS-10 is adopted (in its entirety). However, in the present paper we are discussing the messy middle ground where the entire ocean community has not yet adopted TEOS-10, and we are suggesting a way to interpret the output of EOS-80 based ocean models using concepts from TEOS-10. In this situation, three of the five arbitrary constants *are* important.

When it comes to re-interpreting an EOS-80 based ocean model as having  $\Theta$  as its prognostic temperature variable, three of the five arbitrary constants are important in making the mean values of  $\Theta$  and  $\theta$  similar at the sea surface. This is by design. The impact of this design is on the air-sea flux that the model draws down from the atmosphere via the flux bulk formulae, as discussed in the text of our manuscript. The three enthalpy-based constants have each been chosen (two by Rainer Feistel, and one  $[c_p^0]$  by Trevor McDougall) to minimize the difference between  $\Theta$  and  $\theta$  at the sea surface. If we had not minimized the difference between these temperatures at the sea surface, the air-sea flux arising from the bulk formulae would be more different than they presently are when  $\Theta$  is used as the SST instead of  $\theta$ .

As described in the text, this difference in the air-sea heat flux when using the two different interpretations of the model's surface temperature is equivalent to the ocean modeler specifying a slightly different set of bulk formulae; this issue is different to the other ones discussed in our paper which go to the thermodynamic consistency of the heat fluxes in the ocean versus those in the atmosphere.

That is, while it is a minor inconvenience to realize that the bulk formulae that the model effectively used is different to the one that is described in the ocean model code, it is much more serious to realize that the ocean has received more heat than the atmosphere thought that it gave the ocean. This is a thermodynamic inconsistency that concerns the core property (air-sea heat fluxes) of coupled modelling. Surely, we should not be complacent about allowing some of this heat flux to just disappear.

Fortunately, as discussed in our manuscript, this issue is solved by interpreting the prognostic temperature variable in EOS-80 based ocean models as being Conservative Temperature. We show that doing so means that the equation of state in the EOS-80 based ocean model is not as accurate as it could be, but there are also many other aspects of ocean models that we know are not perfect; just think about the selection of diffusion coefficients, and the temporal drift of deep ocean temperatures. These are all aspects of our science that we oceanographers continue to work on, but at least let's not continue to lose heat at the airsea interface, especially when we have hit upon such an easy fix.

Finally, potential temperature is no more measurable than is Conservative Temperature. Both are the result of a thought experiment involving an adiabatic and isentropic change of pressure. What is measured is in situ temperature, not potential temperature or Conservative Temperature. 2.  $\theta - \Theta$  is a physically-meaningless object that is completely devoid of physical meaning. Because standard physics teaches us that two quantities can only be compared if they are of the same nature, it follows that one should regard the quantity  $\theta - \Theta$  as completely devoid of physical meaning, since it is the difference between a temperature and an energy, whose value depends on the specification of three arbitrary parameters. Yet, the authors seem to suggest that the values of  $\theta - \Theta$  - a physically meaningless object - should be regarded as somehow representative of the errors arising from treating  $\theta$  as conservative in EOS-80 models. How is that possible? Have the authors somehow being taught differently in their physics classes?

First, both potential temperature and Conservative Temperature have the same units (K).

Second, our answer to the referee's Major Point #1 has addressed the issue of the arbitrary constants.

Third, the differences when running an ocean model with potential temperature versus with Conservative Temperature have been derived two ways to date in the literature, as described in the blue text on page 5 above. The difference,  $\theta - \Theta$ , is representative of the errors made in treating potential temperature as a conservative variable in the case where an ocean model is forced with given surface fluxes [see section A.13 of the TEOS-10 Manual, IOC SCOR and IAPSO (2010)]. In this case there is no error in meridional heat fluxes calculated in either case; the error shows up only in the temperatures themselves [and, as the present manuscript emphasizes, it is all in the *interpretation* of the model's prognostic temperature variable]. With different boundary conditions (such as restoring boundary conditions) the error in assuming that potential temperature values and (b) the potential temperature fluxes.

**3. One of the premises of the syllogism used by the authors to prove their**

**argument is flawed**. As far as I can judge, the authors arrive at their conclusions that  $\theta$  should be interpreted as CT in EOS-80 based models by using the following syllogism:

- Numerical models assume potential temperature to be conservative.
- We know that potential temperature is not conservative.
- Therefore, potential temperature in EOS80-based ocean models cannot truly be potential temperature and hence should be interpreted as Conservative Temperature.

Although I agree that the use of syllogisms represents a valid tool in logic to derive a conclusion deductively, it is also well understood that the validity of doing so crucially depends on the validity of the premises. While the validity of `We know that potential temperature is not conservative' is indubitable thanks to McDougall (2003), this is not so of "Numerical models assume potential temperature to be conservative", which is arguably quite a misleading way to characterize pre-TEOS10 ocean modelling practice. Indeed, a much fairer characterization closer to the truth is 'EOS80 based models assume that the errors made in using a constant heat capacity to compute surface fluxes of potential temperature and in neglecting interior non-conservation terms are sufficiently small that they are irrelevant in practice'. If this characterization is used instead of the authors' premise, their syllogism no longer makes sense. As argued above, the authors would have a much stronger case if they could demonstrate the impact of correcting the potential temperature equation in an EOS-80 based model, which would be much more easily understandable by numerical ocean modellers.

The three bullet points at the beginning of this reviewer's 3rd major point do not offer an accurate summary of the thesis of our paper. Rather, the main reason for suggesting that the temperature variable in EOS-80 based coupled models is best interpreted as being CT is that this is the only way to have conservative exchanges of heat between the ocean and atmosphere. Otherwise, coupled climate models lose heat at the air-sea interface, and this is not a good start to the science of climate simulations. That is, we consider it a very basic feature of a coupled climate model that the heat that the atmosphere delivers to the ocean should actually arrive in the ocean (rather than disappearing). Our manuscript shows that this goal is quite easy to achieve.

Our paper examines the pros and cons of requiring that the same heat flux enter the ocean as leaves the atmosphere. That is, we set ourselves the task of enquiring whether it is possible to fix the leaky air-sea interface that EOS-80 based ocean models have when their model temperature is interpreted as being potential temperature. We do this by interpreting the EOS-80 based ocean model's temperature variable as being CT. Much of our paper is about this point, as concisely summarized in Section 6 of the paper.

Also, we do not make any statement about the intentions or assumptions that ocean modellers have when setting up their model equations. Rather, we make statements about what ocean models do. In particular, all ocean models to date, including the EOS-80 based ocean models, treat their model's temperature variable as a conservative variable, and all have a constant value for the isobaric heat capacity.

4. CT is only 2 or 3 times more conservative than θ. The conservative-ness of CT has been greatly exaggerated so far I admit to being quite confused about the authors' explanation of the origin for the non-conservative production/destruction of θ and CT, which conflicts with the exact results of Tailleux (2010). Indeed, according to the latter, the non-conservative terms  $\dot{\theta}_{irr} = \dot{\theta}_{irr}^{diff} + \dot{\theta}_{irr}^{visc}$  and  $\dot{\Theta}_{irr} = \dot{\Theta}_{irr}^{diff} + \dot{\Theta}_{irr}^{visc}$  are the sum of a diffusive and viscous contribution, with  $\dot{\theta}_{irr}^{visc} \approx \dot{\Theta}_{irr}^{visc} \approx \varepsilon_K$ , where  $\varepsilon_K$  is the local viscous dissipation rate. Yet, the authors's comparison of the relative non-conservativeness of θ and CT appears to be based solely on the diffusive part. Assuming that viscous dissipation balances about 3 TW of mechanical energy input by the wind, tides and surface buoyancy fluxes (which is most likely a significant underestimate) yields an equivalent surface flux of 10 mK.m-2. Based on Graham and McDougall (2013)'s estimates, a summary of the diffusive and viscous contribution to the non-conservation of each variable is therefore:

| Non-conservation     | diffusive      | viscous         | $\operatorname{sum}$ |
|----------------------|----------------|-----------------|----------------------|
| $\dot{	heta}_{irr}$  | $16mK.m^{-2}$  | $10 m K.m^{-2}$ | $26mK.m^{-2}$        |
| $\dot{\Theta}_{irr}$ | $0.3mK.m^{-2}$ | $10mK.m^{-2}$   | $10.3 m K.m^{-2}$    |

While these results show that  $\Theta$  is about 50 times more conservative under the action of diffusive mixing -- admittedly an important result in itself and worth mentioning -- its total degree of non-conservativeness accounting for viscous dissipation is only a factor 2-3 better than that of  $\theta$  overall. A priori, it is the total degree of non-conservativeness that should be compared, not just that due to diffusion effects, in order to establish whether it is justified to treat CT as exactly conservative in a numerical ocean model. Moreover, as shown by Tailleux (2015), the non-conservation arising from using a constant  $c_p$  in the estimation of the surface fluxes for  $\theta$  tends to be balanced by the non-conservation due to diffusive effects, at least globally, which means that one should expect to compensate, at least to some extent. As a result, it is by no means obvious that Prof. McDougall is correct. Again, this could easily be tested by running an EOS-80 based model with a corrected evolution equation for  $\theta$ .

As a community of scientists, we are not only concerned with accurately modelling the *globally integrated* ocean heat content, but we also want to minimize *regional* errors in heat content and other variables. While the mean error in the global heat content of the ocean may be equivalent to only  $10 \text{mWm}^{-2}$  or  $16 \text{ mWm}^{-2}$  (depending on which study one selects), the regional variation of the depth-integrated non-conservative diffusive source term of  $\theta$  is ten times this, at  $\pm 120 \text{mWm}^{-2}$  (see Graham and McDougall (2013) and Table 1 of our manuscript). This range of  $240 \text{mWm}^{-2}$  contains 95% of the values of the depth-integrated diffusive non-conservative source terms in the  $\theta$  budget. This error is much larger than the contribution of the viscous dissipation to the global heat budget; this is true no matter whether 1TW or 3TW is assumed for this number.

Looking forward to a future generation of ocean models, we note significant research effort being directed towards carrying a prognostic equation for kinetic energy and its dissipation  $\varepsilon$ . When this research matures, it will be possible to add  $\varepsilon$  into the temperature evolution equation as the ocean model is run forward in time. Doing so with  $\Theta$  as the model temperature variable makes sense (since the diffusive non-conservative error in  $\Theta$  is no more than 1mK, see Table 1) but makes no sense if  $\theta$  were the model's temperature variable. This is because the errors we have identified of  $\pm 120$  mWm-2 in the missing depth-integrated diffusive source terms of  $\theta$  are much larger than those arising from by ignoring the dissipation of kinetic energy  $\varepsilon$ . So we disagree with the reviewer's reasoning.

**Specific Comments**

1. Line 63: *namely heat content ("temperature")* Given that standard thermodynamics teaches us that the concepts of "heat" and "temperature" should not be confused, starting the paper by confusing the two concepts does not bode well for the following, especially coming from the previous chair of WG127 and current chair of JSC who are supposed to teach us the right way of doing thermodynamics.

We agree that these two are different - this is the reason why "temperature" was in quote marks. Since we then spent 5 manuscript pages explaining this difference in section 2 "Background: thermodynamic measures of heat content" we clearly agree with the reviewer and we certainly are not trying to mislead. In response to this concern, we have reworded to

"...heat content (or its related parameter, temperature)..."

2. Lines 64-65. For computational reasons, it is useful for numerical schemes involved to be conservative [...] The authors seem to make this a central tenet of their argumentation, even though adding non-conservative terms in a conservative equation does not pose any particular challenge from a numerical viewpoint. Why do the authors consider it would problematic to add the missing non-conservative terms in the potential temperature equation and use the correct boundary condition for the surface fluxes of  $\theta$ ?

One answer to this question of the reviewer is that we are dealing with existing models and model runs, and are trying to avoid telling people that their work is useless and must be rerun with a new and untested set of non-conservative equations. The utility of conservative schemes is not a proposition, rather, it is a simple statement of existing practice. Ocean models have, to date, treated both their salinity and their temperature variables as being conservative variables, as manifest by their values being affected by the convergence of fluxes. Doing so enables ocean modellers, and those analyzing the output of ocean models, to check that the numerics are not creating spurious internal sources and sinks, and to check that the particular air-sea flux and geothermal flux that they have been told that the ocean is using, is actually the version that is used in the model. These are very valuable consistency checks that are routinely employed in ocean models used for climate.

We agree that there is no fundamental numerical issue with adding a non-conservative source term to an ocean model code (as we understand it, atmospheric models do it as a matter of course, where their temperature variable is sometimes in situ temperature). However, doing so would mean abandoning this valuable checking procedure for the ocean. The more important issue, as explained above in the blue text on pages 3-4, is that it is not possible to accurately deduce the source terms to add to an ocean model when its temperature variable is taken to be potential temperature; it is neither possible at the sea surface nor in the interior. Nor is it possible to accurately calculate the section-integrated heat flux when the model's temperature variable is potential temperature, because of the complications caused by the for interior source terms.

We contrast this difficulty of potential temperature to the preferred use of Conservative Temperature  $\Theta$  as the model's temperature variable. Using  $\Theta$  allows a modeller to retain the conservative numerical transport schemes, allowing for consistency with the actual evolution equations.

In addition, if or when the ocean modelling community wishes to use the modeled dissipation of kinetic energy  $\varepsilon$  as a source term in the model's temperature evolution equation, this is a sensible thing to attempt when using  $\Theta$ , but it makes no sense to do so

when using  $\theta$  as the model's temperature variable because the missing diffusive nonconservative source terms are an order of magnitude larger than the  $\varepsilon$  source term.

3. Lines 70-72. The property of any "heat" variable to be non-conservative is a generic property of any fluid, which is not limited to seawater, and which can be defined independently of the development of any thermodynamic standard for seawater.

We agree with this point. That is, neither entropy, potential temperature nor Conservative Temperature are 100% conservative. We have stated such in the paper.

4. Lines 75-77. I agree that it is now widely recognized that potential temperature is not truly conservative. However, there is nothing in thermodynamics that says that the appropriate measure of heat should be conservative or approximately conservative, which it seems important to point out. The idea that "heat" should be conservative is idiosyncratic to Prof. McDougall and has absolutely no root in classical thermodynamics or anywhere else in the development of the subject. The only two conservative quantities for seawater idealized as a binary fluid are salinity and total energy. As shown by Tailleux (2010), it is not possible for total energy and any `heat' variable to be simultaneously conservative. Assuming `heat' to be conservative is strictly equivalent to assuming that total energy is not conservative, which the authors appear to have overlooked. Given that recent developments seem to focus on the construction of energetically consistent models, e.g., Eden et al. (2014), one should anticipate that ocean modellers will seek to understand how to add the missing non-conservation of "heat" in their models in order to achieve total energy conservation.

The reviewer is incorrect to say that "The idea that "heat" should be conservative is idiosyncratic to Prof. McDougall ...". Rather, in McDougall's publications he has always identified all the non-conservative source terms of  $\Theta$ , including the contribution of the dissipation of turbulent kinetic energy  $\varepsilon$ , to the non-conservative production of  $\Theta$ . And when ocean models become sufficiently sophisticated that  $\varepsilon$  is available to the model at run time, then the largest source of the non-conservation of  $\Theta$  will be able to be incorporated. When or if  $\varepsilon$  is incorporated into a model's temperature equation, there will, however, be an operational question/trade-off decision to make between;

- (i) making the evolution equation of  $\Theta$  even more accurate by including  $\boldsymbol{\varepsilon}$  , or,
- (ii) retaining the conservative nature of the temperature variable so as to continue to enable overall volume-integrated tracer budget numerical checks, and to ensure basin-wide heat budgets can still be performed without the complications caused by accounting for interior source terms.

Note also, as described above, this incorporation of  $\varepsilon$  into a model's temperature equation will not be able to be done accurately by retaining potential temperature as the model's temperature variable.

A significant part of this comment relates to a proposed alternative, Total Energy. As we explain in the appendix to this reply, Total Energy is in fact not conservative, nor is it a "potential" variable, nor (since it includes kinetic and gravitational potential energy) is it purely thermophysical, making it difficult to work with in a practical sense. 5. Lines 70-80. *This empirical fact is an inherent property of seawater*. I disagree. Nearly all fluids a priori suffer from the same problem, as can easily be demonstrated.

We are puzzled here. We say that potential temperature is produced or destroyed by mixing in seawater, and RT disagrees by saying that this is true for nearly all fluids. We did not make any claims about fluids other than seawater, so it seems we are in agreement.

6. Lines 82-83. [...] results in inherent contradictions. "Contradiction" is a loaded word here, because all that the authors have established so far is that treating potential temperature and Conservative Temperature as conservative is only approximate, and that the approximation is a better one for the latter than for the former. Using the term "contradiction" frames the problem as one that should be solved by logic alone, whereby an illogical approach can only be corrected by a logical one. In contrast, "an approximation" can be improved by using a more accurate formulation, such as would be achieved by adding the missing non-conservation terms in the potential temperature equation and using the correct flux of potential temperature. In using the term "contradiction", the authors signal their intention of rooting their arguments in a "right" versus "wrong", rather than by a direct demonstration based on comparing two EOS-80 based model using the incorrect and correct evolution equation for  $\theta$ .

We agree and have thus removed "inherent" to not mislead the reader into thinking there is a logical fallacy. We kept "contradiction" since, even for  $\Theta$ , there are ignored source terms. As the reviewer correctly notes, our favoring  $\Theta$  is due to its substantially smaller source term and thus its substantially smaller "contradiction".

7. Lines 88-89. *even at the cost of introducing problems elsewhere* Why would we want to [do] that when solutions exist to solve problems without adding new ones elsewhere?

Our paper is aimed at those ocean model groups that have not yet changed to using TEOS-10. We provide them a way to interpret their model output in a way that has the heat flux that the atmosphere thinks that it delivers to the ocean, to actually arrive in the ocean (rather than disappearing). We regard this to be an essential property of a coupled atmosphere-ocean-ice model, particularly when the aim is to simulate climate, past, present and future. Further, we refer the reviewer again to the issues with implementing non-conservative source terms for potential temperature listed in the blue text on pages 3-4 above.

8. Lines 94-96. For example, the insistence that a model's temperature variable is potential temperature involves errors in the air-sea heat flux in some areas that are as large as the mean rate of global warming This is quite a misleading statement, given that these errors are at least partly compensated by the error arising from neglecting the non-conservation of potential temperature, as suggested by the results of Tailleux (2015).

We disagree. What we say here is correct and is supported by our Table 1. The paper of Tailleux (2015) only addresses the *volume-integrated* heat budget, in which there is the well-known (since 2003) significant cancellation between positive and negative source terms of potential temperature. We are concerned with more than simply the volume-integrated heat budget. Graham and McDougall (2013) shed some light on the *regional* variation of the airsea flux errors if potential temperature is used as the model temperature variable. They exhibited these errors in the form of depth-integrated heat imbalances as a function of longitude and latitude.

9. Lines 96-99. Heat is not a conservative property but total energy is. Why do the authors insist on conserving heat but not total energy? Why do they consider it is more logical or rational to conserve heat but not total energy?

See the appendix to this reply. Total Energy is not a conservative variable.

10. Lines 105-108. It is well known that in-situ temperature is not an appropriate measure of the "heat content" [...] I find it very strange that the authors should discuss what is or what is not an appropriate measure of heat content in the absence of consensus on what should be the "true" definition of heat content.

We have changed the wording to "it is well-known that in-situ temperature is not a satisfactory measure of the "heat content" of a water parcel..."

11. Lines 119 Section A.17 of IOC et al. (2010). It is interesting to see that this Appendix only discusses the diffusive part of the non-conservation of  $\theta$  and  $\Theta$ , completely overlooking the role of viscous dissipation, and that only the non-rigorous derivations of Graham and McDougall (2013) are cited when Tailleux (2010) gives the exact and explicit forms of non-conservation for the Navier-Stokes equations written in terms of both  $\theta$  and  $\Theta$ .

Tailleux (2010) was not the first paper to derive the evolution equations for potential temperature, entropy and Conservative Temperature. Rather it was McDougall (2003) that did this first for all three variables. Furthermore, (i) Tailleux (2010) is based on the incorrect assumption that Total Energy is a conservative variable (again, see the Appendix to this Reply to the reviewer), and (ii) McDougall (2003) and Graham and McDougall (2013) derive their equations for turbulent mixing in the ocean, whereas the equations developed by Tailleux (2010) and Tailleux (2015) apply to molecular diffusion. As shown by Graham and McDougall (2013), when one replaces the molecular fluxes with turbulent fluxes, the correct turbulent results of Graham and McDougall (2013) are not recovered. These issues are fully explained in the Appendix to this Reply to the reviewer.

**12. Line 141. Why is it unfortunate?**

It would be fortunate if such a heat-like variable was available since it would mean that the present practice of ocean models of treating their temperature variable as being conservative would be exact rather than approximate. 13. Line 149-150 [...] has a mean non-conservation error in the global ocean of only about 0.3 mW m-2. As shown by Tailleux (2010) (his Equation (25)) and Tailleux (2015) (his equation (26)), the exact expression for the non-conservative production/ destruction of  $\Theta$  is

$$\dot{\Theta}_{irr} = \frac{1}{c_p^0} \left[ \varepsilon_k - \mathbf{F}_S \cdot \nabla \left( \mu - \frac{T\mu_r}{\theta} \right) - \mathbf{F}_\Theta \cdot \nabla \left( \frac{Tc_p^0}{\theta} \right) \right] \tag{1}$$

and is seen to include the viscous dissipation term  $\varepsilon_k$ , which the authors subsequently say is of the order of  $3mW.m^{-2}$ . Presumably, the  $0.3mW/m^2$  only refer to the 'diffusive' nonconservation of  $\Theta$ . The true nonconservation of  $\Theta$  is therefore at least  $3.3mW.m^{-2}$ , which is only 5 times less than the  $16mW.m^{-2}$  mean nonconservation of  $\theta$ . This number increases to  $10.3mW.m^{-2}$  if a more realistic estimate of  $10mW.m^{-2}$ is used for viscous dissipation, as mentioned in my major comments section.

**Please see our response to Specific Comment 14 immediately below.**

14. Re-reading Graham and McDougall (2013), I realise that the authors estimated the rate of viscous dissipation  $\varepsilon_k$  from the formula  $\varepsilon_k = DN^2/\Gamma$ . Their estimate corresponds to a surface integrated value of  $3.10^{-3} \times 3.10^{14} = 0.9$  TW, which is way too small. Clearly, there is at least 3 TW of mechanical energy input due to the wind and surface buoyancy fluxes, not more. A more reasonable estimate of the total viscous dissipation is therefore closer to  $10 \text{ mW.m}^{-2}$ , which is only a factor of 2 smaller than the nonconservation of  $\theta$ .

We are not only concerned with modelling the *globally integrated* ocean heat content correctly, but we also wish to minimize *regional* errors in heat content and other variables. While the error in the global heat content of the ocean associated with interpreting the model's temperature variables as being  $\theta$  may be equivalent to only 10mWm-2 or 16 mWm-2 (depending on which study one selects), the regional variation in the depth-integrated non-conservative diffusive source terms in the  $\theta$  budget is ten times this, at  $\pm 120$ mWm-2 (see Graham and McDougall (2013) and Table 1 of our manuscript). This error is much larger than the contribution of the viscous dissipation to the global heat budget, no matter whether 1TW or 3TW is assumed for this number.

15. Lines 163-164. Not really, given the above arguments. The authors are clearly applying double standards here.

Please see our response to Specific Comment 14 immediately above.

16. Lines 165-167. This is a revisionist view of history, given that Conservative Temperature was introduced by McDougall (2003), long before the IAPWS group was formed, and is not actually part of the UNESCO endorsed part of TEOS-10 (as far as I am aware). This decision was Prof. McDougall's alone, and was not part of the TEOS-10 work. The SCOR/IAPSO WG127 was approved in 2005, and its first meeting took place in Warnumünde in May 2006.

We thank RT for emphasizing the need for historical accuracy.

Conservative Temperature, CT, is part of TEOS-10. When TEOS-10 was adopted by the IOC, the TEOS-10 Manual became the official description of TEOS-10, and in the TEOS-10 Manual there are many sections that derive the evolution equations and recommend the use of Conservative Temperature for analysing ocean observations and in ocean models.

The Chairman of the IOC, the President of SCOR and the President of IAPSO announced the replacement of EOS-80 with TEOS-10 in a series of scientific journals including the following announcements in Ocean Modelling and Deep-Sea Research,

```
https://www.sciencedirect.com/science/article/pii/S1463500311001545
or https://doi.org/10.1016/S1463-5003(11)00154-5
```

and

https://www.sciencedirect.com/science/article/pii/S0967063711001348 or https://doi.org/10.1016/j.dsr.2011.07.005

In these announcements it says, among other things, "In particular, Conservative Temperature  $\Theta$  accurately represents the "heat content" per unit mass of seawater, and is to be used in place of potential temperature  $\theta$  in oceanography."

We agree that the idea of Conservative Temperature predates WG-127. The history of Conservative Temperature in TEOS-10 can be read in the following article on the history of the development of TEOS-10 in this paper on which Remi Tailleux is a co-author (see particularly pages 167-170):

https://os.copernicus.org/articles/8/161/2012/ (Pawlowicz, R., T. McDougall, R. Feistel and R. Tailleux, 2012: An historical perspective on the development of the Thermodynamic Equation of Seawater – 2010: *Ocean Sci.*, **8**, 161-174. http://www.ocean-sci.net/8/161/2012/os-8-161-2012.pdf )

In brief, the idea that potential enthalpy might be a good heat-like variable came to TMcD on Cape Cod while swimming in Crooked Pond before breakfast on 1st July 1994. By morning tea that day he was convinced that the idea was viable. It was presented to Nick Fofonoff in the form of a seminar to him and his students a few weeks later. Nick had no objections to the idea, but he must have then forgotten about it as he thereafter co-authored the Bacon and Fofonoff (1996) paper. The idea did not appear in the literature until nine years after 1994, in McDougall (2003), although another paper [Cunningham, S. A., 2000: Journal of Marine Research, 58, 1-35.] did use the idea, quoting a pre-print of what became McDougall (2003).

Perhaps we should have said "officially define" instead of "define" – since any paper can define something but TEOS-10 has an official standing. We have thus replaced "to define a new variable," with "to adopt …".

17. Lines 180-183. The question of why potential temperature is non-conservative was actually answered earlier by Tailleux (2010), who showed that the non-conservation of any heat variable is dictated by the first law of thermodynamics (the law of total energy conservation). This is again a revisionist view of history where Graham and McDougall (2013) attempts to get credit for something that needs to be attributed to Tailleux (2010).

We disagree. The original development of the non-conservation of potential temperature, entropy and in particular, Conservative Temperature, was published by McDougall (2003), not Tailleux (2010). Moreover, as explained in the Appendix to this reply, the Tailleux (2010) paper claims to be based on the assumption that the Total Energy is a conservative variable. This is incorrect; Total Energy is not a conservative variable. However, in fact, Tailleux (2010) ignored the key non-conservative production term,  $-\nabla \cdot (P\mathbf{u})$ , and actually arrived at correct expressions for the non-conservative production of potential temperature, Conservative Temperature and of entropy (Eqn. (B.7) of Tailleux (2010) and Eqn. (B10) of Graham and McDougall (2013) are identical). But these expressions are written in terms of the molecular fluxes of heat and salt; fluxes that include the Soret and Dufour effects. In their Appendix B, Graham and McDougall (2013) showed that the expressions of Tailleux (2010), being written in terms of the molecular fluxes with turbulent ones; doing so leads to disobeying the

Second Law of Thermodynamics.

By contrast, McDougall (2003) and Graham and McDougall (2013) derived their key results by considering turbulent mixing in the ocean. Their equations do obey the Second Law of Thermodynamics, and because (apart from double-diffusive effects) mixing processes in the ocean are turbulent, the approach of these papers is applicable to the ocean, and to ocean models.

18. Lines 199 and below. How conservative is conservative temperature? As said above, this section only describes the non-conservation of Conservative Temperature arising from diffusive mixing and completely omits viscous dissipation. Given that the latter dominates, this section is at best misleading.

We have now taken the opportunity to repeat a reference, at this place in the text, to the fact that  $\varepsilon$  also contributes to the non-conservation of  $\Theta$ . This is also the place in our manuscript where we have inserted reference to the erroneous assumption made by both Tailleux (2010) and Tailleux (2015) that Total Energy  $\mathcal{E}$  is conserved when fluid parcels mix.

19. Line 382. What the authors call naive is what I call common sense. The authors should expect that many oceanographers will feel insulted here.

We do not aim to insult. We have thus replaced "naïve" with "At first glance then, it seems reasonable..."

20. Line 448. I dispute that this is thermodynamic best practice if the authors fail to understand the results of Tailleux (2010).

We hope that the Appendix to this reply makes it clear that the authors do understand the results of Tailleux (2010) and Tailleux (2015), but that that they disagree with the foundation of these papers. Both of these papers stated that Total Energy  $\mathcal{F}$  is a conservative quantity (which it is not), but then these papers ignored the non-conservative production term,  $-\nabla \cdot (P\mathbf{u})$ , that appears in the evolution equation of  $\mathcal{F}$ . By making these two self-balancing errors, Tailleux (2010) arrived at correct expressions for the non-conservative production of potential temperature, Conservative Temperature and of entropy, but these expressions were written in terms of the molecular fluxes of heat and salt, and since the mixing processes at work in the ocean are turbulent fluxes, these expressions are not immediately applicable. This is explained in more detail in our Appendix to this Reply.

21. Lines 479. The authors confuse the terms "contradiction" and "approximation" -- This is an idiosyncratic interpretation because this is not how idealized modelling should be viewed. Numerical modellers and oceanographers understand that potential temperature is non-conservative; it is therefore unfair to accuse them of assuming potential temperature to be conservative. Rather, they treat it as conservative because they assume that the small non-conservative terms and heat flux errors do not matter on the time scales generally considered. "Contradiction" and "approximation" are two completely different concepts, which it is crucial to distinguish in the present discussion, since the authors use the first interpretation in order to be able to accuse EOS-80 based modelling practice as being illogical. The interpretation "Models

assume the potential temperature to be conservative" has been disproven and there is no reason to accuse ocean modellers of ignoring this result. On the other hand, the interpretation that "models assume that the non-conservation of potential temperature is sufficiently small that it can be neglected in practice" has not been disproven yet. Indeed, disproving such an approach can only be achieved by running an EOS-80 based model with the incorrect and corrected equation for potential temperature. Such experiments are urgently needed so that we can stop with all the speculation.

We appreciate the need to distinguish "contradiction" (a logical fallacy) from "approximation". Our usage of "contradiction" is clearly in reference to the errors associated with the interpretation of the model's temperature variable as potential temperature rather than Conservative Temperature, with "contradiction" referring to the inaccuracies that are now well documented in the literature. Furthermore, there are numerical modellers co-authoring this paper, and they have no concern with pointing out these inconsistencies. The modelling community must move forward. Furthermore, as explained in our response to RT's comments on lines 64-65, the proposal from Tailleux (2015) that potential temperature should be retained as the temperature variable of an ocean model, together with adding source terms, is unworkable, because it cannot be done accurately.

22. Lines 735. It is not true that Conservative Temperature is consistent with the first law of thermodynamics, because it assumes that all the heat goes into heat and none into work. Indeed, it is well known from Lorenz's theory of available potential energy that there is about 0.5TW of the surface buoyancy fluxes going into the production of available potential energy. This suggests that potential enthalpy includes APE -- a work-like quantity -- as well as heat, not just heat.

This point has been well discussed in the literature, namely that changing from potential temperature to  $\Theta$  reduces the non-conservative diffusive source terms' contribution to heat fluxes (additional to the  $\varepsilon$  term) by a factor of ~100. Importantly, in adopting  $\Theta$  in place of potential temperature, there has been no claim that there is zero production of available potential energy. We do not say or imply that.

McDougall (2003) and Graham and McDougall (2013) avoid these pesky issues surrounding Total Energy, available potential energy and the like. Rather than being concerned with variables such as these Total-Energy-and-its-constituent energy components, McDougall (2003) and Graham and McDougall (2013) consider quite a different variable, namely the potential enthalpy  $h^m$  referenced to the pressure  $p^m$  at which an individual mixing event takes place. The pursuit of  $h^m$  has proved rewarding ever since 1st July 1994.

**Summary of our Reply to the Reviewer**

The main disagreements that this reviewer has with our paper are based on the following three viewpoints with which we disagree,

- a concentration by the reviewer only on the terms that contribute to the globally volume-integrated heat content of the ocean, rather than also being concerned with errors in modelling temperature in specific geographic locations,
- (ii) a hope (that also pervades the Tailleux (2015) paper), that retaining potential temperature as a model's temperature variable could be made to be competitive, in terms of accuracy and practicality, compared with adopting Conservative Temperature in numerical ocean circulation models, and,
- (iii) what we believe is an error made in Tailleux (2010) and Tailleux (2015) in the choice of physical property that is assumed to be conserved under turbulent mixing.

The error made by Tailleux (2010) and Tailleux (2015), referred to in point (iii) above, underlie many of this reviewer's comments; in particular, his "Specific comments" numbered 4, 9, 10, 11, 17, 20 and 22. This error is the assumption that Total Energy,  $\mathcal{T} = u + \frac{1}{2}\mathbf{u}\cdot\mathbf{u} + \Phi$ , is a conservative variable. Because this issue underlies so many of the reviewer's comments and is central to both the Tailleux (2010) and Tailleux (2015) papers, TMcD directly addresses this issue in the following Appendix rather than in our direct response to the review.

The research of McDougall (2003), Graham and McDougall (2013), Tailleux (2010) and Tailleux (2015) have all followed a similar path in that they worked from a variable that was taken to be conserved when two fluid parcels mixed in the ocean, and then they used this assumed knowledge to derive expressions for the non-conservation of  $\Theta$ ,  $\theta$  and  $\eta$ . While the approach of Tailleux (2010, 2015) is flawed because Total Energy was taken to be to be conservative, these papers also ignored the key non-conservative term,  $-\nabla \cdot (P\mathbf{u})$ , so that they actually arrived at the correct evolution equations for  $\Theta$ ,  $\theta$  and  $\eta$  (for example, Eqn. (B.7) of Tailleux (2010) and Eqn. (B10) of Graham and McDougall (2013) are identical). However, these equations are written in terms of the molecular fluxes of heat and salt, and it is not possible to use these expressions to evaluate the non-conservation of these variables for turbulent mixing processes. If this is attempted, an erroneous term of magnitude  $0.2\rho\varepsilon/\hat{h}_{\Theta}$  appears.

Before beginning the Appendix, we authors admit that two of us have given lectures and drafted other material that has also stated that Total Energy is a conservative variable (which it is not). These statements have since been corrected.

**Appendix:** Total Energy $\mathcal{E} = u + \frac{1}{2}\mathbf{u} \cdot \mathbf{u} + \Phi$ is not a conservative variable**

**24th April 2021 by Trevor J McDougall**

In this appendix we prove that Total Energy  $\mathcal{E} = u + \frac{1}{2}\mathbf{u} \cdot \mathbf{u} + \Phi$  is not a conservative variable. As a reminder, a variable *C* is called "conservative" if, when two fluid parcels are turbulently mixed together, the total amount of *C*-substance in the mixed fluid is the sum of the amount of *C*-stuff in the two initial fluid parcels.

We note that the papers of McDougall (2003) and Graham and McDougall (2013), and the approach taken by TEOS-10 have based their analyses upon

- (i) the conservation of potential enthalpy referenced to the pressure of a mixing event,
- (ii) rather than [as advocated in Tailleux's review of our manuscript, and by Tailleux (2010) and Tailleux (2015)] upon the assumed conservation Total Energy  $\mathcal{E} = u + \frac{1}{2}\mathbf{u} \cdot \mathbf{u} + \mathbf{\Phi}$ .

**The evolution equations of several energy-type variables**

Here the evolution equation equations are listed for several variables, copied from the TEOS-10 Manual, IOC et al. (2010), (but here without the terms due to the non-conservation of Absolute Salinity).

$$\left( \rho \left[ \frac{1}{2} \mathbf{u} \cdot \mathbf{u} + \Phi \right] \right)_{t} + \nabla \cdot \left( \rho \mathbf{u} \left[ \frac{1}{2} \mathbf{u} \cdot \mathbf{u} + \Phi \right] \right)$$

$$= \rho d \left( \frac{1}{2} \mathbf{u} \cdot \mathbf{u} + \Phi \right) / dt = -\mathbf{u} \cdot \nabla P + \nabla \cdot \left( \rho v^{\text{visc}} \nabla \frac{1}{2} \left[ \mathbf{u} \cdot \mathbf{u} \right] \right) - \rho \varepsilon.$$
(B.12)

$$\rho\left(\frac{\mathrm{d}h}{\mathrm{d}t} - \frac{1}{\rho}\frac{\mathrm{d}P}{\mathrm{d}t}\right) = \rho\left(\frac{\mathrm{d}u}{\mathrm{d}t} + P\frac{\mathrm{d}v}{\mathrm{d}t}\right) = \rho\left(\left(T_0 + t\right)\frac{\mathrm{d}\eta}{\mathrm{d}t} + \mu\frac{\mathrm{d}S_{\mathrm{A}}}{\mathrm{d}t}\right)$$
$$= -\nabla \cdot \mathbf{F}^{\mathrm{R}} - \nabla \cdot \mathbf{F}^{\mathrm{Q}} + \rho\varepsilon.$$
(B.19)

$$\left(\rho \mathcal{E}\right)_{t} + \nabla \cdot \left(\rho \mathbf{u} \mathcal{E}\right) = \rho \,\mathrm{d} \mathcal{E}/\mathrm{d} t = -\nabla \cdot \left(P \,\mathbf{u}\right) - \nabla \cdot \mathbf{F}^{\mathrm{R}} - \nabla \cdot \mathbf{F}^{\mathrm{Q}} + \nabla \cdot \left(\rho v^{\mathrm{visc}} \nabla \frac{1}{2} \left[\mathbf{u} \cdot \mathbf{u}\right]\right). \tag{B.15}$$

$$\left(\rho\mathcal{B}\right)_{t} + \nabla \cdot \left(\rho \mathbf{u} \mathcal{B}\right) = \rho \,\mathrm{d} \mathcal{B}/\mathrm{d} t = \partial P/\partial t \Big|_{x,y,z} - \nabla \cdot \mathbf{F}^{\mathrm{R}} - \nabla \cdot \mathbf{F}^{\mathrm{Q}} + \nabla \cdot \left(\rho v^{\mathrm{visc}} \nabla \frac{1}{2} \left[\mathbf{u} \cdot \mathbf{u}\right]\right). \tag{B.17}$$

$$\partial(\rho u)/\partial t + \nabla \cdot (\rho \mathbf{u} u) = \rho \frac{\mathrm{d} u}{\mathrm{d} t} = -P \frac{1}{v} \frac{\mathrm{d} v}{\mathrm{d} t} - \nabla \cdot \mathbf{F}^{\mathrm{R}} - \nabla \cdot \mathbf{F}^{\mathrm{Q}} + \rho \varepsilon.$$
(B.18)

$$\partial(\rho h)/\partial t + \nabla \cdot (\rho \mathbf{u} h) = \rho \frac{\mathrm{d} h}{\mathrm{d} t} = \frac{\mathrm{d} P}{\mathrm{d} t} - \nabla \cdot \mathbf{F}^{\mathrm{R}} - \nabla \cdot \mathbf{F}^{\mathrm{Q}} + \rho \varepsilon.$$
(A.13.2)

Eqn. (B.12) is the evolution equation for the sum of kinetic and gravitational potential energies. Eqn. (B.19) is the First Law of Thermodynamics (with its first line being the Fundamental Thermodynamic Relationship), while Eqns. (B.15) and (B.17) are, respectively, the evolution equations for Total Energy,  $\mathcal{I} = u + \frac{1}{2}\mathbf{u}\cdot\mathbf{u} + \Phi$ , and of the Bernoulli function,  $\mathcal{B} = h + \frac{1}{2}\mathbf{u}\cdot\mathbf{u} + \Phi$ . Eqn. (B18) is the evolution equation for internal energy, u, and Eqn. (A.13.2) is the evolution equation for enthalpy, h. In these equations the source term due to the non-conservative nature of Absolute Salinity has been omitted, for simplicity. Specific entropy is  $\eta$ . The radiative flux of heat is  $\mathbf{F}^{R}$ , while  $\mathbf{F}^{Q}$  stands for the molecular heat flux and the air-sea and geothermal fluxes of enthalpy, while  $\boldsymbol{\varepsilon}$  is the rate of dissipation of kinetic energy.

**What have McDougall (2003), Graham & McDougall (2013) and TEOS-10 done?**

The benefits of potential enthalpy and Conservative Temperature have largely been justified in these papers of 2003, 2013, and in the TEOS-10 Manual, from the viewpoint of evolution equations, but the benefits, and the original motivation that led to McDougall (2003), can be summarized by the following physically motivated three-point argument.

- 1. First, note that the air-sea heat flux is a flux of potential enthalpy,  $h^0$ , namely the flux of potential enthalpy referenced to a fixed surface reference pressure [and  $h^0$  is, by decree, exactly  $c_n^0$  times the flux of Conservative Temperature].
- 2. Second, note that the Taylor-series analysis of the non-conservative production of potential enthalpy,  $h^0$ , (as can be found in appendix A.18 of the TEOS-10 Manual) shows that while it is the potential enthalpy  $h^m$  referenced to the pressure  $p^m$  of a mixing event, that is conserved when parcels mix at  $p^m$ , a negligible error is made when  $\Theta$  is assumed to be conserved during the mixing event at pressure  $p^m$ . This estimating of the non-conservation of  $h^0$  proceeds from the conservation of  $h^m$  is described in detail in Graham and McDougall (2013) and is summarised in Appendix A.18 of the TEOS-10 Manual. Most of this small production of  $\Theta$  is due to the dissipation of kinetic energy,  $\varepsilon$ , and a much smaller part is due to the inherent (or diffusive) non-conservation of  $\Theta$ . The conservation equation for  $h^m$ , for a mixing process occurring at pressure  $p^m$  is

$$\frac{\partial(\rho h^m)}{\partial t} + \nabla \cdot (\rho \mathbf{u} h^m) = \rho \frac{\mathrm{d} h^m}{\mathrm{d} t} = -\nabla \cdot \mathbf{F}^{\mathrm{R}} - \nabla \cdot \mathbf{F}^{\mathrm{Q}} + \rho \varepsilon. \qquad (A.13.2 \text{ at } p^m)$$

3. Third, note that the ocean circulation can be decomposed into a series of adiabatic and isohaline movements during which  $\Theta$  is absolutely unchanged (because of its "potential" nature), followed by a series of turbulent mixing events during which  $\Theta$  is almost totally conserved (see point 2).

These three points, taken together, show that  $\Theta$  is the quantity that is advected and diffused in an almost conservative fashion and whose surface flux is exactly proportional to the air-sea heat flux.

Point 2 above relies on the conservative nature of enthalpy when mixing occurs at a given pressure,  $p^m$ . This conservative nature can be deduced by examining Eqn. (A.13.2). At this pressure  $p^m$ , the left-hand side of Eqn. (A.13.2) is the evolution equation of potential enthalpy,  $h^m$ , referenced to the pressure  $p^m$ , while the right-hand side contains (1) the divergence of molecular and radiative heat fluxes and (2) the positive-definite dissipation term  $\rho\varepsilon$ . Integrating over a material volume that is larger than, but contains the two parcels that are undergoing mixing (even when the volume changes shape and the volume decreases during the mixing process), we deduce that the non-conservation of potential enthalpy  $h^m$  is caused only by the dissipation term  $\rho\varepsilon$ . The use of  $h^m$  rather than enthalpy itself allows fluid parcels from say 1 dbar above  $p^m$  and 2 dbar below  $p^m$  to be moved adiabatically and isentropically to the pressure at which the mixing event occurs, while exhibiting no change in  $h^m$  during these motions that occur prior to the mixing event itself.

Before moving on, this volume-integrated conservation statement will be derived in detail. When Eqn. (A.13.2 at  $p^m$ ) is spatially and temporally integrated over a moving and contracting volume in which a mixing event occurs, the Leibnitz differentiation of the volume integral ensures that the relevant velocity that appears at the surface of the volume is the velocity *through* this moving boundary, the dia-surface velocity,  $\mathbf{u}^{dia}$ . This can be understood by

considering the time differentiation of the volume integral of the total amount of  $h^m$ -substance in the volume, as on the left-hand side of the first line of the equation below. Now, doing this differentiation of the volume integral with respect to time, the last term on the right-hand side of the first line of this equation arises from the fact that the boundary is moving through space, with  $\mathbf{u}^{\text{boundary}}$  being the velocity of the bounding surface of the volume. In the second line, Eqn. (A.13.2 at  $p^m$ ) has been used to replace the temporal derivative  $\partial(\rho h^m)/\partial t$  term that appears in the first line. The third line converts four of the volume integrals into boundary area integrals using the divergence theorem (and  $\mathbf{u}^{\text{dia}} = \mathbf{u} - \mathbf{u}^{\text{boundary}}$ ).

$$\frac{\partial}{\partial t} \left( \int_{V} \rho h^{m} dV \right) = \int_{V} \left( \rho h^{m} \right)_{t} dV + \int_{S} \rho h^{m} \mathbf{u}^{\text{boundary}} \cdot d\mathbf{S}$$
$$= -\int_{V} \nabla \cdot \left( \rho h^{m} \mathbf{u} + \mathbf{F}^{R} + \mathbf{F}^{Q} \right) dV + \int_{S} \rho h^{m} \mathbf{u}^{\text{boundary}} \cdot d\mathbf{S} + \int_{V} \rho \varepsilon dV$$
$$= -\int_{S} \rho h^{m} \mathbf{u}^{\text{dia}} \cdot d\mathbf{S} - \int_{S} \left( \mathbf{F}^{R} + \mathbf{F}^{Q} \right) \cdot d\mathbf{S} + \int_{V} \rho \varepsilon dV$$

The volume we chose to examine is a material volume whose boundary moves with the fluid parcels. Since at the edge of this volume there is no mixing, the fluid and the volume move together and there is no dia-surface velocity, so the first term on the right-hand side of the last line is zero. Since there are no radiative or molecular fluxes of heat at the boundary of the volume, the middle terms on the last line are also zero. This leaves just the last term on this line of the equation. Hence we see that the difference between the final and the initial volume integrals of  $\rho h^m$ , that is, the difference in the total amount of  $h^m$ -substance in this volume after the mixing event compared with before the mixing event, is the time integral of the volume integral of  $\rho \varepsilon$  over the mixing event. No other energy-like variable is as clean as this.

In summary, McDougall (2003) and Graham and McDougall (2013), as well as the TEOS-10 recommendations, are based on the almost-conservative nature of  $h^m = h(S_A, \Theta, p^m)$ , for turbulent mixing process that occurs at pressure  $p^m$ . In contrast, Tailleux (2010, 2015) and the many remarks of this review of our paper, have assumed that a different variable, namely Total Energy  $\mathcal{T} = u + \frac{1}{2}\mathbf{u}\cdot\mathbf{u} + \Phi$ , is an almost-conserved variable when fluid parcels mix at  $p^m$ . We will now show that this is not the case. Moreover, the TEOS-10 approach has yielded expressions for the non-conservation of potential temperature, Conservative Temperature and entropy that apply to an ocean in which the mixing processes are turbulent, whereas the Tailleux (2010, 2015) papers have led to expressions for the non-conservation of these quantities that apply only when the mixing in the ocean is done by molecular diffusion.

**What did Tailleux (2010) and Tailleux (2015) do?**

**Background**

The papers Tailleux (2010), Tailleux (2015), and the review of our paper by Remi Tailleux, all take Total Energy  $\mathcal{I} = u + \frac{1}{2}\mathbf{u} \cdot \mathbf{u} + \mathbf{\Phi}$  to be a conservative variable. Here, following the work of McDougall, Church and Jackett (2003),

McDougall, T. J., J. A. Church and D. R. Jackett, 2003: Does the nonlinearity of the equation of state impose an upper bound on the buoyancy frequency? *Journal of Marine Research*, **61**, 745-764. http://dx.doi.org/10.1357/002224003322981138

we will show that  $\mathcal{F}$  is not a conservative variable. Actually, after having assumed that  $\mathcal{F}$  is a conservative variable, the papers Tailleux (2010, 2015) then went on to ignore the non-conservative source term  $-\nabla \cdot (P\mathbf{u})$  that is part of the evolution equation for  $\mathcal{F}$ , and so these papers ended up deriving correct expressions for the non-conservative production of potential temperature, Conservative Temperature and of entropy. However these expressions were written in terms of the molecular fluxes of heat and salt, and, as shown in Appendix B of Graham and McDougall (2013), these expressions cannot be used by simply replacing the molecular fluxes with turbulent ones. Hence these expressions are not able to be used in the ocean or in ocean models where the mixing processes are predominantly turbulent.

**What about the $-\nabla \cdot (P\mathbf{u})$ term when mixing occurs?**

The advantage of the evolution equation for the Total Energy  $\mathcal{E} = u + \frac{1}{2}\mathbf{u}\cdot\mathbf{u} + \Phi$ , Eqn. (B.15), is that its right-hand side is the divergence of a flux. This is by construction; this is how the Total Energy evolution equation is derived. Given the divergence nature of the right-hand side of Eqn. (B.15), there is a tendency to assume that this means that Total Energy is conserved when two fluid parcels mix. This is not the case, as will now be demonstrated.

Following McDougall, Church and Jackett (2003), consider the mixing of two seawater parcels at pressure  $p^m$  in a single vertical water column (say, in a tall rigid tube of constant area, as illustrated in Figure 4 of that paper, repeated below). After the parcels have mixed, the final volume of the mixed fluid is less than the sum of the two initial volumes (note that this non-conservative decrease in volume on mixing is a general property of seawater and many other, although not all, fluid mixtures), no matter what the initial contrasts in temperature and salinity, see Graham and McDougall (2013)). This contraction-on-mixing at  $p^m$  means that due to this mixing, the total ocean volume has decreased, with seawater parcels at shallower locations slumping to slightly greater depths. When considering all these fluid parcels that lie above the mixing event, the  $-\nabla \cdot (P\mathbf{u})$  source term can be written as the non-zero term  $-\nabla \cdot (P\mathbf{u}) = -\mathbf{u} \cdot \nabla P = \partial P / \partial t \Big|_{x, y, z}$  since all the fluid parcels move in an adiabatic, isohaline and isobaric manner with  $\nabla \cdot \mathbf{u} = 0$  and with dP/dt = 0. These parcels, which were not involved in the mixing at pressure  $p^m$ , retain their Absolute Salinity, their Conservative Temperature, their enthalpy, their internal energy, and their pressure, but their height, z, and their geopotential,  $\Phi = gz$ , has decreased. These fluid parcels thus experience a decrease in their Total Energy, that is,  $d\mathcal{E}/dt < 0$ , even though they have experienced no mixing.

Now consider what occurs at the location of the mixing at  $p^m$ . While the average height and geopotential of this fluid has changed a little during the mixing, the larger contribution to the change in Total Energy  $\mathcal{L}$  is due to the work done by the surrounding ocean, via the pressure acting on the change in volume,  $-P\nabla \cdot \mathbf{u}$ . With the Absolute Pressure (in Pa) at the mixing location being  $P^m$ , the increase in the Total Energy of this mixed fluid arising from the

 $-\nabla \cdot (P\mathbf{u})$  term on the right-hand side of Eqn. (B.15) is approximately  $P^m$  times the volumeand-time integral of  $-\nabla \cdot \mathbf{u}$  which, by using the continuity equation, can be seen to be the volume-and-time integral of  $-v^{-1} dv/dt$  over the water being mixed. As shown by McDougall, Church and Jackett (2003), the gravitational potential energy that all the fluid above the mixing location loses (adiabatically and isentropically) due to contraction-on-mixing during the mixing process at  $p^m$ , is close to that gained by the mixed fluid in this non-conservative fashion as an increase of the mixed fluid's internal energy, u. The difference between these quantities is the work done by the atmospheric pressure acting on the changing height of the sea surface due to the contraction-on-mixing, since this adds to the depth-integrated Total Energy of the water column.

Figure 4. Sketch of the slumping of a strictly one-dimensional water column (like a very tall test tube) due to turbulent mixing in the vicinity of pressure  $p_m$ . Each parcel in the water column above  $p_m$  retains its enthalpy, its internal energy and its pressure. The reduction in gravitational potential energy of the entire water column appears as an increase in the internal energy of the fluid that undergoes the mixing.

There is a tendency to think that because the  $-\nabla \cdot (P\mathbf{u})$  term is the divergence of a flux, that it will integrate to zero over a patch of turbulent mixing, however this is not the case. The reason is that at the boundary of a fixed volume in space in which the mixing is occurring, the velocity of the fluid is non-zero, and part of this is due to the contraction-on-mixing. In the evolution equation (A.13.2 at  $p^m$ ) the divergence term  $\nabla \cdot (\rho \mathbf{u}h^m)$  can be integrated to zero over the mixing volume because of the presence of the temporal term,  $\partial (\rho h^m) / \partial t$ , in that equation (this was explained in detail on page 22 above). By contrast, in the case of the

 $-\nabla \cdot (P\mathbf{u}) = -\nabla \cdot (Pv \rho \mathbf{u})$  term, there is no corresponding temporal term,  $\partial (\rho Pv) / \partial t = \partial P / \partial t$ , in Eqn. (B.15) with which to form a paired Leibnitz differentiation of the volume integral. In fact, by adding and subtracting this temporal derivative term, one arrives at the evolution equation for the Bernoulli function, Eqn. (B.17).

To be totally explicit about the non-conservation of Total Energy, the analysis that is presented on page 22 above, is now performed for Total Energy,  $\mathcal{E} = u + \frac{1}{2}\mathbf{u} \cdot \mathbf{u} + \Phi$ .

$$\frac{\partial}{\partial t} \left( \int_{V} \rho \mathcal{F} \, \mathrm{d}V \right) = \int_{V} \left( \rho \mathcal{F} \right)_{t} \, \mathrm{d}V + \int_{S} \rho \mathcal{F} \mathbf{u}^{\text{boundary}} \cdot \mathrm{d}\mathbf{S}$$

$$= -\int_{V} \nabla \cdot \left( \rho \mathcal{F} \mathbf{u} + P \mathbf{u} + \mathbf{F}^{R} + \mathbf{F}^{Q} - \rho v^{\text{visc}} \nabla \frac{1}{2} \left[ \mathbf{u} \cdot \mathbf{u} \right] \right) \mathrm{d}V + \int_{S} \rho \mathcal{F} \mathbf{u}^{\text{boundary}} \cdot \mathrm{d}\mathbf{S}$$

$$= -\int_{S} \rho \mathcal{F} \mathbf{u}^{\text{dia}} \cdot \mathrm{d}\mathbf{S} - \int_{S} \left( \mathbf{F}^{R} + \mathbf{F}^{Q} - \rho v^{\text{visc}} \nabla \frac{1}{2} \left[ \mathbf{u} \cdot \mathbf{u} \right] \right) \cdot \mathrm{d}\mathbf{S} - \int_{S} P \mathbf{u} \cdot \mathrm{d}\mathbf{S}$$

Again, the right-hand side of the first line is the Leibnitz temporal differentiation of the volume integral on the left, the second line has used the evolution equation (B.15) for Total Energy, while the third line has collected two of the advection terms into one, involving the dia-surface velocity  $\mathbf{u}^{\text{dia}} = \mathbf{u} - \mathbf{u}^{\text{boundary}}$ . For our turbulent mixing scenario of the above figure, the first and middle surface integrals on the last line of this equation are both zero, but the last term, being the volume integral of  $-\nabla \cdot (P\mathbf{u})$  is non-zero, and because of this non-zero term, Total Energy  $\mathcal{X}$  is not a conservative variable.

**Total Energy is not a "potential" variable**

This thought process has shown that Total Energy is not conserved when two fluid parcels mix at a given pressure. Neither is Total Energy a "potential" variable; this can be seen as follows. Noting that the total energy  $\mathcal{F}$  is related to the Bernoulli function by  $\mathcal{F} = \mathcal{B} - Pv$  and even if we take the whole ocean to be in a steady state so that  $\mathcal{B}$  has the "potential" property (see Eqn. (B.17)), it is clear that  $\mathcal{F}$  does not have the "potential" property in this situation. That is,  $\mathcal{F}$  is not "quasi-material". In such a steady-state ocean, the change in  $\mathcal{F} = \mathcal{B} - Pv$  caused by an adiabatic and isohaline change in pressure of 1 Pa can be shown to be  $-v(1 - Pv/c^2)$  where c is the sound speed. This expression is only slightly different to -v, so that for an increase of pressure of 1000dbar, the decrease in total energy  $\mathcal{F}$  is approximately the same as that caused by a decrease in Conservative Temperature of ~2.4°C. This means that, even in a steady-state ocean, total energy  $\mathcal{F}$  is as useless as is enthalpy h as far as being a marker of fluid flow, since they are both very far from being "potential" variables.

**Total Energy is not a thermophysical variable**

Neither is Total Energy  $\mathcal{I}$  a thermophysical variable since it is not a function only of Absolute Salinity, temperature and pressure.

**$-\nabla \cdot (P\mathbf{u})$ is not a "reversible" mixing term**

The approach taken by all of the above papers, namely McDougall (2003), Graham and McDougall (2013), Tailleux (2010) and Tailleux (2015), is to start with a variable that is taken to be conserved (under certain assumptions), when seawater parcels mix at constant pressure  $p^m$ . This assumed knowledge is then used to find the expressions for the non-conservative production of Conservative Temperature, potential temperature, and of entropy. So, at the core of this technique is to begin with a variable that is conserved when fluid parcels mix at pressure  $p^m$ .

When potential enthalpy  $h^m = h(S_A, \Theta, p^m)$  is the variable-of-choice, the only non-conservation is due to the sign-definite dissipation of kinetic energy,  $\varepsilon$ ; the last term in Eqn. (A.13.2). When Total Energy  $\mathcal{T} = u + \frac{1}{2}\mathbf{u}\cdot\mathbf{u} + \Phi$  is taken to be the variable of choice, the non-conservation occurs even when there is no mixing taking place locally. Moreover, concentrating now on the location where the mixing is occurring, the non-conservative production of Total Energy is proportional to the absolute pressure,  $P^m$ , and from Figure 3(c) of McDougall, Church and Jackett (2003) we see that the non-conservative production of Total Energy  $\mathcal{T}$  is larger than the dissipation of kinetic energy  $\varepsilon$  in 1.8% of the volume of the global ocean.

However, in a further wrinkle, despite claiming that Total Energy is a conservative variable, Tailleux (2010) and Tailleux (2015) do not actually take the right-hand side of the Total Energy evolution Eqn. (B.15) equation to be zero. Rather, these papers assume that mixing does not contribute to the  $-\nabla \cdot (P\mathbf{u})$  term, and its influence is not examined. While this term is clearly the divergence of a flux, we have shown that this term cannot be ignored in the quest to quantify the non-conservation of variables such as  $\Theta$ ,  $\theta$  and  $\eta$ .

Rather, as we have shown above, the  $-\nabla \cdot (P\mathbf{u}) = -P\nabla \cdot \mathbf{u} - \mathbf{u} \cdot \nabla P$  term is non-zero when mixing occurs in the ocean; the first term on the right here,  $-P\nabla \cdot \mathbf{u}$ , causes a non-conservative increase in  $\mathcal{Z}$  at the location of the mixing, while the second term,  $-\mathbf{u} \cdot \nabla P$ , causes  $\mathcal{Z}$  to not be conserved when a water column loses gravitational potential energy, even when this occurs by an adiabatic and isentropic slumping motion, due, for example, to mixing occurring deeper in the water column.

**Evolution equations for $\Theta$ , $\theta$ and $\eta$ in terms of the molecular fluxes of heat and salt**

The evolution equation for  $\Theta$ , written in terms of the molecular fluxes of heat and salt, is

$$\rho \frac{\mathrm{d}\Theta}{\mathrm{d}t} = -\nabla \cdot \left( \frac{\mathbf{F}^{\mathrm{R}} + \mathbf{F}^{\mathrm{Q}} - \hat{h}_{S_{\mathrm{A}}} \mathbf{F}^{\mathrm{S}}}{\hat{h}_{\Theta}} \right) + \left( \mathbf{F}^{\mathrm{R}} + \mathbf{F}^{\mathrm{Q}} \right) \cdot \nabla \left( \frac{1}{\hat{h}_{\Theta}} \right) - \mathbf{F}^{\mathrm{S}} \cdot \nabla \left( \frac{\hat{h}_{S_{\mathrm{A}}}}{\hat{h}_{\Theta}} \right) + \frac{\rho \varepsilon}{\hat{h}_{\Theta}}$$
(CT molecular1)

(Tailleux (2010) and Eqn. (B4) of Appendix B of Graham and McDougall (2013)) where the partial derivatives  $\hat{h}_{\Theta}$  and  $\hat{h}_{S_A}$  of  $\hat{h}(S_A, \Theta, p)$  are functions of  $S_A$ ,  $\Theta$  and pressure. This equation can be found by simply rearranging the First Law of Thermodynamics, Eqn. (A.13.2), after taking enthalpy to be in the functional form  $\hat{h}(S_A, \Theta, p)$ . The molecular fluxes of salt and heat in this expression are given by the following expressions from the TEOS-10 Manual,

$$\mathbf{F}^{\mathrm{S}} = -\rho k^{\mathrm{S}} \left( \nabla S_{\mathrm{A}} + \frac{\mu_{P}}{\mu_{S_{\mathrm{A}}}} \nabla P \right) - \left( \frac{\rho k^{\mathrm{S}} T}{\mu_{S_{\mathrm{A}}}} \left( \frac{\mu}{T} \right)_{T} + \frac{B}{T^{2}} \right) \nabla T , \qquad (B.26)$$

$$\mathbf{F}^{\mathbf{Q}} = -\frac{1}{T^2} \left( C - \frac{B^2}{A} \right) \nabla T + \frac{B\mu_{S_{\mathbf{A}}}}{\rho k^S T} \mathbf{F}^{\mathbf{S}} = -\rho c_p k^T \nabla T + \frac{B\mu_{S_{\mathbf{A}}}}{\rho k^S T} \mathbf{F}^{\mathbf{S}}, \qquad (B.27)$$

which involve the molecular diffusivities of salt,  $k^{S}$ , and temperature,  $k^{T}$ , as well as the crossdiffusion coefficient *B*. These expressions contain the well-known cross-diffusion Soret and Dufour fluxes, as well as the flux of salt down the gradient of pressure.

Is this  $\Theta$  evolution equation, Eqn. (CT molecular1), useful in quantifying the nonconservation of  $\Theta$  in the ocean and in ocean models, given that we know that the dominant mixing processes are turbulent, not molecular? During the turbulent mixing process the average of the middle terms in this equation,  $(\mathbf{F}^{R} + \mathbf{F}^{Q}) \cdot \nabla(1/\hat{h}_{\Theta}) - \mathbf{F}^{S} \cdot \nabla(\hat{h}_{S_{A}}/\hat{h}_{\Theta})$ , using Eqns. (B.26) and (B.27) for  $\mathbf{F}^{S}$  and  $\mathbf{F}^{Q}$ , involve complicated products of the gradient of in situ temperature, the gradient of pressure, the gradient of Conservative Temperature, and the gradient of Absolute Salinity. These products of spatial gradients then need to be averaged over the time and space of the turbulent mixing event. This formidable set of correlations have not yet proved possible to understand. Eqn. (CT molecular1) can also be written in the form

c

$$\rho \frac{\mathrm{d}\Theta}{\mathrm{d}t} = -\nabla \cdot \mathbf{F}^{\Theta} - \frac{\mathbf{F}^{\Theta} \cdot \nabla \hat{h}_{\Theta}}{\hat{h}_{\Theta}} - \frac{\mathbf{F}^{S} \cdot \nabla h_{S_{\mathrm{A}}}}{\hat{h}_{\Theta}} + \frac{\rho\varepsilon}{\hat{h}_{\Theta}}$$
(CT molecular2)

where the molecular flux of Conservative Temperature  $\mathbf{F}^{\Theta}$  can be shown to be  $(\mathbf{F}^{R} + \mathbf{F}^{Q} - \hat{h}_{S_{A}}\mathbf{F}^{S})/\hat{h}_{\Theta}$ , but this does not simplify the problem of averaging over all the spatial gradients that are involved in the expressions for the molecular fluxes. We do know, from Appendix B of Graham and McDougall (2013), that it is not possible to simply replace the molecular fluxes of CT and salt in Eqn. (CT molecular2) with the corresponding turbulent fluxes. If this is attempted, an erroneous term of magnitude  $0.2\rho\varepsilon/\hat{h}_{\Theta}$  appears. Hence the laminar evolution equation of Conservative Temperature in which the fluxes are molecular fluxes, as derived in Tailleux (2010), does not lead to a means of accurately quantifying the nonconservation of Conservative Temperature in the turbulently mixed ocean.

**What other heat-like variables have been proposed in the literature?**

Prior to Conservative Temperature being adopted by the oceanographic community, the dominant heat-like variable whose net meridional flux in the ocean was compared to the corresponding airsea heat flux, was potential temperature multiplied by a fixed isobaric heat capacity.

Bacon and Fofonoff (1996) [Bacon, S. and N Fofonoff, 1996: Oceanic heat flux calculation. *J Atmos. Oceanic Technol.* **13**, 1327-1329] advocated a different measure of heat content for oceanographic use, namely,  $c_p(S_A, \theta, p^0) \theta$ , being potential temperature multiplied by the isobaric heat capacity that the seawater parcel would have if moved adiabatically and isentropically to the sea surface pressure. Section 7 of McDougall (2003) analysed this proposal and showed that this variable is no more accurate as a measure of the heat content of a fluid parcel than is  $\theta$ .

Warren (1999) [Warren, B. A., 1999: Approximating the energy transport across oceanic sections. *J. Geophys., Res.*, **104**, 7915-7919.] proposed the use of internal energy, *u*, but, as explained above, internal energy is not conserved when fluid parcels mix, with the non-conservation being due to the work done by the environment's pressure as the volume reduces. Warren (1999) also proposed  $\overline{c_p}\theta$  as an approximation to internal energy, where  $\overline{c_p}$  is the average value of the isobaric heat capacity evaluated at the sample's salinity, the sea surface pressure, and over a range of potential temperatures between zero Celsius and the parcel's potential temperature  $\theta$ . McDougall (2003) showed that this variable was not a particularly good approximation to internal energy, but that it was in fact equal to  $h^0(S_A, \theta) - h^0(S_A, 0)$ , that is, to the potential enthalpy of the fluid parcel minus the potential enthalpy of a fluid parcel at the same salinity but at zero Celsius temperature; clearly, this second term is unwanted.

This short discussion illustrates that various authors have searched for a heat-like variable whose transport in the ocean can be accurately compared with the air-sea heat flux.

**Summary of the Appendix**

This appendix can be summarized as follows. Graham and McDougall (2013) have demonstrated that potential enthalpy  $h^m$  referenced to the pressure  $p^m$  at which a mixing event occurs, is

- (i) a "potential" variable,
- (ii) a thermophysical variable, and
- (iii) an "isobaric conservative variable" for turbulent diffusive mixing processes at pressure  $p^m$ ; the dissipation of kinetic energy can be added as a source term.

In contrast, Total Energy  ${\mathcal I}$  has none of these three desirable attributes.

Tailleux (2010, 2015) claimed that Total Energy is a conservative oceanographic variable, but this is not the case; rather, there is the non-conservative source term  $-\nabla \cdot (P\mathbf{u})$  in the evolution equation for  $\mathcal{F}$ . Tailleux (2010, 2015) then went on to ignore this source term and actually arrived at the correct set of evolution equations for potential temperature, Conservative Temperature, and entropy, but these evolution equations were written in terms of the molecular fluxes of heat and salt, and are not applicable to the turbulently mixed ocean. So, to date, only the approach of McDougall (2003) and Graham and McDougall (2013) has yielded the evolution equation for Conservative Temperature that is applicable to a turbulent ocean.

---

## Referee Report (RR1)

**Review of:**
**The interpretation of temperature and salinity variables in numerical ocean model output, and the calculation of heat fluxes and heat content**
**By McDougall, Barker, Holmes, Pawlowicz, Griffies, and Durack**

**Summary and recommendation**

This main goal of this paper is to develop arguments seemingly making it possible for the potential temperature (PT) of EOS80-based models to be `interpretable' in some sense as Conservative Temperature (CT), which if true, would allow ocean modellers to directly compare the PT of EOS80 ocean models with the CT of TEOS10 ocean models, instead of comparing the like for like, which has been the accepted practice so far.

This paper is difficult to review and understand because it relies nearly exclusively on nonstandard arguments and abstract reasoning, as well as on nonconventional views about the nature of ocean models and of their dependent variables. In my first review, my initial reaction was that the paper had to be wrong, but I was not able to fully pinpoint exactly why. Having now read the paper 3 times and having had more time to reflect on its message, I now understand that the primary cause of my discomfort is the fact that this paper appears to assume that the evolution equation and boundary conditions satisfied by a physical quantity have some bearing on the definition of such a physical quantity, which seems to conflict with what is normally assumed in standard physics (at least, the way I understand it).

Take potential temperature for instance. As is well known, such a physical quantity is generally regarded as being rigidly defined as the notional temperature that a parcel would reach if brought adiabatically to the ocean surface at the mean atmospheric pressure, which is sufficient to fully define it. In particular, the evolution equation and boundary conditions that we may then formulate to predict its temporal definition are not normally supposed to have any bearing on its definition. Indeed, according to my understanding of physics, the definition of a physical quantity and its assumed evolution equation are generally regarded as entirely separate businesses. It follows that if we decide to approximate the evolution equation and boundary conditions for PT instead of using the most accurate one available, the usual view is that this will only introduce errors and uncertainties in the simulated PT, but not alter the character of PT itself. In this paper, however, the authors appear to take a different view. Specifically, they contend that if the evolution equation used to predict the temporal behaviour of PT is not the most accurate one available, but an approximated version of it that resembles the evolution equation for CT, then PT loses its PT character somehow to assume that of CT. Thus, even if PT is initialised with observed values of PT and if the equation of state used assumes PT as its argument, the authors contend that PT will drift towards CT after some `long spin-up time' if PT is treated as strictly conservative. (The authors do not provide the evolution equation supposed to be satisfied by the drift, nor do they discuss the physical quantities controlling the relaxation time scales controlling the drift, which would allow us to check the authors' views).

Having realised that the main cause of my discomfort was due to the authors allowing the evolution equation and boundary conditions satisfied by a physical quantity to interfere

with its definition, in contrast to what I think is the normal practice, it became much easier to understand the reasons for otherwise many very unclear statements and assertions. For instance, it now made more sense to me why Prof. McDougall and the authors would contend that potential enthalpy should be regarded as the variable defining heat content in the ocean. Indeed, a review of the literature on the subject (Bryan 1962; Bacon and Fofonoff 1996; Saunders 1995; Warren 1999) prior to McDougall (2003) clearly reveals that the quantity $c_{p0}\theta$ used so far as definition of heat content had been regarded as some approximation to the non-mechanical part of the total energy, its 'heat' quality resulting from the difficulty to convert it into mechanical energy, as per the second law of thermodynamics. Indeed, what standard thermodynamics tells us is that the 'heat' forms of energy cannot be converted with 100% efficiency into `work' forms of energy. In the classical view, therefore, 'heat' is regarded as a property of the fluid, as a form of energy that is difficult to convert into mechanical energy, irrespective of how surface heat transfer affects it. The authors appear to take a completely opposite view, however, namely that potential enthalpy is the relevant definition of heat because its surface flux captures the entirety of the surface heat transfer, irrespective of its degree of convertibility with mechanical energy. This is why in my first review I expressed the opinion that Prof. McDougall's approach was idiosyncratic, not to cause offence, but to point out how radically different its premise appeared to be compared to that of previous approaches. My criticism was addressed to the fact that the authors appeared to present their arguments in support of potential enthalpy as the relevant definition of heat as being self-evident, without mentioning to the reader how different its premise is compared to that of previous approaches, nor its ad-hoc character. For instance, adopting the authors' views, how would one define heat if the ocean were in fact primarily thermally forced along its uneven topography? As potential enthalpy referenced to a spatially varying bottom pressure? But then, heat would be a function of potential temperature, salinity, and horizontal position. Would that be acceptable? From a more fundamental viewpoint, shouldn't one seek a definition of heat that is equally applicable to the atmosphere as to the ocean?

To go full circle, the authors also contend that the salinity variables used in models should be interpreted as preformed salinity, on the grounds that current ocean models treat such variables as strictly conservative, which the authors argue is true only of preformed salinity. According to the authors, both temperature and salinity variables will 'drift' towards CT and S* after some undefined 'long spin-up time' regardless of how they are initially defined or initialised provide that they are treated as strictly conservative.

Because my understanding of physics is that the assumed evolution equation and boundary conditions of a physical quantity have no bearing on the definition of a physical quantity, my view is that the paper is based on unsound physics. Now, I also must acknowledge the fact that this paper touches on fundamental aspects of physics that are rarely if ever explicitly discussed. The fact that such eminent oceanographers appear to have such a different understanding of physics than I, combined with the fact that the views expressed in this paper did not appear to bother the second reviewer, Prof. Fox-Kemper, a lead author of a chapter in the latest IPCC report, suggests that the issues touched upon are not well understood by the community, and hence that there may be value in publishing this paper along my review in order for the community to reflect on where it stands on the issues discussed.

**Major issues**

**Potential source of divisions –** In the event that only part of the ocean modelling community adopts the authors' recommendations, with the remaining part disagreeing with them and therefore sticking to the currently accepted practices, what would be the authors' suggestion for resolving the resulting schism in the community? Wouldn't it be wise/useful for the SCW or the CLIVAR ocean modelling working group to organise some kind of world-wide poll about the issues discussed to identify to what extent oceanographers agree or disagree with the authors' view that it is ok for the evolution equation and boundary conditions to interfere with the definition of a physical quantity? I hope that the authors can agree that the development of incompatible ocean modelling practices cannot really be good for the field and is likely to complicate the writing up of the 'ocean' chapter of the next IPCC report. It seems to me that the authors should address this issue in their paper, i.e., the possibility that not everyone will agree with their recommendations.

**Specific comments**

Line 68 – Saying that density depends on 'heat content' is dangerous and provocative since the concept of 'heat content' is controversial and likely to remain so for the foreseeable future. Why not stick to uncontroversial and non-provocative practices?

Lines 70-71 – I don't understand the point here. The evolution equations for potential temperature and conservative temperature are non-conservative, whether we think it is useful or not. Treating such quantities as conservative necessarily entails an approximation that is the modellers' decision and has nothing to do with the conservativeness of the numerical schemes.

Lines 73-74 – Some numerical ocean models formulate their temperature equation in advective form, in which case the said conservative property is not satisfied (as far as I am aware)

Lines 77-79 – It is precisely for the same reasons that many scientists advocate that one should close the energy budget of ocean models, which cannot be done without retaining the non-conservation of heat in the temperature equation of ocean models, e.g., Tailleux (2010), Dewar, Shoonover, McDougall and Klein, Fluids (2016).
https://doi.org/10.3390/fluids1020009
The authors make the implicit assumption that models that treat their temperature variable as conservative but not their total energy are more reliable than the models doing the opposite, i.e., treat their total energy as conservative but not their temperature variable. Shouldn't this be left as an open issue for the community to think about?

Line 97 – I don't understand the term `contradictions' being used here. Anybody else would describe the approximations made as resulting in errors/uncertainties in need of being quantified, not contradictions. The terms neglected have been shown to be small, and therefore consistently neglected by ocean modellers. Even if one agreed to use the term

'contradiction' here, logic would dictate that resolving the contradiction would be to use the correct equation for potential temperature. Arguing that one should switch to conservative temperature may be a viable alternative, but it is not the logical choice that follows from saying that neglecting some terms in the potential temperature is illogical or contradictory, since the contradiction is eliminated by retaining the terms that the authors say one should not neglect.

Line 99 – May be the 'contradictions' that the authors refer to have been ignored because they are not real contradictions, and just simply because the terms neglected are so small that retaining them would not make any difference, which would be the natural thing to discuss.

Lines 103-104 – 'at the cost of introducing problems elsewhere'. This seems a very strange line of reasoning to me, as the analysis of the problem clearly reveals that the problem that the authors discuss can easily be corrected without introducing any problems elsewhere, by using the correct equation for potential temperature instead of the approximate one. I find it hard to understand why the authors find it worthwhile to discuss inferior solutions.

Lines 109-111 – "For example, the insistence that a model's temperature variable is potential temperature involves errors in the air-sea heat flux in some areas that are as large as the mean rate of current global warming'' – This is just not true, because the problem that the authors raise is not due to using potential temperature as such, but with not using the exact evolution equation and boundary conditions for it. As shown by Tailleux (2015), correcting the equation for potential temperature to address the authors' criticism would be very easy to implement. It is misleading for the authors to put the blame on potential temperature, where the blame lies in fact with ocean modellers not using the most accurate equation for potential temperature. Potential temperature is a great temperature variable, which has the properties that it has, and there is a priori no problem in using it if the correct evolution equations and boundary conditions are used. IOC et al. (2010) clearly misunderstands this. The best practice is not using CT instead of PT. The best practice is to use the most accurate evolution equations and boundary conditions, regardless of which variable is used, both variables being perfectly acceptable if used consistently.

Line 165 – 'has a mean non-conservation error' Why do the authors call it an 'error'? The non-conservation of any temperature variable is a real physical process as far as I am aware.

Line 165 – The authors need to say that the number of '0.3 mW m-2' relies on neglecting the non-conservation of potential enthalpy arising from the Joule heating due to the viscous dissipation rate, and that if the latter was retained, this number would be much larger and not that different from that for potential temperature.

Line 178 – As pointed out in my first review, the viscous dissipation must balance the mechanical power input by winds and tides, which provides a useful sanity check. 3 TW is a very lower bound for this, which amounts to 10 mW/m2, which is more than 3 times larger than that of Graham and McDougall, 2013. Graham and McDougall 2013 estimate is therefore implausibly too small.

Line 181 – Potential enthalpy had been in used as the thermodynamic variable used in the GISS model, as pointed out in my first review. It would seem justified to cite Russell et al. here, and point out that the variable has been in use way before McDougall (2003) re-discovered it.

Line 195 – Why is potential temperature not conservative?
This paragraph seems to mix up the a priori unrelated issues of 'conservativeness' and 'how to define heat content'. For clarity, it would be best to discuss the problem of how to define heat somewhere else, since the two issues are only indirectly related. I also find it strange that the section title only mentions potential temperature, given that the physical reasons why `heat'-like variables are non-conservative are a priori the same for PT and CT, so why leave CT out of the section title?

As to the explanation for non-conservativeness, the simplest in my view is to say that both PT and CT are non-conservative because:
- Neither PT nor CT mixes linearly (under diffusive effects alone), i.e., the PT or CT of the mixture of two water samples is different from the mass weighted average of the two samples.
- In a turbulent ocean, PT and CT also systematically increases during mixing events due to turbulent dissipation of kinetic energy by viscous processes

Then, the conditions can be separated for PT and CT. For instance, the authors could say that the non-conservativeness of CT is controlled by the temperature dependence of $T/\theta$ whereas the non-conservativeness of PT is controlled by the temperature dependence of $T\,c_{pr}/\theta$ – If $c_{pr}$ were assumed constant and $T/\theta$ were a function of pressure only, as for a dry atmosphere, then both CT and PT would be considerably more conservative than in seawater and would have identical degree of non-conservativeness. The fact that PT is less conservative than CT is due to the temperature dependence of $c_{pr}$ – with a lesser role for the salinity dependence. This is shown by Eqs (23) and (25) of Tailleux (2010), which I think the authors should refer to. The method developed by Tailleux (2010) (or Tailleux (2015)) is the most general currently available and is valid for the full Navier-Stokes equations. At the moment, this method is the one that underlies the construction of energetically consistent approximations. The methods discussed in IOC et al. (2010) and Graham and McDougall (2013) are much less general. Moreover, they fail to incorporate viscous dissipation as part of the definition of non-conservation of PT and CT.

Line 215 – "This suggestion has been made, for example, by Tailleux (2015)"
First, I don't think that the suggestion has been made by anybody else. Second, the method proposed by Tailleux (2015) is merely to make use of the passage relations $\theta = \theta(S_A, \Theta)$ and $\Theta = \Theta(S_A, \theta)$ to reformulate the evolution equation for CT used by a TEOS10 to obtain a mathematically equivalent one but for potential temperature. In other words, the evolution equation for potential temperature can be obtained by a simple change of variables from that for conservative temperature. Alternative, one could also diagnose the non-conservative terms in the evolution equation for potential temperature to close the energy budget of the EOS80 numerical ocean model considered, as in Tailleux (2010). Both strategies circumvent the difficulties raised by the authors and show that improving the equation for potential temperature would be a trivial exercise.

Given that both Tailleux (2010,2015) have proposed concrete solutions to compute the non-conservative terms to be added to the evolution equation for potential temperature, I find it odd and rather non-collegial for the authors to assert that such approaches would be unworkable. If the authors do not understand how to improve the evolution equation for potential temperature, this does not mean that is necessarily true of everybody else. Instead of unfairly disparaging Tailleux's work, the authors could simply say that Tailleux's suggestions remain to be implemented and tested and compared with a CT-based formulation.

Lines 239-242 – I think that the authors misunderstand and misrepresent Tailleux (2010,2015)'s approach. Indeed, Tailleux (2010)'s approach is fully deductive and rigorous, contrary to what the authors seem to suggest. Specifically, Tailleux's approach to obtain a mathematically explicit expression for the non-conservation of CT an PT is identical to that used by Prigogine and the Belgian school of non-equilibrium thermodynamics (improved by Lesley Woods, 1975) to obtain a mathematical expression for the non-conservative production of entropy. Physically, this approach consists in defining the non-conservation of specific entropy (and by extension that of CT or PT) as what is needed to make total energy conservative as per the law of energy conservation. Here, the term 'conservative' means that all the terms entering the evolution equation for total energy can be written as the divergence of a flux, which is the usual definition. In their paper, the authors seem to confuse the term 'conservative' with the property of 'mixing linearly', as when they say total energy is not conservative they clearly mean that total energy does not mix linearly. Saying that total energy is non-conservative is very confusing.

Now, the full evolution equation for the specific enthalpy in seawater takes the form:

$$\frac{Dh}{Dt} = -\frac{1}{\rho}\nabla \cdot \left(\rho F_q\right) + v\frac{Dp}{Dt} + \varepsilon_k$$

(see Eq. B19 of the latest version of TEOS10 manual with remineralisation term removed)
In a turbulent ocean, neither the pressure term nor turbulent viscous dissipation can be neglected, so Graham and McDougall (2003) assertion that the locally referenced potential enthalpy mixes linearly if one neglects viscous dissipation is inconsistent with the fact that pressure always fluctuates in a turbulent ocean. However, it appears to be true that the first-principles expressions for the non-conservation of CT and PT obtained by Tailleux (2010) can also be obtained by treating specific enthalpy as if it were linearly mixed, i.e., by omitting the pressure term in the above equation. However, because Tailleux (2010) and Tailleux (2015) are rooted in a fully deductive and first-principles approach, which is not the case of Graham and McDougall (2013), the correct way to justify the assumption made by Graham and McDougall (2013) is by showing that it follows from the exact results of Tailleux (2010), not the reverse.

Lines 253-256 – 'However, these expressions are written in terms of molecular fluxes and it is not possible to use these expressions to evaluate the non-conservation in a turbulently mixed ocean' I really don't understand where does this come from. Again, the authors appear to assume that because they do not know how to do something, this should also be the case of everybody else, which seems to me to go against the collegial nature of science.

Moreover, the Navier-Stokes equations are well accepted to describe both laminar and turbulent motions, so clearly Tailleux (2010,2015)'s expressions pertain to a turbulently mixed ocean, contrary to what the authors say. What is true, however, is that the expressions remain to be linked to turbulent fluxes or microstructure measurements in order to allow for their evaluation. One way this could be done is by using expressions such as the Osborn-Cox model linking the dissipation of temperature variance to the turbulent heat diffusivity as follows:

$$\kappa \frac{|\nabla T'|^2}{\left(\frac{d\overline{T}}{dz}\right)^2} = K_T$$

In the left-hand side, the terms involve the molecular fluxes of temperature as well as the mean temperature profile, whereas in the right-hand side appears the turbulent diffusivity for the mean temperature. The authors' remarks have incited me to rework on the issue in order to show that such expressions can indeed be linked to microstructure measurements and evaluated from first principles, as I hope to show in a forthcoming publication.

Line 331 – What this describes is 'density salinity' – My understanding is that density salinity is always different from absolute salinity except when all the haline contraction coefficients for each of the chemical constituents are identical. May be this can be mentioned and commented upon.

Line 341 – Can the authors clarify whether the relation is actually between S* and SA, or between S* and SD (density salinity).

Lines 361-367 – Can the authors comment on the differences in computational efficiency of the equation of state between the Jackett and McDougall (1995) and Roquet et al. (2015). This information is important for ocean modellers to decide whether to switch or not to switch.

Lines 381-392. I agree that TEOS10 has conclusively shown that variations in composition potentially matters for estimations of the thermal wind. However, it is also essential that the equations of motion used by ocean models be mathematically and physically well posed. As far as I understand the problem, while it is obvious that the equations of motions based on the use of reference composition salinity are well posed, it seems to me that this is not the case if we use absolute salinity (or rather density salinity). Moreover, as well as making the equations of motion ill posed, the use of density salinity also seems to screw up the energetics by introducing spurious sinks and sources of energy. As far as I am aware, TEOS10 never wrote down a mathematically consistent set of equations based on the use of absolute salinity. I would very much like to see the qualitative considerations about the importance of the variations in composition accompanied by the authors writing down a full set of equations of motion that can be studied by mathematicians and dynamicists like me. If the authors cannot produce a mathematically well posed set of equations using absolute salinity, it seems to me that they should not promote it as a meaningful basis for ocean modelling. If model equations using absolute are ill posed as I think they are, the

consequences is that it is a priori impossible to be sure of how to interpret the results of McCarthy et al. (2015). To me, this is a key issue that the IOC et al. (2010) and the authors appear to have overlooked.

Line 406 – What is the way to compute Cp(S*,theta,0) using the TEOS10 software? Can they provide the appropriate lines of code that would need to be invoked to compute it?

Line 412 – The fact that the authors use sometimes S*, sometimes SA is confusing.

Lines 414-415 – May be add a physical explanation for why the temperature or rain is not treated consistently.

Lines 444-445 – That's the authors interpretation. The alternative and more common interpretation is that these errors are accounted for in the estimation of errors and uncertainties affecting the simulated PT field.

Lines 453 – To ensure that the model equations are well posed, many ocean models will assume that salinity argument is reference salinity rather absolute salinity. Again, I have yet to see a consistent set of equations based on absolute salinity.

Lines 481-488 – I think that this paragraph is going to cause considerable confusion in the community as it seems inconsistent with the way things have been described before. First, IOC et al. (2010) says that the new equation of state is defined in terms of absolute salinity (while in fact using density salinity to estimate absolute salinity, even though the two are supposed to be somewhat different). Now, the authors appear to say that it is defined in terms of preformed salinity S*, which is always numerically different from absolute salinity. Does that mean that the authors are actually already moving away from the recommendations of TEOS10? Nothing of what the authors say about salinity in this paper makes any sense to me. I just don't understand where all this come from, and I suspect I won't be the only one. I think that it would greatly help if the authors could write the model equations that ocean modellers are supposed to solve with the proposed interpretation, may be in an appendix.

Lines 489-492 – What is this based on exactly?

Lines 499-503 – "The model's salinity variable will drift towards being preformed salinity." I really don't understand why. How can the authors make such an assertion without substantiating it. For instance, could the authors write down an evolution equation for the drift that would clarify the relaxation time scale and convince us that what the authors describe has a counterpart in the mathematical world?

Lines 536 – This is not how models work. Indeed, as far as I understand the issue, the temperature variable used by a model is not a matter of interpretation, it is a matter of declaration. The first step in constructing a model is to declare what its dependent variables should be. Once the variables have been declared, the next step is to decide on the evolution equations and boundary conditions that one will use to describe their temporal evolutions. To me, it is essential that models be based on precise definitions and

declarations, not interpretations, so that what we do can be easily understood by our colleagues mathematicians and atmospheric scientists. I am pretty sure that mathematicians cannot understand what the authors mean by 'interpretation', which is bound to leave them very confused. My impression is that the authors use the term 'interpretation' because they want the reader to accept their view that the physical meaning of the variables used by an ocean model is open to discussion, which seems questionable at best.

Line 543 – I disagree that this is a conclusion. It looks much more like an opinion or assertion. It would be useful if the authors could provide the reader with some experiments to run that would enable the ocean modelling community to test its validity.

Lines 556-557 – Again, my view is that the evolution equation and boundary conditions have no bearing on the definition of a physical quantity. For this statement to be acceptable, one has to accept that the definition of a physical quantity is not independent of its assumed evolution equation and boundary conditions, in contrast to what is generally done (as far as I understand it).

---

## Author Response (AR2)

We thank the reviewer for his latest reading of our manuscript, and for providing comments below (in black). Our responses appear in blue font.

**Review of:**
**The interpretation of temperature and salinity variables in numerical ocean model output, and the calculation of heat fluxes and heat content**
**By McDougall, Barker, Holmes, Pawlowicz, Griffies and Durack**

**Summary and recommendation**

This main goal of this paper is to develop arguments seemingly making it possible for the potential temperature ($\theta$) of EOS-80 based models to be `interpretable' in some sense as Conservative Temperature ($\Theta$), which if true, would allow ocean modellers to directly compare the $\theta$ of EOS-80 ocean models with the $\Theta$ of TEOS-10 ocean models, instead of comparing the like for like, which has been the accepted practice so far.

This is not quite what we have done in this manuscript. The first part of this sentence is fine, but from "which if true ...", should be replaced with "which if true, would allow ocean modellers to directly compare the temperature output of EOS-80 ocean models with the temperature output of TEOS-10 ocean models."

This paper is difficult to review and understand because it relies nearly exclusively on nonstandard arguments and abstract reasoning, as well as on nonconventional views about the nature of ocean models and of their dependent variables. In my first review, my initial reaction was that the paper had to be wrong, but I was not able to fully pinpoint exactly why. Having now read the paper 3 times and having had more time to reflect on its message, I now understand that the primary cause of my discomfort is the fact that this paper appears to assume that the evolution equation and boundary conditions satisfied by a physical quantity have some bearing on the definition of such a physical quantity, which seems to conflict with what is normally assumed in standard physics (at least, the way I understand it).

We do indeed say that the interpretation of an output variable of a model depends on the evolution equation and the boundary conditions of that variable in the model. Along with its initial condition, the evolution equation and boundary conditions fully determine the values that a variable takes throughout its evolution.

Take potential temperature for instance. As is well known, such a physical quantity is generally regarded as being rigidly defined as the notional temperature that a parcel would reach if brought adiabatically to the ocean surface at the mean atmospheric pressure which is sufficient to fully define it. In particular, the evolution equation and boundary conditions that we may then formulate to predict its temporal definition are not normally supposed to have any bearing on its definition. Indeed, according to my understanding of physics, the definition of a physical quantity and its assumed evolution equation are generally regarded as entirely separate businesses. It follows that if we decide to approximate the evolution equation and boundary conditions for $\theta$ instead of using the most accurate one available, the usual view is that this will only introduce errors and uncertainties in the simulated $\theta$, but not alter the character of $\theta$ itself. In this paper, however, the authors appear to take a different view. Specifically, they contend that if the evolution equation used to predict the temporal behavior of $\theta$ is not the most accurate one available, but an approximated version of it that resembles the evolution equation for $\Theta$, then $\theta$ loses its $\theta$ character somehow to assume that of $\Theta$. Thus, even if $\theta$ is initialised with observed values of $\theta$ and if the equation of state used assumes $\theta$ as its argument, the authors

contend that $\theta$ will drift towards $\Theta$ after some `long spin-up time' if $\theta$ is treated as strictly conservative. (The authors do not provide the evolution equation supposed to be satisfied by the drift, nor do they discuss the physical quantities controlling the relaxation time scales controlling the drift, which would allow us to check the authors' views).

This discussion does indeed reflect what we have said in the manuscript. The air-sea heat flux, and the lack of interior non-conservative source terms are both consistent with our initial hypothesis that the temperature output of an EOS-80 ocean model can be interpretated as being Conservative Temperature. With this as a possible interpretation, we then investigated the implications of making this interpretation, given the different equation of state that the EOS-80 model uses. We concluded that this hypothesis is a good one, and importantly, it avoids an embarrassing error in the air-sea heat flux that is associated with the usual potential temperature interpretation. While we have lived with this embarrassing error for more than a century, it is good to finally be rid of it.

Having realized that the main cause of my discomfort was due to the authors allowing the evolution equation and boundary conditions satisfied by a physical quantity to interfere with its definition, in contrast to what I think is the normal practice, it became much easier to understand the reasons for otherwise many very unclear statements and assertions. For instance, it now made more sense to me why Prof. McDougall and the authors would contend that potential enthalpy should be regarded as the variable defining heat content in the ocean. Indeed, a review of the literature on the subject (Bryan 1962; Bacon and Fofonoff 1996; Saunders 1995; Warren 1999) prior to McDougall (2003) clearly reveals that the quantity $c_{p0}\theta$ used so far as the definition of heat content had been regarded as some approximation to the non-mechanical part of the total energy, its 'heat' quality resulting from the difficulty to convert it into mechanical energy, as per the second law of thermodynamics. Indeed, what standard thermodynamics tells us is that the 'heat' forms of energy cannot be converted with 100% efficiency into 'work' forms of energy. In the classical view, therefore, 'heat' is regarded as a property of the fluid, as a form of energy that is difficult to convert into mechanical energy, irrespective of how surface heat transfer affects it. The authors appear to take a completely opposite view, however, namely that potential enthalpy is the relevant definition of heat because its surface flux captures the entirety of the surface heat transfer, irrespective of its degree of convertibility with mechanical energy. This is why in my first review I expressed the opinion that Prof. McDougall's approach was idiosyncratic, not to cause offence, but to point out how radically different its premise appeared to be compared to that of previous approaches. My criticism was addressed to the fact that the authors appeared to present their arguments in supportof potential enthalpy as the relevant definition of heat as being self-evident, without mentioning to the reader how different its premise is compared to that of previous approaches, nor its ad-hoc character. For instance, adopting the authors' views, how would one define heat if the ocean were in fact primarily thermally forced along its uneven topography? As potential enthalpy referenced to a spatially varying bottom pressure? But then, heat would be a function of potential temperature, salinity, and horizontal position. Would that be acceptable? From a more fundamental viewpoint, shouldn't one seek a definition of heat that is equally applicable to the atmosphere as to the ocean?

It is not that the boundary conditions and the evolution equation "interfere(s) with the definition" of a model's temperature variable. Rather than interfering with a definition, what we have done is to offer an explanation of a model variable. We make the point that if one chooses to interpret the temperature variable in an EOS-80 model as being potential temperature, then this comes at the cost of a substantial error in the air-sea heat flux; the ocean receives more heat than the atmosphere delivers to it! The alternative interpretation, namely that the EOS-80 model's temperature is Conservative Temperature does not suffer from this problem. Rather it suffers from the problem that the horizontal density gradient is a little in error. It is indeed

disappointing that not all ocean model codes have converted to the ten-year old equation of state, TEOS-10, but nevertheless, this manuscript will help those analyzing and comparing the outputs of ocean models based on EOS-80 and TEOS-10.

Regarding the remainder of the above paragraph, the non-conservation of Conservative Temperature has been extensively discussed in the literature in McDougall (2003) and in Graham and McDougall (2013). These papers discuss how the geothermal heat flux at depth can be converted into a flux of Conservative Temperature.

To go full circle, the authors also contend that the salinity variables used in models should be interpreted as preformed salinity, on the grounds that current ocean models treat such variables as strictly conservative, which the authors argue is true only of preformed salinity. According to the authors, both temperature and salinity variables will 'drift' towards θ and S* after some undefined 'long spin-up time' regardless of how they are initially defined or initialized, provided that they are treated as strictly conservative.

Yes, that's right. Preformed Salinity is a conservative variable and is almost exactly equal to both Absolute Salinity and Reference Salinity at the sea surface (where the sources of freshwater in the model are located). The CMIP models are typically spun up over some 2000 years which is plenty of time for some random initial condition to be flushed out of the coupled system.

Because my understanding of physics is that the assumed evolution equation and boundary conditions of a physical quantity have no bearing on the definition of a physical quantity, my view is that the paper is based on unsound physics. Now, I also must acknowledge the fact that this paper touches on fundamental aspects of physics that are rarely if ever explicitly discussed. The fact that such eminent oceanographers appear to have such a different understanding of physics than I, combined with the fact that the views expressed in this paper did not appear to bother the second reviewer, Prof. Fox-Kemper, a lead author of a chapter in the latest IPCC report, suggests that the issues touched upon are not well understood by the community, and hence that there may be value in publishing this paper along with my review in order for the community to reflect on where it stands on the issues discussed.

Rather than interfering with a definition of a variable, what we have done is to offer an explanation of the model's variables. When a coupled model is run, there is no high authority who decrees what the model variables are. Rather, these models contain evolution equations and boundary conditions, which together, dictate how all the variables evolve. It is then up to we scientists to interpret the variables in a consistent fashion that is cognizant of the equations and boundary conditions to which the variable has been subject during the running of the model.

**Major issues**

**Potential source of divisions –** In the event that only part of the ocean modelling community adopts the authors' recommendations, with the remaining part disagreeing with them and therefore sticking to the currently accepted practices, what would be the authors'suggestion for resolving the resulting schism in the community? Wouldn't it be wise/useful for the SCW [the JCS?] or the CLIVAR ocean modelling working group to organize some kind of world-wide poll about the issues discussed to identify to what extent oceanographers agree or disagree with the authors' view that it is ok for the evolution equation and boundary conditions to interfere with the definition of a physical quantity? I hope that the authors can agree that the development of incompatible ocean modelling practices cannot really be good for the field and is likely to complicate the writing up of the 'ocean' chapter of the next IPCC report. It seems to me that the authors should address this issue in their paper, i.e., thepossibility that not everyone will agree with their recommendations.

As with the development of TEOS-10, our view is that the science should first be published, and then the community can absorb what is published, and a consensus will likely emerge.

**Specific comments**

Line 68 – Saying that density depends on 'heat content' is dangerous and provocative since the concept of 'heat content' is controversial and likely to remain so for the foreseeable future. Why not stick to uncontroversial and non-provocative practices?

Thanks. We have adopted this suggestion; "temperature" now replaces "heat content".

Lines 70-71 – I don't understand the point here. The evolution equations for potential temperature and conservative temperature are non-conservative, whether we think it is useful or not. Treating such quantities as conservative necessarily entails an approximation that is the modellers' decision and has nothing to do with the conservativeness of the numerical schemes.

Here we discuss what ocean models actually do; they treat their temperature variable as being a conservative variable, so we have made no change to the text.

Lines 73-74 – Some numerical ocean models formulate their temperature equation in advective form, in which case the said conservative property is not satisfied (as far as I am aware).

All ocean models known to us formulate their temperature evolution equation in flux-divergence form, without any non-conservative interior source terms. This common practice was documented in Section 9.1 of the review paper Griffies et al (2000) (Ocean Modelling, vol 2, pages 123-192) where it is stated that "All models employ flux form advection schemes, hence allowing for trivial conservation of total tracer amount." That paper documented the practice in 12 community ocean models circa 2000. In the subsequent 21 years, there has been no change in regards to the practice of how ocean models formulate their tracer transport equation. Although numerical methods differ, the underlying equation remains flux-form.

We observe that some atmospheric weather models make use of the advective form to facilitate their use of semi-Lagrangian numerical methods. But it is well documented (e.g., Lin and Rood 1996, MWR vol 124; Lin 2004, MWR vol 132) that such advective form semi-Lagrangian tracer transport schemes do not conserve heat, water, and chemical tracers, thus making them unsuitable for seasonal to climate purposes or for any tracer studies.

Lines 77-79 – It is precisely for the same reasons that many scientists advocate that one should close the energy budget of ocean models, which cannot be done without retaining the non-conservation of heat in the temperature equation of ocean models, e.g., Tailleux (2010), Dewar, Shoonover, McDougall and Klein, *Fluids*, 2016, https://doi.org/10.3390/fluids1020009.  The authors make the implicit assumption that models that treat their temperature variable as conservative but not their total energy are more reliable than the models doing the opposite, i.e., treat their total energy as conservative but not their temperature variable.  Shouldn't this be left as an open issue for the community to think about?

Ocean models write their temperature evolution equation so that it is conserved in the ocean interior.  Also, as our manuscript points out, and as we showed in pages 20-28 of our response to this reviewer's first review, https://doi.org/10.5194/gmd-2020-426-AC1 , Total Energy is not a conservative variable (and so should not be treated as such when constructing an ocean model in the future).

In more detail, ocean models have, to date, assigned to both their temperature and salinity variables the following two properties,

1. the "potential property", and
2. the "conservative property".

The "potential property" means that when the pressure acting on a fluid parcel changes, without any exchange of heat or salt (that is, during an adiabatic and isentropic pressure change), then the property remains unchanged.  The "conservative property" means that when two fluid parcels are mixed, the total amount of that property in the original two fluid parcels is the same as the amount of that property in the final mixed sate.  Total Energy is not a potential variable, since an adiabatic and isentropic change in pressure alters the Total Energy of a fluid parcel.  Nor does Total Energy possess the conservative property, because both contraction-on-mixing and wave processes allow the internal energy (and Total Energy) of the final mixed fluid parcel to be different to the sum of the internal energies (and Total Energies) of the two original fluid parcels.

For a variable to possess the "conservative property", it is not sufficient that the material derivative of that property is given by the divergence of a flux.  Rather, what is needed is that the material derivative of a conservative variable must be equal to the divergence of a flux that is zero in the absence of mixing at that location.  That is, the flux whose divergence appears on the right-hand side of the evolution equation of a conservative variable must be a diffusive flux (whether a molecular or a turbulent type of diffusive flux).  This feature allows one to integrate over a region in which a mixing event is occurring and be confident that there is no flux through the bounding area that lies outside of the fluid that is being mixed.  This is not possible for Total Energy, because even when integrating out to a quiescent surface that encloses an isolated patch of turbulent mixing, the flux divergence term $-\nabla \cdot (P\mathbf{u})$ can still be non-zero there.

Note that the volume integral of Total Energy, integrated over the volume of the global ocean, only changes due to the surface boundary conditions (heat fluxes, and fluxes due to precipitation, evaporation, and changes in atmospheric pressure) at the sea surface and at the ocean floor.  Indeed, the fact that the material change of Total Energy is caused by the divergence of various fluxes is simply because of the way that the Total Energy evolution equation is constructed (see Appendix B of IOC et al., 2010).  That is, it is a direct result of our anthropogenic assertion that "Total Energy cannot be created or destroyed"; but note that it can be transported.  This property does not however bestow on Total Energy the conservative property, simply because one of the fluxes that appear on the right-hand side of the Total Energy equation is not a diffusive flux and does not go to zero in the absence of mixing processes.  Not only does contraction-on-mixing contribute to  $-\nabla \cdot (P\mathbf{u})$, but this term also has contributions from various wave processes.

Line 97 – I don't understand the term `contradictions' being used here. Anybody else woulddescribe the approximations made as resulting in errors/uncertainties in need of being quantified, not contradictions. The terms neglected have been shown to be small, and therefore consistently neglected by ocean modellers. Even if one agreed to use the term 'contradiction' here, logic would dictate that resolving the contradiction would be to use the correct equation for potential temperature. Arguing that one should switch to conservative temperature may be a viable alternative, but it is not the logical choice that follows from saying that neglecting some terms in the potential temperature is illogical or contradictory, since the contradiction is eliminated by retaining the terms that the authors say one should not neglect.

We retain the text as is, since to interpret the temperature variable of an EOS-80 ocean model as being potential temperature means that a different heat flux enters the ocean than leaves the atmosphere. This is contradictory, and it affects a quantity (the air-sea heat flux) that is of immense importance for modelling climate. An error in an equation of state, or the absence of source terms in a salt or temperature budget have the nature of approximations to ocean physics (similar to the choice of an uncertain turbulent diffusivity), but to make an (avoidable) error in the driving air-sea heat flux of an ocean model is a more basic and more embarrassing physical error that, in 2021, we should no longer countenance.

Line 99 – May be the 'contradictions' that the authors refer to have been ignored because they are not real contradictions, and just simply because the terms neglected are so small that retaining them would not make any difference, which would be the natural thing to discuss.

As we show in the manuscript, the damage that is done to the air-sea heat flux at a given horizontal location by the interpretation that the temperature variable of an EOS-80 ocean model is potential temperature is not small in comparison to the globally averaged rate that our planet is being anthropogenically warmed. That is, in regions that are comparable in area to an ocean basin, a heat budget analysis using EOS-80 and potential temperature would find a false trend as large as the globally averaged rate that our planet is warming.

Lines 103-104 – 'at the cost of introducing problems elsewhere'. This seems a very strange line of reasoning to me, as the analysis of the problem clearly reveals that the problem that the authors discuss can easily be corrected without introducing any problems elsewhere, by using the correct equation for potential temperature instead of the approximate one. I find it hard to understand why the authors find it worthwhile to discuss inferior solutions.

As we show in the manuscript, it is not possible to accurately formulate the correct air-sea flux of potential temperature, nor is it possible to accurately include the interior non-conservative source term for the potential temperature evolution equation.

Lines 109-111 – "For example, the insistence that a model's temperature variable is potential temperature involves errors in the air-sea heat flux in some areas that are as large as the mean rate of current global warming'' – This is just not true, because the problem that the authors raise is not due to using potential temperature as such, but with not using the exact evolution equation and boundary conditions for it. As shown by Tailleux (2015), correcting the equation for potential temperature to address the authors' criticism would be very easy to implement. It is misleading for the authors to put the blame on potential temperature, where the blame lies in fact with ocean modellers not using the most accurate equation for potential temperature. Potential temperature is a great temperature variable, which has the properties that it has, and there is a priori no problem in using it if the correct evolution equation and boundary conditions are used. IOC et al. (2010) clearly misunderstands this. The best practice is not using $\theta$ instead of $\theta$. The best practice is to use the most accurate evolution equations and boundary conditions, regardless of which variable is used, both variables being perfectly acceptable if used consistently.

Again, it is not possible to accurately do as the reviewer suggests here, and this is why IOC et al. (2010) adopted Conservative Temperature as the temperature to be pursued in ocean

models.  We show in the manuscript, it is not possible to accurately formulate the correct air-sea flux of potential temperature, nor is it possible to accurately include the interior non-conservative source term for the potential temperature evolution equation.  It is hard to see any advantage in attempting to bandage up potential temperature when the adoption of Conservative Temperature is the much cleaner approach.  The evolution equations for potential temperature, for Conservative Temperature and for entropy were derived and compared in McDougall (2003) and in Graham and McDougall (2013), and there are clear benefits in adopting Conservative Temperature in ocean models.  The choice to adopt Conservative Temperature was made well after the evolution equations for potential temperature and Conservative Temperature were known, and it was made after considering both options (see the discussion of this point in McDougall, 2003).

Line 165 – 'has a mean non-conservation error'.  Why do the authors call it an 'error'?  The non-conservation of any temperature variable is a real physical process as far as I am aware.

The "error" refers to the error made when the non-conservative terms are ignored.

Line 165 – The authors need to say that the number of '0.3 mW m-2' relies on neglecting the non-conservation of potential enthalpy arising from the Joule heating due to the viscous dissipation rate, and that if the latter was retained, this number would be much larger and not that different from that for potential temperature.

Thanks.  We agree and this has now been added.

Line 178 – As pointed out in my first review, the viscous dissipation must balance the mechanical power input by winds and tides, which provides a useful sanity check.  3 TW is a very lower bound for this, which amounts to 10 mW/m2, which is more than 3 times larger than that of Graham and McDougall, 2013.  The Graham and McDougall 2013 estimate is therefore implausibly too small.

Thanks.  We agree that the Graham and McDougall (2013) estimate of the dissipation of kinetic energy is too small, as it doesn't adequately include the dissipation in the upper ocean.  The text has been changed accordingly.

Line 181 – Potential enthalpy had been in used as the thermodynamic variable used in the GISS model, as pointed out in my first review.  It would seem justified to cite Russell et al. here, and point out that the variable has been in use way before McDougall (2003) re-discovered it.

The use of potential enthalpy in ocean models by Russell et al (1995) was mentioned as far back as the original theoretical paper of McDougall (2003).

Line 195 – Why is potential temperature not conservative?
This paragraph seems to mix up the a priori unrelated issues of 'conservativeness' and 'how to define heat content'. For clarity, it would be best to discuss the problem of how to define heat somewhere else, since the two issues are only indirectly related. I also find it strange that the section title only mentions potential temperature, given that the physical reasons why 'heat'-like variables are non-conservative are a priori the same for $\theta$ and $\Theta$, so why leave $\Theta$ out of the section title?

As to the explanation for non-conservativeness, the simplest in my view is to say that both $\theta$ and $\Theta$ are non-conservative because:

- Neither $\theta$ nor $\Theta$ mixes linearly (under diffusive effects alone), i.e., the $\theta$ or $\Theta$ of the mixture of two water samples is different from the mass weighted average of the two samples.
- In a turbulent ocean, $\theta$ and $\Theta$ also systematically increases during mixing events due to turbulent dissipation of kinetic energy by viscous processes.

Then, the conditions can be separated for $\theta$ and $\Theta$. For instance, the authors could say that the non-conservativeness of $\Theta$ is controlled by the temperature dependence of $T/\theta$ whereas the non-conservativeness of $\theta$ is controlled by the temperature dependence of $T\, c_{pr}/\theta$. – If $c_{pr}$ were assumed constant and $T/\theta$ were a function of pressure only, as for a dry atmosphere, then both CT and $\theta$ would be considerably more conservative than in seawater and would have identical degree of non-conservativeness. The fact that $\theta$ is less conservative than $\Theta$ is due to the temperature dependence of $c_{pr}$ – with a lesser role forthe salinity dependence. This is shown by Eqs (23) and (25) of Tailleux (2010), which I think the authors should refer to. The method developed by Tailleux (2010) (or Tailleux (2015)) is the most general currently available and is valid for the full Navier-Stokes equations. At the moment, this method is the one that underlies the construction of energetically consistent approximations. The methods discussed in IOC et al. (2010) and Graham and McDougall (2013) are much less general. Moreover, they fail to incorporate viscous dissipation as part of the definition of non-conservation of $\theta$ and $\Theta$.

This discussion is incorrect on multiple grounds. First, Total Energy is not a conservative variable (see above) and so the derivation of Tailleux (2010, 2015) is flawed; that is, it is energetically inconsistent, as shown in our response to the reviewer's first review of this present paper. Second, it is not correct to say that McDougall (2003) and Graham and McDougall (2013) ignore the contribution of the dissipation of kinetic energy to the production of potential temperature and Conservative Temperature. This term appears in the relevant evolution equations in those papers.

Line 215 – "This suggestion has been made, for example, by Tailleux (2015)''.
First, I don't think that the suggestion has been made by anybody else. Second, the method proposed by Tailleux (2015) is merely to make use of the passage relations $\theta = \theta(S_A, \Theta)$ and $\Theta = \Theta(S_A, \theta)$ to reformulate the evolution equation for $\Theta$ used by a TEOS-10 model to obtain a mathematically equivalent one but for potential temperature. In other words, the evolution equation for potential temperature can be obtained by a simple change of variables from that for conservative temperature. Alternatively, one could also diagnose the non-conservative terms in the evolution equation for potential temperature to close the energy budget of the EOS-80 numerical ocean model considered, as in Tailleux (2010). Both strategies circumvent the difficulties raised by the authors and show that improving the equation for potential temperature would be a trivial exercise.

Given that both Tailleux (2010, 2015) have proposed concrete solutions to compute the non-conservative terms to be added to the evolution equation for potential temperature, I find it odd and rather non-collegial for the authors to assert that such approaches would be unworkable. If the authors do not understand how to improve the evolution equation for

potential temperature, this does not mean that is necessarily true of everybody else.  Instead of unfairly disparaging Tailleux's work, the authors could simply say that Tailleux's suggestions remain to be implemented and tested and compared with a Θ-based formulation.

     We stand by what we have written, and if further detail were needed it can be found in our response to the reviewer's first review, at https://doi.org/10.5194/gmd-2020-426-AC1.  Again, it is not possible to accurately do as the reviewer suggests here, and this is why IOC et al. (2010) adopted Conservative Temperature as the temperature to be pursued in ocean models.  We can see no advantage in attempting to approximately bandage up potential temperature when adopting Conservative Temperature is the much cleaner approach.

Lines 239-242 – I think that the authors misunderstand and misrepresent Tailleux (2010, 2015)'s approach.  Indeed, Tailleux (2010)'s approach is fully deductive and rigorous, contrary to what the authors seem to suggest.  Specifically, Tailleux's approach to obtain a mathematically explicit expression for the non-conservation of Θ an $\theta$ is identical to that used by Prigogine and the Belgian school of non-equilibrium thermodynamics (improved by Lesley Woods, 1975) to obtain a mathematical expression for the non-conservative production of entropy.  Physically, this approach consists in defining the non-conservation of specific entropy (and by extension that of Θ or $\theta$) as what is needed to make total energy conservative as per the law of energy conservation.  Here, the term 'conservative' means that all the terms entering the evolution equation for total energy can be written as the divergence of a flux, which is the usual definition.  In their paper, the authors seem to confuse the term 'conservative' with the property of 'mixing linearly', as when they say total energy is not conservative, they clearly mean that total energy does not mix linearly. Saying that total energy is non-conservative is very confusing.

     Now, the full evolution equation for the specific enthalpy in seawater takes the form:

$$\frac{Dh}{Dt} = -\frac{1}{\rho}\nabla\cdot\left(\rho F_q\right) + v\frac{Dp}{Dt} + \varepsilon_k$$

(see Eq. B19 of the latest version of TEOS10 manual with remineralization term removed).  In a turbulent ocean, neither the pressure term nor turbulent viscous dissipation can be neglected, so Graham and McDougall's (2003) assertion that the locally referenced potential enthalpy mixes linearly if one neglects viscous dissipation is inconsistent with the fact that pressure always fluctuates in a turbulent ocean.  However, it appears to be true that the first-principles expressions for the non-conservation of Θ and $\theta$ obtained by Tailleux (2010) can also be obtained by treating specific enthalpy as if it were linearly mixed, i.e., by omitting the pressure term in the above equation.  However, because Tailleux (2010) and Tailleux (2015) are rooted in a fully deductive and first-principles approach, which is not the case of Graham and McDougall (2013), the correct way to justify the assumption made by Graham and McDougall (2013) is by showing that it follows from the exact results of Tailleux(2010), not the reverse.

     We do understand the approach of Tailleux (2010, 2015) and we have shown that these papers have made errors in the development of their equations.  In the Appendix of our response to the reviewer's first review at https://doi.org/10.5194/gmd-2020-426-AC1 we show that Total Energy is not a conservative variable (as we have defined this term above).  The reason why Tailleux (2010, 2015) thought that Total Energy is a conservative variable is that these papers ignored the non-conservative nature of the $-\nabla\cdot\left(P\mathbf{u}\right)$ term.  Rather, as in well known in the thermodynamic literature, when fluid parcels mix together at a certain pressure, the thermodynamic quantity that is conserved is enthalpy.  This is the basis of the McDougall (2003) and Graham and McDougall (2013) papers.

Lines 253-256 – 'However, these expressions are written in terms of molecular fluxes, and it is not possible to use these expressions to evaluate the non-conservation in a turbulently mixed ocean'. I really don't understand where does this come from. Again, the authors appear to assume that because they do not know how to do something, this should also be the case of everybody else, which seems to me to go against the collegial nature of science. Moreover, the Navier-Stokes equations are well accepted to describe both laminar and turbulent motions, so clearly Tailleux (2010,2015)'s expressions pertain to a turbulently mixed ocean, contrary to what the authors say. What is true, however, is that the expressions remain to be linked to turbulent fluxes or microstructure measurements in order to allow for their evaluation. One way this could be done is by using expressions such as the Osborn-Cox model linking the dissipation of temperature variance to the turbulent heat diffusivity as follows:

$$\kappa \frac{|\nabla T'|^2}{\left(\frac{d\overline{T}}{dz}\right)^2} = K_T$$

On the left-hand side, the terms involve the molecular fluxes of temperature as well as the mean temperature profile, whereas on the right-hand side appears the turbulent diffusivity for the mean temperature. The authors' remarks have incited me to rework on the issue in order to show that such expressions can indeed be linked to microstructure measurements and evaluated from first principles, as I hope to show in a forthcoming publication.

The referee now seems to agree that the approaches of Tailleux (2010, 2015) "remain to be linked to turbulent fluxes ...". But, the McDougall (2003) and Graham and McDougall (2013) approach *has already been written in terms of the turbulent fluxes*. This is what several sections of those papers do. See for example, Eqn. (38) of Graham and McDougall (2013). That is, this link to turbulent mixing that referee discusses in this comment has already been done, and has been published in 2003 and in more detail in 2013.

Line 331 – What this describes is 'density salinity' – My understanding is that density salinity is always different from absolute salinity except when all the haline contraction coefficients for each of the chemical constituents are identical. May be this can be mentioned and commented upon.

The Absolute Salinity of TEOS-10 is defined to be its "density salinity"; see section A.4 of IOC et al. (2010), the paper by Wright et al. (2011) and page 169 of the summary paper of Pawlowicz, McDougall, Feistel and Tailleux (2012). This definition of Absolute Salinity was adopted primarily for two reasons. First, specific volume is the thermophysical quantity whose sensitivity to the variations in seawater composition has the most impact on ocean and climate circulation and fluxes. Second, it is possible to measure the specific volume of a liquid in an SI-traceable manner, so providing a link between Absolute Salinity and SI-traceability (such an SI-traceable route to the measurement of Practical Salinity had been lacking prior to the adoption of Absolute Salinity by TEOS-10).

Line 341 – Can the authors clarify whether the relation is actually between S* and SA, or between S* and SD (density salinity).

The Absolute Salinity is the same as "density salinity"; see the previous response above.

Lines 361-367 – Can the authors comment on the differences in computational efficiency of the equation of state between the Jackett and McDougall (1995) and Roquet et al. (2015). This information is important for ocean modellers to decide whether to switch or not to switch.

The equation of state of Jackett and McDougall (1995) is a rational function whereas that of Roquet et al. (2015) is a straightforward polynomial. While they both are approximately equally as computationally efficient for the evaluation of specific volume, the Roquet et al. (2015) is substantially more efficient when the partial derivatives (with respect to SA and CT) are calculated. Computer codes actually do more evaluations of these partial derivatives than they do of specific volume itself (for example, in the neutral physics part of the code), so the Roquet et al. (2015) form of the equation of state comes with this computationally advantage (as well as being a function of the modern salinity and temperature variables).

Lines 381-392. I agree that TEOS-10 has conclusively shown that variations in composition potentially matters for estimations of the thermal wind. However, it is also essential that the equations of motion used by ocean models be mathematically and physically well posed. As far as I understand the problem, while it is obvious that the equations of motions based on the use of reference composition salinity are well posed, it seems to me that this is not the case if we use absolute salinity (or rather density salinity). Moreover, as well as making the equations of motion ill posed, the use of density salinity also seems to screw up the energetics by introducing spurious sinks and sources of energy. As far as I am aware, TEOS-10 never wrote down a mathematically consistent set of equations based on the use of absolute salinity. I would very much like to see the qualitative considerations about the importance of the variations in composition accompanied by the authors writing down a full set of equations of motion that can be studied by mathematicians and dynamicists like me. If the authors cannot produce a mathematically well posed set of equations using absolute salinity, it seems to me that they should not promote it as a meaningful basis for ocean modelling. If model equations using absolute salinity are ill posed as I think they are, the consequence is that it is a priori impossible to be sure of how to interpret the results of McCarthy et al. (2015). To me, this is a key issue that the IOC et al. (2010) and the authors appear to have overlooked.

No ocean model has yet been published that has included the non-conservative source terms in its evolution equation for salinity. In section A.20 of IOC et al. (2010) we described how we thought it should be done. But only when an ocean modelling group decides to adopt this suggestion (or modifies it) will we, as a global community of oceanographers, learn more about this. The reviewer talks of the non-conservation of energy (but, which type of energy?). Perhaps more basic than this is the non-conservation of Absolute Salinity. Then, at some stage we will need to be adding the non-conservative dissipation of kinetic energy to the right-hand side of the CT evolution equation in ocean models. And there is yet another way that energy is not conserved in an ocean model. This is due to the fact that in the ocean interior, the most consistent interpretation of the SA and CT in an ocean model is as the thickness-weighted mean values of SA and CT (see the Temporal-Residual Mean papers). The difference between these TRM values of SA and CT and the Eulerian-mean versions also leads to the non-conservation of energy, as pointed out by Peter Killworth. Indeed, the non-linear nature of the equation of state of seawater causes many conceptual issues that we are still learning how to deal with; such as thermobaricity, cabbeling, the ill-defined nature of neutral surfaces etc.

Ocean modelling groups are not keen to add non-conservative source terms to either their SA or CT evolution equations, since this would deny these groups, and the many other folk who analyze model output, the ability to evaluate things like the meridional freshwater flux or the meridional heat flux; how would one include the effects of the non-conservative source terms in these calculations?

Line 406 – What is the way to compute Cp(S*,theta,0) using the TEOS10 software?  Can they provide the appropriate lines of code that would need to be invoked to compute it?

Since the models do not carry any non-conservative source terms in their salinity evolution equation, and any sea-surface salinity restoring boundary condition restores to S* (which is virtually the same as SR and SA at the sea surface), the GSW software should be called with the model's salinity variable (if it is a TEOS-10 ocean model) and with u_PS times the model's salinity if the ocean model is a EOS-80 based model.  So the GSW software call should use the code **gsw_cp_t_exact** with the arguments being S*, potential temperature and zero sea pressure.

Line 412 – The fact that the authors use sometimes S*, sometimes SA is confusing.

We trust this is clear now.

Lines 414-415 – May be add a physical explanation for why the temperature or rain is not treated consistently.

The interested reader can read about the difficulties that atmospheric models have in this regard by consulting the references given.

Lines 444-445 – That's the authors interpretation.  The alternative and more common interpretation is that these errors are accounted for in the estimation of errors and uncertainties affecting the simulated $\theta$ field.

We have made the point that having the ocean receive a different heat flux than the atmosphere gives it is simply not acceptable in 2021.  It is embarrassing that as a community we have lived with this discrepancy for the past century.  Fortunately, since the adoption of Conservative Temperature and TEOS-10, there is no longer any need to put up with this incorrect physics in our coupled models.

Lines 453 – To ensure that the model equations are well posed, many ocean models will assume that their salinity argument is reference salinity rather absolute salinity.  Again, I have yet to see a consistent set of equations based on absolute salinity.

The reviewer seems to have the properties of these salinity variables back-to-front.  If an ocean model were to carry Reference Salinity as its prognostic salinity variable, it would need to also carry the non-conservative source terms that represent to biological effects on Reference Salinity.  It is Preformed Salinity that obeys a conservation equation (with no source terms), whereas Reference Salinity and Practical Salinity at depth in the ocean are the result of not only advection and turbulent mixing processes, but also of the non-conservation of these variables due to the dissolution of sinking fecal pellets.  This is explained in IOC et al. (2010), Wright et al. (2011) and Pawlowicz, McDougall, Feistel and Tailleux (2012).

Lines 481-488 – I think that this paragraph is going to cause considerable confusion in the community as it seems inconsistent with the way things have been described before. First, IOC et al. (2010) says that the new equation of state is defined in terms of absolute salinity (while in fact using density salinity to estimate absolute salinity, even though the two are supposed to be somewhat different). Now, the authors appear to say that it is defined in terms of preformed salinity $S^*$, which is always numerically different from absolute salinity. Does that mean that the authors are actually already moving away from the recommendations of TEOS-10? Nothing of what the authors say about salinity in this paper makes any sense to me. I just don't understand where all this come from, and I suspect I won't be the only one. I think that it would greatly help if the authors could write the model equations that ocean modellers are supposed to solve with the proposed interpretation, may be in an appendix.

We are sorry that the reviewer finds these salinity issues confusing, and yes, we agree that he is not alone in this. We trust that the explanations given above help. The key papers regarding the several salinity variables are the papers by Pawlowicz (2010), Pawlowicz, Wright and Millero (2011), section A.4 of IOC et al. (2010), the paper by Wright et al. (2011) and the summary paper of Pawlowicz, McDougall, Feistel and Tailleux (2012).

Lines 489-492 – What is this based on exactly?

It is based on the fact that (1) the salinity variable in an ocean model is treated as a conservative variable, and (2) that this variable is restored at the sea surface in the same way that is Preformed Salinity (which is equal to both Reference Salinity and Absolute Salinity there). These are the features of Preformed Salinity. Hence the salinity variable that these models have been carrying all these years is Preformed Salinity, even if previous ocean modellers have thought that their models carried a different type of salinity. Just because there are hundreds or perhaps thousands of published papers saying that the model output is Practical Salinity does not mean that their interpretation is correct.

Lines 499-503 – "The model's salinity variable will drift towards being preformed salinity." I really don't understand why. How can the authors make such an assertion without substantiating it? For instance, could the authors write down an evolution equation for the drift that would clarify the relaxation time scale and convince us that what the authors describe has a counterpart in the mathematical world?

This assertion of ours is based on the fact that (1) the salinity variable in an ocean model is treated as a conservative variable, and (2) that this variable is restored at the sea surface in the same way that is Preformed Salinity (which is equal to both Reference Salinity and Absolute Salinity there). These are the features of Preformed Salinity. Hence the salinity variable that these models have been carrying all these years is Preformed Salinity, even if previous ocean modellers have thought that their models carried a different type of salinity.

Lines 536 – This is not how models work. Indeed, as far as I understand the issue, the temperature variable used by a model is not a matter of interpretation, it is a matter of declaration. The first step in constructing a model is to declare what its dependent variables should be. Once the variables have been declared, the next step is to decide on the evolution equations and boundary conditions that one will use to describe their temporal evolutions. To me, it is essential that models be based on precise definitions and declarations, not interpretations, so that what we do can be easily understood by our colleagues mathematicians and atmospheric scientists. I am pretty sure that mathematicians cannot understand what the authors mean by 'interpretation', which is bound to leave them very confused. My impression is that the authors use the term 'interpretation' because they want the reader to accept their

view that the physical meaning of the variables used by an ocean model is open to discussion, which seems questionable at best.

We totally disagree with this comment (and previous such comments). How could the temperature or salinity variable of an ocean model be what a person simply declares it to be? One could for example declare that the model's temperature variable is ten times the in situ temperature in degree F. Such a declaration does not make it so. Rather, the model's temperature and salinity variables must be interpreted in terms of how these variables are forced and transported in the model.

We have realized that in the case of temperature, we must not allow the ocean to receive more (or less) heat than the atmosphere delivers, and this means that it is no longer acceptable to interpret the temperature variable in EOS-80 models as being potential temperature. That this conflicts with the hundreds or perhaps thousands of published papers on ocean models does not invalidate our revised interpretation of the temperature variable of these models. This responsibility of a scientist to interpret what the model's variable is, should not be controversial.

Line 543 – I disagree that this is a conclusion. It looks much more like an opinion or assertion. It would be useful if the authors could provide the reader with some experiments to run that would enable the ocean modelling community to test its validity.

These lines of the manuscript summarize our most prominent result, and how it differs from what one reads in Griffies et al. (2016). We are disappointed that the reviewer does not agree with the thesis of our paper.

Lines 556-557 – Again, my view is that the evolution equation and boundary conditions have no bearing on the definition of a physical quantity. For this statement to be acceptable, one has to accept that the definition of a physical quantity is not independent of its assumed evolution equation and boundary conditions, in contrast to what is generally done (as far as I understand it).

We totally disagree with this comment (and the previous such comments). See our discussion above, for example, in response to the reviewer's comments on line 536. Contrary to what the reviewer says, it is indeed crystal clear that the interpretation *"of a physical quantity is not independent of its assumed evolution equation and boundary conditions …".* How could it be otherwise? For example, if an ocean modeler decided to "define" their model's temperature variable to be the number of moles of human growth hormone per kilogram of seawater, this definition does not make it so. Rather, the temperature variable that comes out of the model is a result of how the variable called temperature is treated inside the model (initial conditions, boundary conditions and evolution equation).